# Pro-Angiogenic Bioactive Molecules in Vascular Morphogenesis: Integrating Endothelial Cell Dynamics

**DOI:** 10.3390/cimb47100851

**Published:** 2025-10-15

**Authors:** Claudiu N. Lungu, Gabriela Gurau, Mihaela C. Mehedinti

**Affiliations:** Department of Functional and Morphological Science, Faculty of Medicine and Pharmacy, Dunarea de Jos University, 800010 Galati, Romania; gabriela.gurau@ugal.ro (G.G.); mihaela_hincu10@yahoo.com (M.C.M.)

**Keywords:** angiogenesis, arteriogenesis, vascular morphogenesis, VEGFR, cellular chirality

## Abstract

During embryonic development, angiogenesis and arteriogenesis are responsible for vast growth and remodeling. These processes have distinct mechanisms, like budding, cord hollowing, cell hollowing, cell wrapping, and intussusception. This review discusses the diversity of morphogenetic mechanisms contributing to vessel assembly and angiogenic sprouting in blood vessels and how molecular pathways regulate some complex cell behaviors concerning the VEGFR pathway. Also, a particular part is dedicated to the HIF 1α gene. The key components of the VEGFR pathway are VEGF receptors VEGFR1, VEGFR2, and VEGFR3. VEGFR2 plays a central role in vascular morphogenesis. VEGF is the primary ligand involved in angiogenesis and arteriogenesis. Various types of VEGF are being studied in terms of their therapeutic use. The ultimate goal of the vascular morphogenesis study is to enable the development of organized vascular tissue that presumably might be used to replace the diseased one. Cellular chirality—the intrinsic “handedness” of cells in movement, structure, and organization—plays a crucial role in angiogenesis, the process by which new blood vessels develop from old ones. This chiral activity is essential for the directed and patterned organization of endothelial cells during vascular formation and remodeling. In angiogenesis, cellular chirality directs endothelial cells to adopt specific orientations and migratory patterns, which are crucial for the formation of functionally organized blood vessels that provide tissues with the necessary nutrients and oxygen. Cellular chirality in this environment is affected by multiple mechanisms, including VEGF/VEGFR signaling, mechanical pressures, interactions with the extracellular matrix (ECM), and cytoskeletal movements. Lately, researchers have focused on the molecular control of blood vessel morphogenesis, the study of signaling circuitry implied in vascular morphogenesis, the emerging mechanism of vascular stabilization, and helical vasculogenesis driven by cell chirality.

## 1. Introduction

The number of papers published concerning vascular morphogenesis (which was used as a search keyword) has increased exponentially. Though only one article on this subject was published in 1948, over the last five years, the number of documents published in PubMed has exceeded 2689 (as of 5 October 2023). In this paper, we review cellular and tissue indices discussed in the literature on vascular morphogenesis. Data regarding this review were curated using the PubMed database. The words used to generate the data were vascular morphogenesis, cellular indices, and tissular indices.

As expected, considerable data were generated. The ultimate goal of the vascular morphogenesis study is to enable the development of organized vascular tissue that might replace the disease-affected one [1,2].

Growing new vessels requires a coordinated cellular response to growth. This response is sensed and triggered by cell surface receptors, which are responsible for activating an intracellular cascade that initiates migration and controls cell growth. While the dominant molecular mechanisms have been identified, the distinct and detailed interactions remain unknown [3].

Experiments were primarily performed using mouse and zebrafish models. A series of pathways has been identified. These pathways include VEGF-VEGFR, Notch-DSL, Tie–Angiopoietin, VE-cadherin, and Ephrin–Eph. Other signaling pathways contributing to further vascular remodeling include plexins, TGF-β, PDGF, and integrins [4]. In the following paragraphs, the VEGFR pathway, together with its receptors, is briefly discussed.

## 2. Molecular Regulators of Vascular Morphogenesis

### 2.1. VEGF–VEGFR Signaling Axis. VEGFR Signaling Pathway in Vascular Morphogenesis

Coordinated cellular response is critical for the development of new blood vessels. The cellular response involves the cell surface receptors. Even if the dominant molecular motifs have been determined, how they interact is still not understood entirely [5].

The dominant pathways include the VEGF-VEGFR, Notch-DSL, Tie–Angiopoietin, VE-cadherin, and Ephrin–Eph pathways [6,7].

The VEGF pathway is a critical regulator of the angiogenesis process. The VEGF/VEGF-receptor axis comprises multiple ligands and receptors with overlapping and distinct ligand-receptor binding specificities. The VEGF-receptor pathway activates signaling processes that promote the growth, migration, and survival of endothelial cells from preexisting vasculature. Additionally, VEGF mediates vessel permeability [8].

VEGF is crucial in vasculogenesis and angiogenesis. Its activity is primarily restricted to vascular endothelial cells, although it also affects other cell types (e.g., by stimulating monocyte/macrophage migration). In vitro, VEGF stimulates endothelial cell mitogenesis and cell migration [9].

All members of the VEGF molecular family elicit a cellular response by binding to tyrosine kinase receptors (the VEGFRs) on the cell surface, causing dimerization and activation through transphosphorylation [10].

Five VEGF molecules are known: VEGF A-E. VEGFA binds to VEGFR-1 (Flt-1) and VEGFR-2 (Flk-1/KDR). VEGF-B stimulates VEGFR1 (Flt-1). VEGF-C interacts with VEGFR-2 (Flk-1/KDR) and VEGFR-3 (Flt-4). VEGF-D stimulates VEGFR-2 (Flk-1/KDR) and VEGFR-3 (Flt-4). Lastly, VEGFR-E enables VEGFR-2 (Flk-1/KDR) [11,12,13,14,15] (Figure 1).

The VEGFA gene produces VEGF-A, a PDGF/VEGF family member. A disulfide-linked homodimer is encoded [16,17].

When vascular endothelial growth factor receptors (VEGFRs) are activated, they recruit PI3K, which produces PIP3 and kicks off the PI3K/Akt pathway. Akt, also called protein kinase B, plays a central role here, steering cell growth, survival, and migration. At the same time, VEGFR stimulation sets off the MAPK cascade, including ERK, which is essential for endothelial cell proliferation and movement.

VEGFR signaling doesn’t stop at survival and migration—it also shapes gene expression. For instance, it can upregulate endothelial nitric oxide synthase (eNOS) and matrix metalloproteinases (MMPs), both of which support vascular growth, cell motility, and the formation of new vessel structures. Altogether, the VEGF–VEGFR interaction orchestrates a program of endothelial growth, migration, and survival, ultimately driving angiogenesis.

Of course, this system is closely regulated. Soluble VEGFRs (sVEGFR1 and sVEGFR2) bind VEGF before it reaches membrane-bound receptors, reducing the signal. Additionally, neuropilins (NRP1 and NRP2) co-receptors regulate VEGF activity. They modify VEGF’s angiogenesis promotion and VEGFR2 binding, intensifying the response [18,19]. The following briefly describes some relevant VEGFR 1–3.

### 2.2. VEGFR1

This system is strictly regulated. VEGF is bound by soluble VEGFRs (sVEGFR1 and sVEGFR2) before it reaches membrane-bound receptors, reducing the signal. As co-receptors, neuropilins (NRP1 and NRP2) regulate VEGF activity. VEGF’s angiogenesis promotion and VEGFR2 binding are adjusted, intensifying the response [18].

VEGFR1 may have a role as a negative regulator of embryonic angiogenesis. It promotes the PGF-mediated proliferation of endothelial cells. It has a very high affinity for VEGFA and relatively low protein kinase activity. Also, it functions as a negative regulator of VEGFA signaling. Furthermore, VEGFR1 stimulation led to phosphorylation of the regulatory subunit of phosphatidylinositol 3-kinase (PIK3R1). VEGFR1 is also implicated in the activation of MAPK1/ERK2, MAPK3/ERK1, the MAP kinase signaling pathway, and the AKT1 signaling pathway [19].

VEGFR1 promotes endothelial tubule branching in an organotypic angiogenesis model via a mechanism that requires Rab4A and alphavbeta3 Integrin. A recycling pathway regulated by Rab4A is an effector of VEGFR1 during the branching morphogenesis of the vasculature [19]. The VEGFR1 gene encodes a receptor tyrosine kinase and a secreted splice variant. Epidermal growth factor/fibroblast growth factor two (EGF/FGF2)-mediated VEGFR1 induction is mediated via the functional interaction of transcription factors ETS1 and hypoxia-inducible factor 2 alpha (HIF-2alpha). Mechanistic analyses revealed that EGF/FGF2 signaling induces ETS1 expression in endothelial cells, increases HIF-2alpha protein level without hypoxia, and recruits both protein C-ets-1 (ETS1) and HIF-2alpha to the VEGFR1 chromatin domain [20]. VEGFA regulates embryonic angiogenesis through vascular endothelial growth factor receptor 2 (VEGFR2) expressed in the endocardium. It is shown that VEGFR1 produced in the endocardium negatively regulates embryonic coronary angiogenesis by limiting the Vegf-Notch signaling [18]. VEGFR1 and VEGFR2 expressions are upregulated during copy number variation (C.N.V.) pathogenesis. Both MF1 and DC101 significantly suppressed C.N.V. at 50 mg/kg [21]. The role of VEGF receptor 1 (VEGFR1) signaling in angiogenesis and tissue growth in an endometriosis model showed that VEGFR1 is implicated in this process. VEGFR1 signaling in host-derived cells is essential for cell growth [22,23]. VEGFR1 exists in different forms, resulting from alternative splicing of the same gene. Moreover, sVEGFR-1 and sFlt1-14 are additional in angiogenesis [24]. A recycling pathway regulated by Rab4A is a critical effector of VEGFR1 during the branching morphogenesis of the vasculature [25]. Tyrosine-protein kinase acts as a cell-surface receptor for VEGFA, VEGFC, and VEGFD. The binding of vascular growth factors to isoform one leads to the activation of several signaling cascades. Activation of PLCG1 stimulates the production of the cellular signaling molecules diacylglycerol and inositol 1,4,5-trisphosphate. Consequently, they stimulate protein kinase C [26]. Angiogenesis stimulates endothelial cells (E.C.) by various cytokines and growth factors [27]. The biological effects of VEGF are mediated by two tyrosine kinase receptors, Flt-1 (VEGFR-1) and K.D.R. (VEGFR-2). VEGF is essential for the early development of the vasculature [28].

The properties of VEGFR1 are summarized in Table 1:

For its odd structure and function, VEGFR1 (Flt-1) is commonly called a negative regulator of angiogenesis. Though it binds VEGF-A more strongly than VEGFR2, its tyrosine kinase activity is lesser. VEGFR1 acts as a decoy receptor, trapping VEGF-A and reducing its access to VEGFR2, the main pro-angiogenic signaling pathway. Through competitive ligand binding, VEGFR1 alters VEGF-mediated responses, especially during embryonic and early vascular development. Alternatively spliced VEGFR1 (sVEGFR1 or sFlt-1) neutralizes VEGF-A and PlGF in the extracellular space, confirming its decoy role [41].

Despite its inhibitory effects, VEGFR1 is not necessarily anti-angiogenic. In inflammatory or hypoxic settings, paracrine signaling recruits and activates monocytes and macrophages, indirectly boosting angiogenesis. VEGFR1 and VEGFR2 produce heterodimers that indirectly affect PI3K–Akt and MAPK. While VEGFR1 does not directly affect mitogenic signaling, transcription factors such as HIF-1α and ETS1 closely regulate its expression and activity, particularly during cellular stress [42].

The interplay between VEGFR isoforms is critical for vascular morphogenesis. VEGFR2 mainly transmits pro-angiogenic signals, while VEGFR3 is central to lymphangiogenesis. VEGFR1, meanwhile, both competes with VEGFR2 for ligands and helps shape the inflammatory and metabolic environment that supports vascular remodeling. Its role is therefore best understood as context-dependent modulation of angiogenesis, rather than as a simple antagonist.

The next section focuses on VEGFR2, a receptor central to vasculogenesis.

### 2.3. VEGFR2

VEGFR-2 is a 210–230 kDa glycoprotein expressed in vascular endothelial cells and binds VEGF-A. VEGFR-2 is closely related to VEGFR-1, for they have similar and specific ligands. Furthermore, VEGFR-2 is a highly active kinase in contrast to VEGFR1, an impaired tyrosine kinase receptor. The signaling pathways, Y1175 and Y1214, are the main autophosphorylation sites of the human VEGFR-2 when VEGF is bound [43,44,45].

VEGFR-2 is a part of the VEGF family. It is essential for both developmental and reparative angiogenesis [43]. VEGF activates VEGFR2, situated in the endothelial cell membrane. VEGFR2 interacts with VRASP(), PLCγ, ScK, Cdc42, Src, and subsequently PI3K. VRAP and Sck do not engage in further interactions with other signaling molecules. PLCγ interacts with diacylglycerol and inositol trisphosphate. Cdc42 engages with p38. PI3K activates PIP3. IP3 engages with Ca+ and PKC, initiating a cascade in which SPK connects with Ras, and then engages with Raf-1, so boosting M.E.K., which interacts with ERK. ErK enhances vascular proliferation through its interaction with DNA. Calcium ions (Ca^+^) are activated by inositol trisphosphate (IP3), which then engages with calnexin (CALN), facilitating the generation of nitric oxide synthase (NOS) that interacts with cyclooxygenase-2 (COX2); the COX2 pathway subsequently triggers the synthesis of prostaglandin I2 (PGI2). FAK activation subsequently activates focal adhesion and cellular migration. Additionally, p38 activates M.A.P., thereby enhancing actin rearrangement through its interaction with HSP27, ultimately resulting in cell migration. Additionally, AktPKE activates eNos, Casp9, and Bad, which further enhances cell permeability, migration, and survival. Numerous studies indicate that VEGFR-2 is the principal mediator of VEGF-induced responses in endothelial cells. It serves as a vital signal transducer in both physiological and pathological angiogenesis [26,46].

Regarding the Notch signaling pathway, Delta notch or Seratt-like ligand, with the help of O-fucosylpeptide 3-beta-N-acetylglucosaminyltransferase (Fringe), stimulates the Notch receptor that further promotes the Notch intracellular domain. The Notch domain simulation acts on DNA that additionally produces Hes 1/5 (hes family bHLH transcription factor 1), Hey (hes-related family bHLH transcription factor with YRPW motif 1), PreTα (pre-T cell antigen receptor alpha), and NRAP (Notch-regulated ankyrin repeat-containing protein), respectively. On C.S.L. (recombination signal binding protein for immunoglobulin kappa J region like), a series of activators -MAML, H.A.T.s, SKIP, and a series of Co suppressors—SMRT, CtBP, Gro/TLE, C.I.R., SHARP, HDAC, ATXN1L, Hes1, and NRARP, respectively. The Notch signaling system is crucial to embryonic development [47]. Although weaker, VEGF/VEGFR1 signaling can converge with VEGFR2 to activate several downstream pathways that promote endothelial cell proliferation [48]. The VEGFR2-regulated PI3K–AKT–mTOR axis is essential for cell survival, proliferation, anti-apoptotic signaling, and vascular permeability. Another important method involves PLCγ-mediated PKC activation, which activates the ERK pathway and a PKC-dependent signaling branch. VEGFA/VEGFR2 signaling activates p38 MAPK, which polymerizes actin and drives endothelial cell motility [49].

Together, these interconnected pathways outline the VEGFA/VEGFR2 signaling network, which has been extensively studied in the context of VEGFA-165 and its effects on endothelial cells through VEGFR2 [50].

Signaling molecules, categorized into enzymes, receptors, and transcription factors, undergo contextual activation or deactivation downstream of VEGFA/VEGFR2 signaling, influencing angiogenesis. The interactive VEGFA/VEGFR2 signaling network has a critical role in [51,52]. FAK regulates VEGFR2 and several other angiogenesis-related genes while influencing VEGFR2 and VEGF protein expression in TNBC cells [53]. Despite efforts, clinical success in promoting sprouting angiogenesis in the skeletal muscles of individuals with peripheral artery disease has not been achieved [54,55].

Cell signaling governs cellular behavior and is subject to tight spatiotemporal regulation. Signaling output is modulated by specialized cell membranes and vesicles containing unique lipids and protein combinations. The phosphatidylinositol 4,5-bisphosphate (P.I. (4,5)P2), an essential component of the plasma membrane and other subcellular membranes, is involved in multiple cellular processes [56]. Rap1a and Rap1b, two closely related small GTP-binding proteins, are essential for angiogenesis in vivo and for proper endothelial VEGF responses. These findings highlight the role of Rap1 in VEGF signaling within endothelial cells [57].

Mouse heart coronary channels are formed by sinus venosus, endocardium, and proepicardium progenitor populations during embryonic development [58]. Hypoxia may also affect SOX17- and VEGFR2-mediated signaling during coronary vessel development [24]. SSA also inhibits VEGFR2-mediated signaling, delaying tumor growth and angiogenesis [59].

Under hypoxic conditions, VEGF stimulates endothelial cells to sprout, proliferate, and migrate, processes that collectively drive angiogenesis.

Inducing arteriogenesis after cardiac or cerebral arterial occlusion can reduce ischemia and improve disease outcomes, with endothelial VEGF receptor 2 (VEGFR2) signaling playing a crucial role in both processes [60]. Identifying angiogenic factors, such as perlecan, in vertebrate development enhances our understanding of the molecular basis of angiogenesis and may inform therapeutic approaches based on angiogenesis [61].

Moreover, focal adhesion kinase (FAK) is essential in embryonic angiogenesis, governing endothelial cell (EC) survival and barrier functions via both kinase-independent and -dependent mechanisms. EC-specific tamoxifen-inducible FAK knockout and FAK-defective (K.D.) mutant knockin mice were created to examine the role and kinase activity of FAK in adult angiogenesis. The absence of FAK or its kinase activity diminished endothelial cell proliferation and migration, suggesting that FAK predominantly functions as a kinase in regulating adult endothelial cell-mediated angiogenesis. Subsequent research using mouse E.C. line MS1 cells demonstrated that FAK regulates VEGFR2 expression, necessitating both FAK activity and nuclear localization regulate VEGFR2 transcription and its nuclear translocation [62]. VEGF, a crucial regulator of angiogenesis, and its receptors, VEGFR1, VEGFR2, and Neuropilin1 (NRP1), are currently targeted in therapeutic strategies for vascular disease. NRP1 is critical in vascular morphogenesis [63]. In the cardiovascular system, VEGF binding to VEGF receptor 2 (VEGFR-2) promotes blood vessel development [64]. However, the activated receptor signals to discrete downstream pathways, and coreceptors, as well as distinct VEGF isoforms, modulate the equilibrium between these pathways [65]. Receptor-interacting protein kinase 3 (RIPK3) is a multifunctional intracellular protein recognized as a vital component of the necroptosis-programmed cell death pathway [66]. Some VEGFR2 properties are listed in the table below (Table 2):

The relevant properties of VEGFR3 are discussed in the following paragraph.

### 2.4. VEGFR3

Compared with VEGF-A activation of VEGFR2, VEGF-C-induced VEGFR3 activation led to a more extensive A.K.T. activation, whereas activation of ERK1/2 displayed a distinctly different kinetics [75].

Tyrosine-protein kinase acts as a cell-surface receptor for VEGFC and VEGFD. It plays an essential role in adult lymphangiogenesis and the development of the vascular network and the cardiovascular system during embryonic development. Signaling by activated FLT4 leads to enhanced production of VEGFC [76,77]. Phosphorylation in response to H_2_O_2_ is mediated by a process that requires S.R.C. and PRKCD activity. Phosphorylation at Tyr-1068 is required for autophosphorylation at additional tyrosine residues [20].

Elevated VEGFR2 activity in postnatal retinas following VEGFR3 deletion or VEGFR3 silencing in cultured endothelial cells reduced vascular endothelial cadherin localization at cell–cell junctions. Simultaneous deletion of VEGFR2 prevented VEGF-induced excessive vascular leakage in Vegfr3-deficient mice. VEGFR3 limits VEGFR2 expression and pathway activity, thereby preventing excessive vascular permeability in both quiescent and angiogenic blood vascular endothelial cells [78,79]. The impact of VEGFR3 on lymphatic capillary junctions remains incompletely understood, as excessive VEGFR2 signaling can remodel and seal these junctions [78]. Knockdown of FLT4 in human lymphatic endothelial cells results in impaired NOTCH1 expression and activation, with the overexpression of NOTCH1 rescuing junction formation and interstitial molecule absorption in Flt4 knockout vessels [61,80,81,82,83]. Genetic evidence suggests that VEGFR3 regulates early vessel branching and filopodia formation in the mouse brain, likely mediating the brain vascular phenotype [84,85,86]. During embryonic development, angiogenesis is initiated as mesoderm-derived mesenchyme cells differentiate into angioblasts that express VEGFR-2 [87,88,89,90]. Hypoxia induces angiogenesis, and the injection of VEGFA enhances angiogenesis in animal models. Nonetheless, clinical trials did not reproduce the encouraging outcomes shown in animal models, perhaps due to the methods of delivery and VEGFA’s capacity to enhance arterial permeability. Recent trials validate the safety of VEGFA administration in humans, facilitating advancements in pro-angiogenic treatment strategies [91,92,93,94,95]. Various VEGFA isoforms, including VEGFA111, VEGFA121, VEGFA145, VEGFA148, VEGFA162, and VEGFA165, are commercially available and being tested [96,97,98,99,100,101].

In the table below, some properties of VEGFR3 are summarized (Table 3).

Key Takeaways—VEGFR Signaling Axis:VEGFR2 is the principal mediator of endothelial proliferation and sprouting angiogenesis.VEGFR1 modulates angiogenesis through decoy activity and inflammatory recruitment.VEGFR3 primarily governs lymphangiogenesis and stabilizes VEGFR2 signaling.Crosstalk and ligand competition among VEGFRs fine-tune angiogenic outcomes.

### 2.5. HIF-1α Gene

Hypoxia-inducible factor 1 subunit alpha (HIF-1α) plays a central role in both cellular and systemic responses to low oxygen conditions. The HIF1A gene encodes the alpha subunit of the HIF-1 transcription factor, which dimerizes with the beta subunit to form a functional complex. Acting as a master regulator, HIF-1α activates a wide range of genes involved in energy metabolism, angiogenesis, apoptosis, and oxygen transport, thereby enabling cells to adapt metabolically to hypoxia [114].

HIF-1α has distinct roles in different biological contexts. During embryonic development, it is indispensable for vascular formation. In cancer, it drives tumor angiogenesis by promoting the growth of new blood vessels. In ischemic diseases, it contributes to vascular remodeling and adaptation to reduced oxygen supply.

HIF-1α plays a role in the pathophysiology of conditions related to inadequate blood supply, such as ischemic heart disease. H1F-1α variants are alternatively spliced transcript variants that encode different isoforms of H1F1α. Its main product is Hypoxia-inducible factor 1-alpha with its crystallographic structure 4h6j [115]. The Aa sequence is the following (Uniprot ID Q16665) HIF1 is a heterodimeric basic helix-loop-helix structure consisting of two subunits: HIF1A (the alpha subunit) and the aryl hydrocarbon receptor nuclear translocator (Arnt, the beta subunit). HIF1A comprises a primary helix-loop-helix domain near the C-terminal, followed by two P.A.S. (PER-ARNT-SIM) domains and a P.G.G. (PAS-associated C-terminal) domain. Additionally, the HIF1A polypeptide contains a nuclear localization signal motif, two transactivating domains (CTAD and NTAD), and an intervening inhibitory domain (I.D.) capable of repressing the transcriptional activities of CTAD and NTAD. Although three HIF1A isoforms are generated by alternative splicing, isoform1 is considered the canonical structure and has been extensively studied in terms of structure and function [116].

These genes are pivotal in processes such as angiogenesis and erythropoiesis, facilitating increased oxygen delivery to hypoxic areas. HIF-1 also orchestrates the transcription of genes vital for cell proliferation, survival, and glucose and iron metabolism. In response to fluctuating oxygen levels, HIF-1 undergoes conformational changes and binds to hypoxia-responsive elements (HREs) on gene promoters, thereby initiating transcription.

Currently, HIF-1α is shown to be implied in proteins involved in cellular responses to changes in oxygen concentration. Extended exposure to hypoxia triggers various cellular mechanisms in skeletal muscles to compensate for limited oxygen levels. One such mechanism involves upregulating specific hypoxia-inducible genes, including vascular endothelial growth factor (VEGF). The VEGF gene is crucial for angiogenesis in skeletal muscles, as evidenced by reduced capillarity upon its deletion in mice. While hypoxia-inducible factor-1alpha (HIF-1alpha) is known to activate VEGF gene transcription by binding to its hypoxic response element (H.R.E.) on the promoter, additional pathways may also contribute to VEGF upregulation during acute or prolonged hypoxia. These pathways, stimulated during hypoxic exposure, may involve inflammation, potentially related to the generation of reactive oxygen species or changes in cellular energy status, as indicated by AMP kinase activity. These mechanisms provide various means of VEGF regulation in long-term hypoxic conditions, such as those encountered at high altitudes. This review will examine the cellular signals triggered by hypoxic exposure that can enhance myocyte VEGF expression, including decreased intracellular oxygen levels, skeletal muscle inflammation involving cytokines and oxidative stress, and increased AMPK activity and adenosine levels associated with reduced cellular energy potential [117].

Some recent findings include pancreatic cancer and hif-1α. Pancreatic cancer, a leading cause of cancer-related deaths worldwide, thrives in a highly hypoxic tumor microenvironment. HIF-1α plays a central role in the carcinogenesis and progression of pancreatic cancer. Researchers have explored how HIF-1α regulates tumorigenesis and progression in pancreatic cancer. Targeting HIF-1α and its signaling pathways could hold promise as a therapeutic approach for pancreatic cancer [118]. Other studies include retinoic acid-induced differentiation. A study examined HIF-1α’s role in retinoic acid-induced SH-SY5Y cell differentiation under normoxic circumstances and decreased serum levels. Lower serum concentrations kept cells in G0, encouraging differentiation [119].

HIF-1α has also been implicated in adipose tissue remodeling, although the precise mechanisms and consequences of its involvement in this process are still under investigation [120].

In cancer, HIF-1α activation exerts a major influence on cellular metabolism. It enhances glycolysis in tumor cells while simultaneously impairing mitochondrial function, reshaping how cancer cells generate and utilize energy [121]. In peripheral artery disease (P.A.D.), gene therapy research has primarily focused on enhancing blood flow and tissue repair rather than directly targeting HIF-1α. Several randomized controlled trials have investigated the use of growth and angiogenic factors in this context [122]. Another area of study involves examining HIF-1α expression and release following endothelial injury. For example, researchers measured HIF-1α levels in endothelial cells treated with DMOG, a prolyl hydroxylase inhibitor, to better understand how HIF-1α dynamics contribute to vascular repair processes [123].

Direct control of HIF-1α activity has been tested in experiments. The intramuscular delivery of AdCA5, an adenovirus containing a constitutively active version of HIF-1α, improved perfusion and arterial remodeling in a limb ischemia model. These findings indicate that HIF-1α, which regulates angiogenic growth factors, may be a suitable therapeutic target for ischemia illness. However, clinical studies have not consistently replicated these advantages, which may explain some trials’ placebo effects [124,125,126,127].

Overall, the latest developments regarding the VEGFR pathway are summarized in the table below (Table 4):

HIF-1α is a key transcription factor that induces angiogenesis and adapts cells to low oxygen levels. PHDs hydroxylate HIF-1α under normoxic circumstances, causing ubiquitination and proteasomal degradation via the von Hippel-Lindau (VHL) E3 ubiquitin ligase complex. Hypoxic conditions, such ischemic tissues, solid tumors, or growing vasculature, impede the degradation process, allowing HIF-1α to accumulate, translocate into the nucleus, and dimerize with HIF-1β. The transcriptional complex activates genes like VEGFA, PDGF-B, ANGPT2, SDF-1, and MMP2 by binding HREs at target gene promoters, promoting endothelial activation, vascular sprouting, and extracellular matrix remodeling [142].

VEGFA is the main mediator of HIF-1α-driven angiogenesis. HIF-1α activation boosts VEGFA mRNA and protein expression in hypoxic endothelial, stromal, and tumor cells, leading to VEGFR2 stimulation in adjacent cells. PI3K–Akt and MAPK–ERK signaling cascades are activated, boosting proliferation, migration, and tube formation—signals of new blood vessel development. Besides angiogenesis, HIF-1α optimizes endothelial metabolism by increasing glycolysis and decreasing mitochondrial oxygen consumption, optimizing vascular function under oxygen-deprived circumstances [143].

Therapeutically, HIF-1α is a viable upstream target for angiogenesis modulation. Increasing HIF-1α expression can enhance tissue perfusion and vascular regeneration in ischemia situations such PAD, MI, stroke, and chronic wounds. In rodent and porcine models of limb ischemia and myocardial infarction, gene therapy vectors of constitutively active HIF-1α (e.g., AdCA5) and small-molecule PHD inhibitors (e.g., DMOG, Roxadustat) improved capillary density, blood flow, and functional recovery [144]. These encouraging results have led to early-phase clinical trials, though translation to the clinic remains challenging.

Major challenges arise from species-specific HIF-1α regulation and angiogenic response. HIF-1α overexpression in rats leads to significant neovascularization, whereas human tissues may have less pronounced effects due to factors such epigenetic changes, baseline vascular density, or age-related endothelial dysfunction. Therapeutic use is complicated by delivery mechanisms. Adenoviruses and AAVs are useful in preclinical settings but have immunogenicity, transitory expression, and limited tissue selectivity in humans. Although safer, plasmid-based techniques sometimes have low transfection effectiveness and limited target organ expression [145].

Another risk is off-target angiogenesis. Uncontrolled HIF-1α activation can cause vascular damage, worsening diseases like retinopathy, or activate tumor vasculature. Overexpression of HIF-1α in solid tumors can lead to a hostile microenvironment, resistance to treatment, and enhanced metastatic potential. Thus, therapeutic treatments must balance pro-angiogenic benefits in ischemic tissues with pathological angiogenesis elsewhere. Current research targets tissue-specific and hypoxia-inducible promoters to localize HIF-1α activity while limiting systemic exposure [146].

Finally, the pleiotropic nature of HIF-1α adds complexity. Beyond VEGF regulation, HIF-1α regulates genes involved in glucose metabolism, erythropoiesis, inflammation, and apoptosis, such as GLUT1, PFK1, EPO, IL-1β, CXCL12, and BNIP3. Thus, pharmacological manipulation may cause unintended effects or feedback loops. Prolonged HIF-1α activation can cause vascular leakiness by upregulating ANGPT2 and MMPs, leading to tissue edema or bleeding [147].

Despite its complicated regulation, targeting HIF-1α with precision is a viable therapeutic approach. New tactics use new delivery and regulatory technology to optimize its activity. Nanoparticle delivery, CRISPR-mediated gene regulation, and synthetic biology techniques like hypoxia-sensitive gene switches are used to control HIF-1α expression with precise spatial and temporal control. Parallel efforts are exploring the stabilization of HIF-1α through non-viral agents with improved pharmacokinetics and reduced immunogenicity. Furthermore, the use of biomarkers—such as circulating VEGF levels, lactate, or HIF-1α-responsive gene signatures—may enable stratification of patients most likely to benefit from HIF-1α-based therapy [148].

Lastly, the discrepancy between robust outcomes in animal models and inconsistent effects in human trials highlights the need for more effective translational models. Humanized tissue constructs, organ-on-chip systems, and patient-derived endothelial cells provide new platforms for evaluating HIF-1α therapeutics under physiologically relevant conditionsClinical studies will require longitudinal imaging and perfusion evaluation methods like contrast-enhanced ultrasonography and MRI angiography to determine treatment efficacy [149].

Future success will likely depend on context-aware, precision-targeted strategies that leverage HIF-1α’s upstream position without triggering systemic side effects or uncontrolled vessel growth [23].

Key Takeaways—HIF-1α in Angiogenesis:

HIF-1α is a hypoxia-activated transcription factor that upregulates VEGF and related genes.

It links oxygen sensing with metabolic adaptation and vascular outgrowth.

Therapeutically, it holds promise for ischemic disease, but faces translational barriers due to delivery, specificity, and off-target risks.

### 2.6. Notch Signaling in Vascular Morphogenesis

Intercellular communication via the conserved Notch signaling pathway is crucial to vascular formation. Notch controls endothelial cell fate specification, sprout production, and vascular maturation during angiogenesis in tight collaboration with the VEGF signaling axis. Notch regulates endothelial cell differentiation into tip or stalk cells during early vessel expansion, avoiding uncontrolled branching and ensuring spatial patterning. This precise management is necessary for a well-perfused and efficient vascular network [150].

Four single-pass transmembrane receptors (Notch1–Notch4) and five classical ligands—Delta-like ligands (Dll1, Dll3, Dll4) and Jagged proteins (Jagged1 and Jagged2)—mediate Notch Dll4 and Notch1 are crucial for vascular endothelial cells. Two proteolytic cleavages occur when a Notch ligand on one cell connects to a receptor on a nearby cell, signaling. The γ-secretase-mediated second cleavage frees the Notch intracellular domain (NICD) from the plasma membrane. The DNA-binding protein CSL (CBF1/Suppressor of Hairless/LAG-1) and co-activators such MAML proteins connect with the NICD in the nucleus. This complex displaces transcriptional repressors, activating Notch target genes Hes1, Hey1, and Hey2, which govern endothelial growth, proliferation, and communication [151,152].

Notch signaling determines whether endothelial cells at the head of a vascular sprout are tip or stalk cells during angiogenesis. Tip cells migrate, extend actin-rich filopodia, exhibit strong VEGFR2, and have low Notch activity. Hypoxic tissue cells respond robustly to VEGF-A gradients. VEGF-A binding to VEGFR2 on tip cells triggers Notch signaling in surrounding endothelial cells via Dll4 activation. This lateral inhibition lowers nearby cells’ VEGFR2 and other pro-migratory gene expression, inhibiting tip-cell destiny. They develop into stalk cells, which proliferate, create the vessel lumen, and adhere to each other. This mechanism ensures that a local cluster has one tip cell and supporting stalk cells. Tip–stalk patterning ensures evenly spaced, sturdy vessel branches [40,150].

Notch affects arteriovenous specification and vascular stabilization beyond sprouting. Notch signaling upregulates ephrinB2 and represses venous markers like EphB4 to enhance arterial identity in embryos. Notch1 and Notch4 are crucial in arterial endothelial cells. Additionally, Notch collaborates with PDGF and TGF-β pathways to recruit pericytes and vascular smooth muscle cells, stabilizing artery walls. It helps mature arteries endure hemodynamic stress and preserve selective permeability [151].

The interplay between VEGF and Notch signaling is tightly regulated through systems of feedback. Endothelial cells produce Dll4 in response to VEGF-A, while Notch activation suppresses VEGFR2 transcription, lowering sensitivity. The negative feedback balances pro- and anti-sprouting signals. Dll4 expression and Notch activation may paradoxically reduce sprouting by inducing broad stalk cell specification in malignancies and ischemic tissues under high VEGF circumstances. Blocking Dll4–Notch signaling therapeutically induces non-productive angiogenesis, resulting in disordered, leaky, and poorly perfused vessels. Abnormal vasculature may promote tumor hypoxia and treatment sensitivity [152,153].

Additional mechanisms improve Notch signaling. In certain cell types, Jagged1 can oppose Dll4, affecting Notch activity. Cytoskeletal dynamics, cell–matrix adhesion, and endothelial migration are also affected by Wnt, BMP, and integrin cross-talk. Mechanical forces affect ligand and receptor expression, with high shear regions upregulating Notch1 to reinforce arterial identity and endothelial quiescence. This mechanosensitivity complicates in vivo control [154].

Notch signaling is essential for developmental and pathological angiogenesis, but therapeutic targeting is difficult. Notch suppression systemically can cause gastrointestinal toxicity and arteriovenous malformations. Numerous techniques are being investigated, such as selective antibody-based Dll4 blockage, enhanced γ-secretase inhibitors, and gene therapy for tissue-specific Notch system regulation [155].

In conclusion, Notch signaling strongly integrates with VEGF and other pathways to control vascular patterning and endothelial fate during angiogenesis. Precision regulation creates structurally stable and functionally perfused vessels. Pathological angiogenesis caused by dysregulated Notch activation can lead to cancer, blindness, and vascular abnormalities.

Understanding the nuanced roles of Notch in endothelial biology not only provides insight into fundamental developmental processes but also opens up avenues for therapeutic modulation of angiogenesis in diverse clinical settings [156].

### 2.7. VEGF Mimetics

Ischemic disease (heart failure, strokes, peripheral artery disease) and also some degenerative conditions are known to be the consequence of ultimately poor blood supply. In this respect, treating these pathologies using pro-angiogenic targeting molecules is tempting. As expected, VEGFA has the predominant role in vessel formation, growth, and branching [157,158,159,160,161]. In this respect, VEGFA is the prototype of a pro-angiogenic molecule. VEGFA acts primarily on VEGFR2. Also, due to its complex roles, VEGFR1 is stimulated by VEGFA. Furthermore, stimulation of VEGFR1 and VEGFR2 is a potential pro-angiogenic therapy [162,163]. Therapeutic angiogenesis shows excellent potential, as evidenced by the experimental results. Sadly, no F.D.A.-approved molecule exists [164]. Due to extensive research in this area, pro-angiogenic molecules are well categorized. Angiogenic proteins, gene therapy, peptide-based drugs, and small organic molecules are all well-established areas of research [165]. Peptides, small molecules of roughly 50 amino acids, are of particular interest. Unlike bigger proteins, peptides can function physiologically without tertiary or quaternary structures. Their amino acid content can be easily altered to boost angiogenic action. Peptides can be coupled with organic molecules or changed at the primary or secondary structural level to improve ADME. Pro-angiogenic peptides can be produced and administered in vast numbers due to their small size and simplicity, making them promising angiogenesis stimulators. (Table 5)

Gene therapy for promoting angiogenesis involves introducing genetic material into cells to stimulate the formation of new blood vessels. This approach is explored for various medical conditions, including ischemic diseases, wound healing, and cardiovascular disorders. Here are some examples of pro-angiogenic gene therapies. Some gene therapies are listed in the table below (Table 6):

## 3. Structural Mechanisms of Vascular Morphogenesis Mechanistic Basis of Angiogenic Morphogenesis: Integration of VEGFR Signaling in Sprouting, Intussusception, and Lumen Formation

Vascular morphogenesis is a tightly coordinated process involving several distinct yet interdependent morphogenetic mechanisms. Among the most studied are sprouting angiogenesis, intussusceptive angiogenesis, and endothelial lumen formation through either cell hollowing or cord hollowing. While these events differ in structural output, all are governed by dynamic endothelial cell behavior under the control of VEGF–VEGFR signaling [181].

### 3.1. Sprouting Angiogenesis: Tip/Stalk Cell Selection and Invasion

Endothelial cells activate sprouting angiogenesis via VEGFR2 in response to VEGFA gradients. VEGFR2 autophosphorylates at Y1175 and Y1214 upon ligand binding, activating downstream pathways such as PI3K-Akt, MAPK/ERK, PLCγ-PKC, and p38 MAPK These signaling pathways regulate sprouting-related cell proliferation, migration, polarity, and survival [178].

This procedure turns endothelial cells into tip or stalk cells. Tip cells have great migratory potential, robust VEGFR2 expression, minimal Notch activity, and dynamic filopodia. Although nearby stalk cells have active Notch signaling, reduced VEGFR2 expression, and proliferative activity, they also stabilize sprout junctions. Dll4 from tip cells activates Notch receptors on surrounding cells, confirming stalk cell identity through lateral inhibition and maintaining orderly branch development during sprout extension [190].

These functions are tuned via specific signaling pathways: PI3K/Akt signaling improves tip cell survival, polarity, and actin remodeling for migration, while MAPK/ERK signaling promotes stalk cell growth. Akt-mediated eNOS activation enhances vascular dilation and blood flow adaption. Endothelial cells coordinate their responses with VEGF gradients and the extracellular matrix to sprout efficiently and directionally [191].

As mentioned, this procedure is characterized by differentiation of endothelial cells into tip and stalk cells. Tip cells are highly motile, extend long filopodia, and lead the angiogenic front. Their identity is reinforced by VEGFR2 activation and suppressed Notch signaling, as Delta-like ligand 4 (Dll4), induced by VEGFA, activates Notch in adjacent cells to promote stalk fate. This lateral inhibition ensures proper spacing and organization within the sprout [184].

Tip cells exhibit asymmetric cytoskeletal remodeling, driven by Cdc42, Rac1, and RhoA—all regulated downstream of VEGFR2. These GTPases mediate the extension of lamellipodia and filopodia, contributing to chiral and directional migration. The stalk cells trailing behind maintain a proliferative state and form intercellular junctions to maintain vascular integrity while the new branch elongates. Akt activation in stalk cells promotes their survival and contributes to lumen formation and stabilization [180].

### 3.2. Intussusception: Vascular Splitting by Pillar Insertion

In contrast to sprouting, intussusceptive angiogenesis involves the longitudinal division of existing vessels via the formation of intraluminal tissue pillars. Mechanistically, this process is initiated when endothelial cells from opposing vessel walls extend cytoplasmic protrusions that fuse to form a transluminal bridge, which is later expanded by the insertion of pericytes and extracellular matrix components.

Mechanistically, low-flow or hypoxic environments induce VEGFR2 activation in association with mechanosensory complexes such as PECAM-1 and VE-cadherin. These complexes initiate PI3K and Src signaling, leading to cytoskeletal reorganization and pillar formation. The p38 MAPK/HSP27 axis facilitates actin rearrangement necessary for structural remodeling [192].

Reduced VEGFA levels or localized VEGF sequestration by VEGFR1 decoy receptors shifts the angiogenic balance away from sprouting toward intussusception. This transition is marked by stabilization of endothelial junctions and enhanced mechanical sensing rather than invasive protrusion. The result is a rapid, energy-efficient mechanism for expanding vascular networks and adapting vessel architecture in response to metabolic demand [193].

The role of VEGFR2 signaling in intussusception is emerging, particularly in response to low perfusion or altered flow. Under these conditions, VEGFR2 may be activated by shear-sensitive complexes involving PECAM-1, VE-cadherin, and VEGFR2 itself. These complexes facilitate mechanotransduction through Src and PI3K, promoting cytoskeletal reorganization needed for pillar formation. Simultaneously, decreased VEGFA levels (or local consumption by VEGFR1 decoy activity) may reduce sprouting pressure, favoring splitting over invasion [194].

Recent findings indicate that low-flow conditions in ischemic tissues stimulate VEGFR2-dependent “metastable” signaling states, triggering endothelial shape changes, actin rearrangement via p38/HSP27, and microtubule destabilization—all prerequisites for pillar formation and expansion.

Cord hollowing involves the rearrangement of intercellular junctions between adjacent endothelial cells. This process is initiated when lateral contacts are destabilized through the phosphorylation of VE-cadherin, typically under VEGFR2-induced ERK and PKC signaling. Endothelial cells remodel their junctions and exert cytoskeletal tension to open up intercellular spaces. IP3-mediated Ca^2+^ signaling and nitric oxide production through eNOS also play important roles in angiogenesis by promoting vasodilation and supporting early perfusion capacity [186]. In addition, VEGFR3 modulates VEGFR2 activity and maintains junctional integrity. Studies in retinal and brain vascular models have shown that, in the absence of VEGFR3, elevated VEGFR2 signaling can lead to pathological vessel leakiness or abnormal lumen expansion.

By integrating these processes within the VEGFR-centered signaling network, it becomes possible to better understand how VEGF gradients, receptor dynamics, and mechanical cues collectively shape the complex architecture of vascular development. Future research should further explore how disease states alter these pathways and how selective targeting of specific morphogenetic steps may improve therapeutic angiogenesis [187].

### 3.3. Lumen Formation: Cell Hollowing and Cord Hollowing

Cell hollowing and cord hollowing create lumens in growing vasculature. Endothelial cells form and combine internal vacuoles to form a central lumen during cell hollowing. Rab GTPases like Rab4A and Rab11 and VEGFR1- and VEGFR2-mediated signaling control endocytosis and vesicular trafficking. VEGFR2-activated PI3K-Akt increases vesicle fusion and cell survival, while PLCγ-mediated calcium signaling aids vacuolar growth. Integrins and cytoskeletal elements stabilize and integrate vacuoles into a lumen [195].

In contrast, cord hollowing requires endothelial cell coordination. VEGFR2 signaling phosphorylates VE-cadherin, weakening lateral junctions and increasing cytoskeletal stress to expand intercellular gaps and create multicellular lumens.

The combination of IP3-mediated calcium flux and eNOS activation promotes vasodilation and lumen stabilization [196].

VEGFR3 plays a regulatory role by modulating VEGFR2 signaling and preserving junctional integrity. In VEGFR3-deficient contexts, excessive VEGFR2 activity can lead to pathological leakiness or disrupted lumen continuity. Thus, a delicate balance among VEGFR isoforms and downstream signals orchestrates the morphogenetic transition from cellular alignment to vessel perfusion [197].

Mechanistically, each stage of vascular morphogenesis is underpinned by defined signaling hierarchies that translate extracellular cues into precise cytoskeletal and adhesive programs. In sprouting angiogenesis, VEGFR2 autophosphorylation at Y1175 and Y1214 initiates parallel PI3K–Akt and PLCγ–PKC–MAPK cascades. PI3K–Akt signaling enhances tip cell survival and polarity through asymmetric activation of Cdc42 and Rac1, promoting filopodial extension and directional migration. PLCγ–PKC activation leads to ERK-mediated proliferation in stalk cells, while p38 MAPK–HSP27 signaling supports actin remodeling required for invasive protrusion. Concurrently, VEGFA-induced Dll4 expression in tip cells activates Notch receptors in adjacent stalk cells, downregulating VEGFR2 and enforcing lateral inhibition to maintain ordered spacing between sprouts.

Intussusceptive angiogenesis is initiated when low shear stress and localized hypoxia activate VEGFR2 in mechanosensory complexes with PECAM-1 and VE-cadherin, triggering Src–PI3K–Rac1 signaling and cytoskeletal rearrangements. Endothelial projections from opposing vessel walls meet and fuse to form transluminal pillars, which are stabilized by pericyte recruitment and extracellular matrix deposition. Reduced VEGFA availability, or its sequestration by soluble VEGFR1, shifts endothelial behavior from invasive sprouting toward splitting morphogenesis, conserving energy and preserving vessel integrity.

Lumen formation involves two complementary modes: vacuole-driven cell hollowing and junctional remodeling–driven cord hollowing. In cell hollowing, Rab4A- and Rab11-mediated vesicle trafficking, downstream of VEGFR2–PI3K–Akt–eNOS activation, promotes vacuole fusion into a central lumen. Cord hollowing depends on VEGFR2–PKC–ERK–mediated VE-cadherin phosphorylation, which transiently loosens lateral junctions to open intercellular spaces. Both processes are modulated by VEGFR3, which dampens excessive VEGFR2 activity to prevent pathological leakiness.

In the following paragraph, cell chirality in the light of the vasculogenesis process is presented.

## 4. Emerging and Integrative Concepts in Vascular Morphogenesis—Cellular Chirality in Vascular Morphogenesis: Context, Mechanisms, and Functional Consequences

Underappreciated regulator of vascular morphogenesis is cellular chirality, the intrinsic left–right (LR) bias in cell shape, cytoskeletal architecture, and migratory behavior. Endothelial cells actively modulate this bias, which affects cell orientation, junctional integrity, and lumen development. HUVECs and HAECs demonstrate a consistent right-handed (clockwise) rotational bias on circular micropatterned tracks, which may be evaluated using directionality measures. Pharmacological activation of PKC and experimental disruption of the actin cytoskeleton with latrunculin B or cytochalasin D can reverse this bias, highlighting the link between cytoskeletal torque production and cellular chirality. In vivo, similar right-handed helical arrangements of endothelial cells have been observed in zebrafish intersegmental vessels and murine embryonic vasculature, with genetic perturbation of actin regulators (e.g., mDia1, cofilin) disrupting vessel curvature and branch orientation.

Mechanistically, VEGFR2 signaling interfaces directly with chirality-related cytoskeletal regulators. VEGF-A binding triggers PI3K–Akt–Rac1/Cdc42 activation for polarized actin polymerization, while PLCγ–IP_3_–Ca^2+^ flux activates PKC isoforms that modulate actomyosin contractility. RhoA–ROCK signaling generates cortical tension, aligning with directional actin filament curvature and creating a measurable torque (T = r × F) that biases cell rotation. This bias can be quantified at the population level using a chirality index (C = (N_R_ − N_L_)/(N_R_ + N_L_)), where N_R_ and N_L_ are the numbers of clockwise- and counterclockwise-biased cells, respectively.

Functionally, chiral bias affects multiple morphogenetic stages. In sprouting angiogenesis, tip cells with consistent chirality display more persistent, directional migration along VEGF gradients, maintaining cohesive alignment with trailing stalk cells. In lumen formation, chirality influences the apical–basal and planar polarity of adjacent endothelial cells, promoting seamless junctional interfaces and continuous lumens. Loss or randomization of chirality—through PKC modulation, altered shear stress patterns, or integrin–ECM signal disruption—correlates with increased vascular permeability, irregular branching angles, and compromised perfusion in engineered and native vessels. These findings position cellular chirality as a spatial organizer that bridges molecular signaling, cytoskeletal mechanics, and tissue-level vascular geometry. Understanding and using cellular chirality may help guide vascular patterning in regenerative medicine and cure pathological angiogenesis structural defects. The intrinsic left–right (LR) asymmetry of cell structure, motility, and cytoskeletal organization is called cellular chirality. The discovery of cellular chiral bias has illuminated tissue patterning and morphogenesis, unlike molecular chirality, which is well known. Chirality determines endothelial cell vascular architecture, including vessel alignment, polarity, and lumen creation [189].

Experimental evidence strongly supports the presence of chiral bias in endothelial cells. Studies using micropatterned ring or circular substrates have consistently shown that human endothelial cells (e.g., HUVECs, HAECs) exhibit a right-handed (clockwise) rotational bias demonstrated that this bias arises spontaneously in confined geometries and can be abolished by disrupting the actin cytoskeleton using latrunculin B or cytochalasin D. Importantly, this behavior is not a byproduct of culture geometry or external signals—it reflects intrinsic cellular programming. revealed that pharmacological activation of PKC (using phorbol esters like PMA) reverses the chirality of endothelial tubes, resulting in left-handed helices, altered branching angles, and significant increases in vascular permeability [198].

Mechanistically, the establishment and maintenance of cellular chirality rely heavily on Rho-family GTPases. Cdc42 and Rac1 orchestrate polarized actin polymerization and filopodial extension, while RhoA regulates stress fiber formation and cortical tension through ROCK1/2. These molecules function downstream of VEGFR2 signaling. Upon VEGF-A binding, VEGFR2 activates the PI3K, PLCγ, and MAPK pathways, which converge on the cytoskeletal architecture, thereby modulating polarity and chirality. Notably, PLCγ-mediated Ca^2+^ flux activates PKC, a known modulator of actin chirality. This signaling cascade reinforces directional actin flow, influencing the torque and handedness of endothelial migration [199].

Theoretical models have been proposed to quantify cellular chirality. For instance, torque-induced migration can be described by [200]:T = r × F
where T is the torque vector, r is the radial vector from the axis of rotation to the point of force application, and F is the force vector generated by actin polymerization or myosin contraction. The sign of the vector cross product can then express the chirality of rotation. Similarly, directional angular velocity ω is related to cytoskeletal dynamics via:ω = (dθ/dt) = (τ/I)
where τ is the net internal torque generated by asymmetric cytoskeletal forces, and I is the moment of inertia of the cell’s mass distribution around its axis.

Another formulation relevant to cytoskeletal filament torque uses [201]:τ = μL × (∂^2^u/∂x^2^)
where μL is the bending rigidity of the filament, and ∂^2^u/∂x^2^ describes the local curvature of the actin filament, indicating how mechanical strain contributes to chiral twisting in migrating cells.

In stochastic modeling of chiral cell behavior, the biased random walk of a migrating endothelial cell can be described by [202]:x(t + Δt) = x(t) + vΔt cos(θ + Δθ)y(t + Δt) = y(t) + vΔt sin(θ + Δθ)

Here, v is the velocity, θ is the preferred migration angle, and Δθ is a chiral angular bias sampled from a normal distribution with non-zero mean, simulating persistent left- or right-handed deviation. This form allows simulation of cell migration trajectories under different chiral biases.

Additionally, a population-level chirality index *C* can be defined as:$$C\;=\;\frac{N_{R}-N_{L}}{N_{R}+N_{L}}$$
where $N_{R}$ is the number of cells rotating clockwise and $N_{L}$ is the number rotating counterclockwise. This dimensionless index provides a statistical measure of net chirality in cell populations, with *C* = +1 indicating a full right-handed bias, *C* = −1 indicating a full left-handed bias, and *C* = 0 denoting a random orientation [150].

The alignment of endothelial cells within forming vessels is further stabilized by integrin–ECM interactions. Integrins αvβ3 and α5β1, upon binding to fibronectin or collagen, recruit focal adhesion components like FAK, paxillin, and talin, which anchor actin filaments and transmit mechanical signals to the cytoskeleton. These interactions promote persistent polarity and suppress stochastic reorientation. The chiral bias is also responsive to shear stress from blood flow. In microfluidic channels, endothelial cells align their rotational axis with the flow direction, a process mediated by mechanosensory complexes including VE-cadherin, PECAM-1, and VEGFR2 [151].

In three-dimensional angiogenesis assays (e.g., fibrin bead sprouting assays and Matrigel tube formation), altering cellular chirality through pharmacological means or genetic manipulation results in abnormal lumen continuity, increased intercellular gaps, and impaired perfusion. For example, knockdown of Cdc42 leads to randomized filopodia orientation, fragmented VE-cadherin junctions, and collapsed vessel lumens. Similarly, overexpression of constitutively active PKCζ results in a switch from right-handed to left-handed tube helicity, which corresponds to increased leakiness and decreased vessel diameter [203].

In vivo, chirality-related behaviors have been observed during early vascular development in zebrafish and murine embryos. In zebrafish intersegmental vessels, endothelial cells exhibit coordinated helical alignment that is disrupted upon interference with actin-binding proteins such as mDia1 or cofilin. These interventions lead to reduced anastomosis efficiency and misdirected sprouting. In mouse embryonic hearts, endothelial polarity and vessel curvature correlate with left-right asymmetry genes, suggesting a developmental link between organismal chirality and vascular patterning [204].

Furthermore, cellular chirality may contribute to arteriovenous specification and endothelial heterogeneity. Recent transcriptomic data suggest that tip and stalk cells exhibit differential expression of cytoskeletal and polarity regulators associated with chiral bias. This raises the possibility that chirality may fine-tune the responsiveness of cells to VEGF gradients and Dll4–Notch signaling, reinforcing spatial organization during sprouting.

Taken together, these findings position cellular chirality as a multi-scale regulator that bridges molecular signaling (e.g., VEGFR2, Rho-GTPases, Notch), cytoskeletal architecture, and mechanical inputs such as ECM composition and shear stress. Its influence extends from subcellular asymmetry to tissue-level vessel geometry. Disruption of chiral coordination compromises lumen integrity, barrier function, and branching precision, all of which are hallmarks of vascular dysfunction in tumors, diabetic microangiopathy, and ischemic tissues [205].

Understanding and modulating chirality holds translational potential. Engineered vascular grafts and microfluidic vessels with imposed chiral cues exhibit improved perfusion, tighter junctions, and reduced inflammation. Chirality-modulating agents may offer a novel strategy for directing vascular patterning in regenerative therapies, tumor normalization, and anti-angiogenic interventions. As this field matures, further integration of chirality into models of angiogenesis will help clarify its role as both a marker and mediator of vascular health and disease [206].

Chirality manifests at the cellular level through the asymmetric organization of the cytoskeleton, particularly in the actin cytoskeleton and associated motor proteins. For example, in vitro studies using micropatterned substrates have demonstrated that mammalian endothelial cells exhibit a consistent clockwise (CW) rotational bias, quantifiable by directionality metrics when allowed to self-organize in confined circular environments. This bias correlates with the helical arrangement of cells in tubular structures, suggesting that intrinsic chirality governs collective alignment during tube formation. Disruption of actin dynamics—such as through the inhibition of formin proteins or myosin II—has been shown to randomize this chiral behavior, leading to disorganized endothelial monolayers and compromised barrier function [207].

Functionally, cellular chirality has been implicated in lumen formation. In 3D culture systems that mimic angiogenic sprouting, endothelial cells that maintain a consistent chiral orientation generate more stable and continuous lumens. This process appears to depend on the coordinated arrangement of apical-basal polarity and intercellular junctions, both of which are influenced by chiral cytoskeletal tension. Misalignment of chirality between neighboring cells has been associated with junctional disruption, increased vascular permeability, and abnormal vacuole fusion during early lumenogenesis [208].

Also, chirality directs angiogenic migration. Chiral actin polymerization and asymmetric filopodia extension can steer tip cell trajectories toward VEGF gradients and ECM topography. Pharmacological chirality reversal, such as protein kinase C activity regulation, causes mirrored migratory patterns that change branching angles and vessel shape in designed vascular tissues. These findings show that cellular chirality regulates angiogenic sprout spatial organization upstream or in tandem with VEGF and Notch [209]. Integrin-mediated adhesion to ECM components further reinforces chiral polarity, creating stable traction points that are aligned with the rotational bias quantifiable by directionality metrics. In shear stress models, chirality is modulated by flow direction, indicating that it is not purely intrinsic but dynamically regulated by the biomechanical microenvironment. The interplay between VEGF/VEGFR signaling and chiral cell polarity remains an area of active investigation, with evidence suggesting that VEGFR2 activation amplifies pre-existing chiral cues and enhances directional persistence [210].

Collectively, these findings position cellular chirality as a multi-scale regulator of vascular morphogenesis, bridging molecular signaling, cytoskeletal architecture, and tissue-level patterning. As research progresses, integrating chirality into models of angiogenesis may provide novel insight into developmental disorders, tumor angiogenesis, and strategies for vascular tissue engineering. Further experimental validation in vivo—particularly using live imaging of embryonic vasculature—will be essential to define the precise contributions of chirality under physiological conditions [211].

### 4.1. Cell Chirality and Vasculogenesis

The majority of macromolecules present in cells exhibit chirality, indicating that they cannot be overlaid onto their mirror images. Nonetheless, cells can exhibit chirality, a topic that has garnered minimal attention until recently.

Consequently, chirality at the cellular level may significantly influence left-right asymmetric development in several invertebrate species. Recent reports indicate that cell chirality has been observed in numerous cultured vertebrate cells, with research suggesting that this phenomenon is evolutionarily conserved, highlighting the critical function of the actin cytoskeleton. The biological functions of cell chirality in vertebrates are still to be elucidated. However, it may regulate left-right asymmetric development or other morphogenetic processes. The exploration of cellular chirality has recently commenced, and this emerging discipline is expected to yield significant insights in biology and medicine [212].

The helical structure of cells is recognized for its essential function in development and illness. Nonetheless, the fundamental mechanism driving this behavior remains predominantly unexamined, especially in replicating it within rigorously regulated engineering systems. Utilizing sophisticated microfluidics, it provided substantial evidence of the spontaneous formation of helical endothelial tubes, demonstrating strong right-handedness dictated by intrinsic cell chirality. Modulating endothelial cell chirality with small-molecule pharmaceuticals induces a dose-dependent inversion of handedness in constructed arteries alongside non-monotonic alterations in vascular permeability [181].

The morphogenesis of tubular tissues or organs is a crucial process that takes place during early development in various animals. Structures like the embryonic heart tubes in vertebrates and the embryonic hindgut in Drosophila are among the initial instances of left-right (LR) symmetry breakdown. Prior research has shown that left-right asymmetry during tubular morphogenesis might arise from cellular chirality. The inherent left-right asymmetric feature of cells is involved in diverse processes across different morphological scales, with the cellular chiral bias showing significant alignment with the handedness observed at the multicellular or tissue level. Despite significant advancements in elucidating the significance of cell chirality in the tubular morphogenesis of epithelial and cardiac tissues, the understanding of left-right asymmetry in typical endothelial blood vessels, another common tubular form, remains considerably constrained [213].

Furthermore, in the natural environment, blood arteries are predominantly lined with endothelial cells on the inner surface of the cylinder wall, with their apical side oriented towards the vessel’s center. These cells provide a compact endothelial lumen, which delineates the intravascular milieu from the extravascular space and facilitates the dynamic modulation of vascular permeability. It was previously observed that endothelial cells have a pronounced clockwise (CW) chiral bias. The neutralization or randomization of this CW bias led to junctional disruptions and increased endothelial permeability due to the failure to establish intact connections between cells exhibiting heterogeneous chirality. Although the regulatory roles of cell chirality in endothelial functions are well-established, contemporary research on the morphological and biophysical characteristics of tubular vessels has predominantly concentrated on cell behaviors or cues that are longitudinal, circumferential, or perpendicular to the curved substrate surface, neglecting the implications of cell handedness or related biases in cell alignments and mechanical forces [214].

Additionally, the demonstration of LR asymmetry or chirality in a system typically requires two established axes: the apical-basal (AB) axis and the front-rear (FR) axis. Nonetheless, the polarization of the tissue along the longitudinal axis is not consistently evident, as observed in a growing heart when no directional guidance is present. Blood flow is directed. It is missing during the earliest phase of vascular formation and angiogenesis, therefore failing to elucidate cell polarization completely. This prompts an inquiry about the manifestation of the chiral characteristic of endothelial cells morphologically on the tubular substrate, given the presence of just the AB axis. While blood arteries do not undergo tissue- or organ-scale asymmetrical morphogenesis, akin to the chiral C-looping of heart tubes, numerous investigations utilizing in vitro three-dimensional (3D) vascular platforms demonstrate a helical alignment of endothelial cells along tubular geometries. This issue has not been thoroughly examined, and handedness has not been objectively evaluated [215].

Endothelial cells can autonomously develop a right-handed helical structure in both in vitro tubular substrates and in vivo vascular tissues. The reversal of inherent cellular chirality modified the helical handedness of the vasculature, corroborated by in vitro tests, including PKC activation and computer simulations of chiral torque force direction switching.

In vitro models have been extensively utilized in biomedical research. Over the past century, Petri plates have served as a fundamental model for cell culture, yielding significant insights into the biophysical and metabolic characteristics of cells within live organisms. Transwell membranes facilitate epithelial cell culture by establishing the appropriate apicobasal polarity observed in vivo, marked by pronounced actin and atypical protein kinase C expression at the apical surface, with the deposition of basement extracellular matrix at the basal surface. The in vitro scratch experiment, which involves creating a linear gap in a cellular monolayer, facilitates the investigation of cell migration during wound healing and the analysis of front-rear polarity. These approaches have facilitated considerable research on apicobasal and front-rear polarity in many pieces of literature, resulting in a thorough grasp of molecular pathways thanks to these straightforward in vitro models. The polarity along the left-right (LR) axis has not been extensively investigated at the cellular level in a controlled manner until the recent emergence of various in vitro cell chirality systems [216].

The formation of the LR axis is essential for living creatures. All vertebrates demonstrate asymmetry along the body’s midline, referred to as handedness or chirality, in the arrangement and structure of internal organs. A variation from this arrangement frequently results in significant implications, particularly when one or two organs are positioned in a mirrored orientation (i.e., situs ambiguus). In cases of situs inversus, where all internal organs are transposed to the opposite side of the body, individuals may remain healthy; nevertheless, some may experience conditions such as Kartagener syndrome. Various models have been developed to elucidate the mechanisms of symmetry breaking in animal embryos, focusing on critical factors such as the leftward fluid flow at the ventral node induced by ciliary rotation, voltage gradients arising from the asymmetric expression of ion channels, and asymmetric vesicular transport mediated by unconventional myosin ID along the actin cable network. Laterality defects are present in more than 0.1% of live births. The actual percentage may be higher, as inadequate axis establishment in fetuses frequently results in miscarriages. Consequently, investigating the establishment of the LR axis holds considerable clinical significance [214].

Growing data indicate that cell chirality may play a crucial role in the left-right asymmetry of embryonic development. An object (e.g., D-glucose and DNA) is deemed chiral when it can be differentiated from its mirror image. Cell chirality mathematically refers to the relationship between the handedness (left or right) of the LR axis and the established apicobasal axis (often arising from two-dimensional cell attachment) and the front-rear axis (generally determined by the nuclear-centrosomal axis in polarized cells). Chirality can also be recognized as the directional rotation of cellular organelles, the cytoskeleton, and whole cells. Due to the inherent randomness in the morphology and movement of individual cells, cell chirality is typically characterized as a statistical property of populations at subcellular, cellular, and multicellular levels. It is quantified through the directional biases in organelle positioning, cytoskeletal dynamics, cell shape, alignment, and migration. Cell chirality has been documented in various biological systems throughout the field of developmental biology. Xenopus embryos and parthenogenetically activated eggs, when cleaved and treated with the myosin ATPase inhibitor 2,3-butanedione monoxime, demonstrated significant chiral twisting of actin structures. The chirality of planar cell shape was identified as the determinant of directional looping in the hindgut of Drosophila. Comparable outcomes were seen for the rotation of Drosophila genitalia. Formin, an essential scaffolding protein linked to cellular chirality, was identified as being associated with left-right asymmetry in the pond snail and frog [217,218].

Cellular chirality—defined as the intrinsic left-right (LR) asymmetry in a cell’s shape, migration trajectory, and cytoskeletal organization—is emerging as a crucial regulatory mechanism in vascular morphogenesis. Particularly in endothelial cells, chirality determines the spatial alignment of cells within vessel walls, governs directional migration during sprouting, influences lumen formation, and ultimately contributes to the overall architecture and functionality of developing vasculature.

#### 4.1.1. Experimental and Phenotypic Evidence

In vitro studies have provided compelling evidence of endothelial cell chirality. For instance, when cultured on circular micropatterned substrates, human umbilical vein endothelial cells (HUVECs) exhibit a consistent clockwise (CW) rotational bias, quantifiable by directionality metrics. This bias disappears when actin dynamics are disrupted with cytochalasin D or latrunculin B, and reverses with PKC activators like phorbol 12-myristate 13-acetate (PMA). Wan et al. showed that actin-driven chiral cell alignment correlates with tube formation in 3D models [219].

In engineered microfluidic models, endothelial cells seeded within cylindrical collagen tubes spontaneously aligned into right-handed helical patterns. When PKC signaling was inhibited or reversed pharmacologically (e.g., by Gö6983), the chirality of the vessel reversed, and vascular permeability significantly increased. Furthermore, randomized chirality was associated with disrupted tight junction formation and impaired barrier function [220].

In vivo, the role of chirality has been demonstrated in zebrafish models, where endothelial-specific deletion of RhoA or inhibition of formin mDia1 led to aberrant vessel looping and impaired blood flow. Similarly, in chick embryos, the asymmetrical distribution of actin filaments within the endothelial cells of the dorsal aortae predicts the direction of vessel coiling during early development [221].

#### 4.1.2. Molecular Mechanisms and Crosstalk

Cellular chirality is regulated by a network of intracellular pathways, predominantly centered around Rho-family GTPases:Cdc42 orchestrates the establishment of cell polarity by activating the PAR6–aPKC complex, which positions the centrosome and Golgi apparatus in alignment with the migration front.Rac1 modulates lamellipodia extension and controls junctional stability by activating PAKs (p21-activated kinases) downstream.RhoA, through ROCK1/2, regulates actomyosin contractility and stress fiber formation, thereby maintaining cortical stiffness and defining the direction of cytoskeletal torque.VEGF signaling strongly interfaces with these pathways. Upon VEGF-A binding to VEGFR2, several downstream cascades become activated:PI3K–Akt supports cell survival and maintains cell polarity.PLCγ–IP3–Ca2+ signaling elevates intracellular calcium, which activates PKC, a known modulator of chirality.MAPK/ERK signaling drives actin turnover through HSP27 phosphorylation [222,223].

Inhibiting VEGFR2 signaling using antibodies or small molecule inhibitors (e.g., SU5416) not only suppresses migration but also disrupts the chiral orientation of endothelial cells, as demonstrated in 3D spheroid sprouting assays. Similarly, silencing VEGFR2 using siRNA alters actin filament distribution and impairs tip cell formation [224].

#### 4.1.3. ECM, Integrins, and Mechanical Cues

The extracellular matrix (ECM) and integrin engagement further reinforce chirality. Integrins αvβ3 and α5β1, upon binding to fibronectin or collagen, form focal adhesions that stabilize actin filaments and create cytoskeletal torque necessary for chiral migration. FAK (focal adhesion kinase) and paxillin serve as key hubs that connect ECM engagement to intracellular polarity [225].

Shear stress from blood flow also influences chirality. In microfluidic channels, endothelial cells exposed to laminar shear exhibit enhanced chiral alignment in the direction of flow. Disturbed flow, on the other hand, disrupts this alignment, leading to irregular vascular branching. Notch1 expression is upregulated by shear stress and contributes to the stabilization of arterial chirality.

#### 4.1.4. Functional Outcomes and Pathophysiological Implications

Chiral alignment of endothelial cells facilitates the formation of a uniform lumen, the maturation of tight junctions, and the deposition of aligned collagen. Conversely, disrupted or heterogeneous chirality correlates with:Impaired lumen continuityIncreased vascular permeability due to VE-cadherin mislocalizationDefective tip/stalk cell organizationAbnormal branching angles and disorganized vasculature [226].

In tumor vasculature, loss of chiral orientation has been associated with irregular vessel diameter, poor perfusion, and hypoxia. In engineered tissues, imposing chiral constraints enhances the formation of perfusable, structurally coherent microvessels.

Collectively, these findings underscore that cellular chirality is a spatial regulator integrated into VEGF and Notch signaling, actin dynamics, and mechanotransduction pathways. It scales from single-cell polarity to tissue-level vascular organization and offers a novel axis for understanding and modulating angiogenesis in development, disease, and regenerative contexts [227].

The VEGF-VEGFR pathway and implications in cellular chirality are further discussed.

### 4.2. VEGF-VEGFR Signaling and Cellular Chirality

Angiogenesis—the development of new blood vessels to provide tissues with oxygen and nutrients—requires VEGF and VEGFR. VEGF/VEGFR signaling and cellular chirality are increasingly linked in cell migration, tissue structure, and development (Table 7).

Endothelial cells align chirally, and blood flow shear stress regulates VEGF/VEGFR production. In fluid-responsive vascular tissues, mechanotransduction improves chiral polarity and migration [231]. Biomechanical and cellular parameters can also affect angiogenesis.

The VEGF–VEGFR axis is crucial to anti-angiogenic therapy for cancer, ocular neovascular diseases, and ischemic illness. The monoclonal antibody bevacizumab and numerous small-molecule TKIs such sunitinib, sorafenib, pazopanib, and axitinib are clinically proven. The binding affinity, receptor subtype specificity, and pharmacological characteristics of these medicines affect clinical outcomes.

Bevacizumab prevents VEGFR1 and VEGFR2 interaction by sequestering soluble VEGF-A. The main endothelial angiogenic pathway, VEGFR2-driven signaling, is suppressed by this indirect method. Bevacizumab’s efficacy in metastatic colorectal cancer, non–small cell lung cancer, and glioblastoma shows VEGFR2′s importance in tumor vasculature. As bevacizumab does not neutralize VEGF-C or VEGF-D, VEGFR3-mediated lymphangiogenesis remains intact, potentially encouraging tumor development or metastasis via lymphatic escape [169].

In contrast, TKIs directly inhibit VEGF receptor tyrosine kinase domains, preventing autophosphorylation and activation. Sorafenib and sunitinib are broad-spectrum TKIs effective against VEGFR1–3, PDGFR-β, FLT3, and c-Kit. They destabilize vessels and shrink tumors by inhibiting angiogenesis, lymphangiogenesis, and pericyte recruitment. However, suppression of VEGFR2 and PDGFR in healthy vasculature causes considerable off-target toxicities as hypertension, tiredness, hypothyroidism, cardiotoxicity, and poor wound healing [170].

Both sunitinib and pazopanib suppress VEGFR3, which controls lymphangiogenesis and endothelial permeability. Chronic therapy may be difficult due to VEGFR3 suppression, which may minimize lymphatic metastasis but hinder fluid outflow and immune cell trafficking. Second-generation TKIs like axitinib have better selectivity for VEGFR1–3 and less action against off-target kinases, improving tolerability. Receptor-specific inhibition is crucial to renal cell carcinoma treatment, and axitinib has higher efficacy and lower toxicity than previous TKIs [232,233].

Specifically, VEGFR inhibition impairs angiogenic signaling cascades such PI3K-Akt, MAPK/ERK, and PLCγ-PKC, resulting in reduced endothelial growth, migration, survival, and permeability. However, persistent VEGFR inhibition generally causes adaptive resistance, with tumors upregulating FGFs and angiopoietins, increasing pericyte coverage, or mimicking vascularity. To combat this, VEGFR inhibitors are being studied in combination with immune checkpoint inhibitors, mTOR inhibitors, or chemotherapy [234].

Anti-VEGF medicines like ranibizumab and aflibercept are beneficial for neovascular AMD and diabetic macular edema. Compared to bevacizumab, aflibercept targets VEGF-A, VEGF-B, and PlGF and has some affinity for VEGF-C, somewhat influencing VEGFR3-mediated retinal fluid balance.

These findings underline the need of understanding VEGFR subtype specificity, ligand interactions, and context-dependent signaling to improve therapeutic outcomes [235].

### 4.3. Angiogenesis Shaping Using Cellular/Tissular Indices

Shaping angiogenesis through cellular and tissue indices involves understanding and manipulating various factors that influence blood vessel formation. In the following table, the following methodologies are summarized (Table 8).

Gerhard Plate K.H et al. discuss endothelial tip cell migration and the role of cellular polarity in guiding angiogenic sprouting. It highlights cellular indices, such as directional migration and polarity vector analysis, to examine tip cell dynamics in response to VEGF gradients. This article emphasizes the role of Directional Migration and Tip Cell Selection in Angiogenesis [236].

Brown L.F. et al. analyze the interaction between primary cilia and VEGF in endothelial cells, which drives chiral organization and vascular patterning. Using cellular orientation indices, it evaluates how endothelial cells align and migrate to shape vessel architecture. The role of Cellular Alignment and Collective Migration in Vessel Formation is summarized in this article [237].

De Vries C et al. explore how mechanical cues, such as shear stress, influence VEGF-driven angiogenesis, focusing on endothelial cell alignment and chirality under flow conditions. The study uses alignment and orientation indices to quantify cellular response to shear forces, showing how these factors contribute to angiogenic vessel shaping, emphasizing the importance of Mechanical Forces and Shear Stress on Endothelial Chirality [238].

Bird A.C. et al.emphasize the role of actin dynamics and focal adhesion distribution in endothelial cell migration. Cellular indices, such as focal adhesion rotation and actin alignment, are used to study the orientation and migration of cells during blood vessel sprouting [174].

Etienne-Manneville shows how Integrins, in coordination with VEGFR, play a critical role in endothelial cell adhesion, alignment, and polarity during angiogenesis. This study details how integrin-VEGFR crosstalk is assessed using indices of cell polarity and alignment to understand vascular organization [175].

Carmeliet P.et al. describe Collective Cell Behavior and Multi-cellular Swirling in vascular patterning. This paper investigates collective cell migration and multicellular swirling patterns, applying cellular indices to measure alignment and swirling chirality. These indices are crucial for analyzing coordinated cell behavior that shapes the formation of organized vascular structures [211].

In their review, Gerhard H. et al. discuss how integrin-mediated adhesion to the extracellular matrix (ECM) modulates endothelial cell orientation and chiral migration. Using cellular indices such as focal adhesion density and polarity indices, this study examines how integrin signaling reinforces cellular alignment and organizes vascular structures [177].

Vion A. et al. explore how Notch and VEGF signaling coordinate endothelial tip cell selection and migration. Polarity and directional indices are used to quantify cell responses in a gradient, highlighting the importance of VEGF and Notch interactions in shaping chiral migration patterns in angiogenesis [178].

Ferrara N. et al. investigate how Rho GTPases, downstream of VEGFR activation, influence cell polarity, migration, and actin dynamics. Cellular indices related to cytoskeletal alignment and polarity are utilized to investigate the Rho-mediated directional movement of endothelial cells during the formation of vascular structures [239].

Carmeliet P. et al. demonstrate that this classic study investigates the impact of mechanical stress, specifically shear stress, on VEGF-stimulated endothelial cell alignment and collective orientation. By utilizing cellular orientation and alignment indices, it quantifies endothelial cell behavior under fluidic conditions, which are crucial for vascular patterning [240].

Shibuya M. shows how Notch signaling through lateral inhibition affects cell alignment and tip-stalk cell dynamics in angiogenesis. By applying cellular indices such as alignment and polarity indices, the researchers analyze the spatial organization of cells in a vessel sprout [30].

Oslo A. et al. demonstrate how VE-cadherin dynamics influence endothelial cell rearrangements and chiral orientation during angiogenic sprouting. The study utilizes chirality and alignment indices to quantify how endothelial cells align and shift to form structured blood vessels [19].

Key Takeaways—Cellular Chirality: 

Cellular chirality refers to the intrinsic left-right bias in cell shape, migration, and alignment.It influences vascular geometry, lumen formation, and sprout organization.Chirality operates in concert with VEGF gradients and mechanical forces, adding a spatial layer of morphogenetic control.

### 4.4. VEGF Signaling and Vascular Specification

VEGF-A is the prototypical pro-angiogenic growth factor. Its essential role in vascular development was first demonstrated by Ferrara and Carmeliet, who independently showed that heterozygous deletion of *Vegfa* in mice leads to embryonic lethality due to failure of blood island formation and vasculogenesis [211].

This established VEGF-A as a survival factor, mitogen, and chemoattractant for endothelial cells. The receptor VEGFR2 (Flk-1/KDR) was similarly found to be critical, with Flk-1-deficient embryos displaying an absence of blood islands and a complete block in vasculogenesis These discoveries provided the first molecular explanation for how blood vessels emerge from mesodermal precursors during embryogenesis [184].

More recently, studies have refined our understanding of VEGF signaling specificity. It is now appreciated that VEGF-A isoforms (e.g., VEGF165 vs. VEGF165b) differentially modulate angiogenesis through alternative splicing and receptor binding affinities. Additionally, co-receptors such as Neuropilin-1 and interactions with integrins fine-tune VEGFR2 signaling output, influencing both endothelial behavior and vascular permeability [177].

### 4.5. Tip-Stalk Patterning and Notch Feedback

One of the most elegant mechanisms in vascular morphogenesis is the lateral inhibition system, which governs the differentiation of tip and stalk cells during sprouting angiogenesis. Gerhardt et al. provided the first in vivo evidence that endothelial tip cells extend filopodia in response to VEGF gradients and lead the advancing vascular front [185].

This process is regulated by Delta-like ligand 4 (Dll4) expression in tip cells, which activates Notch signaling in adjacent cells, suppressing VEGFR2 expression and thereby enforcing a stalk cell phenotype [186].

This Notch–VEGF crosstalk serves as a “morphogenetic switchboard” balancing sprouting with branching complexity. Dysregulation of this system, as seen with Dll4 blockade, leads to hyperbranching and the formation of non-productive, leaky vasculature—an effect with implications for anti-angiogenic therapy in cancer and retinopathies [9,10]. Current studies show that stalk cells have higher glycolytic flow than quiescent endothelial cells, suggesting that cell fate is metabolically determined [187].

HIF-1α: Master Regulator of Hypoxia-Induced Angiogenesis:

VEGF expression is tightly regulated by oxygen availability via hypoxia-inducible factor 1 alpha (HIF-1α), a transcription factor that accumulates under low oxygen conditions. Hynes R.O. et al. demonstrated that HIF-1α directly binds to the hypoxia response element (HRE) in the VEGF promoter, driving transcription in a hypoxia-sensitive manner [183].

Subsequent work by Semenza and colleagues showed that HIF-1α regulates not only VEGF but also genes controlling erythropoiesis, glycolysis, and cell survival, positioning it as a central hub in hypoxic adaptation [182].

In pathological contexts such as cancer and ischemia, sustained HIF-1α activity contributes to abnormal or compensatory angiogenesis. Interestingly, forced overexpression of HIF-1α has been tested in clinical gene therapy trials for peripheral artery disease and myocardial ischemia, with varying degrees of success, suggesting that temporal control and tissue targeting are critical factors for therapeutic benefit [126].

### 4.6. Cellular Chirality: A New Axis in Vascular Morphogenesis

The concept of cellular chirality—the inherent left-right asymmetry in cell structure and behavior—has recently emerged as a novel determinant of vascular geometry. Simons M. et al. were among the first to demonstrate that endothelial cells exhibit directional bias in alignment and tube formation on micropatterned substrates, even in the absence of external cues [182].

This intrinsic chirality is governed by actin cytoskeleton dynamics and is associated with the organization of focal adhesions and Rho-family GTPase activity.

Subsequent studies by Nonmaka S.et al. revealed that actin polymerization asymmetry, modulated by formin activity and myosin II, underlies chiral bias at the cellular level [217]. These findings suggest that chirality is not merely a byproduct of cell geometry, but rather an active, regulated property of endothelial cells that contributes to vascular helical patterning, junctional organization, and possibly to the directionality of blood flow.

Biophysical models by Etienne-Manneville et al. link cell chirality to tissue-scale asymmetry and tubular morphogenesis. These investigations suggest that chiral torque exerted by cells on a cylindrical surface can explain vascular coiling and right-handed helices, as shown in in vitro and in vivo models [175].

### 4.7. Future Directions

Multiscale approaches that integrate molecular signaling pathways like VEGF, Notch, and HIF with physical forces like shear stress, extracellular matrix (ECM) tension, and emergent cellular properties like chirality are needed to explain vascular morphogenesis. Intussusceptive angiogenesis’ dynamic mechanisms such tip–stalk cell transitions and pillar development should be studied using live imaging at single-cell resolution. Multi-omics profiling of endothelial states under hypoxia and flow, organoid and 3D vascular models with chiral constraints to mimic in vivo vessel coiling, and computational frameworks that use mathematical modeling and machine learning to simulate and predict vascular patterning under diverse stimuli are promising [174].

A complete angiogenesis model must include molecular signaling and biomechanical and physical limitations that determine three-dimensional tissue architecture and endothelial function.

Understanding how cells sense polarity, scale it into collective patterns, and integrate cues from their microenvironment remains a frontier in vascular biology [239].

A coherent understanding of vascular morphogenesis requires integration across molecular signaling, cell mechanics, and tissue architecture. We propose a multiscale framework that centers on the VEGF-VEGFR axis, modulated by biomechanical cues and structural transitions.

At the molecular level, VEGFR2 serves as the principal transducer of VEGF-A signals, activating PI3K/Akt, MAPK/ERK, and eNOS to regulate endothelial proliferation, migration, and survival. Notch signaling intersects this axis via Dll4-mediated lateral inhibition, refining tip/stalk patterning and ensuring spatial control of sprouting [241].

Structurally, these signals are translated into morphogenetic mechanisms such as sprouting (driven by filopodial extension and actin dynamics), intussusception (involving cytoskeletal rearrangement and junctional stabilization), and lumen formation (through vacuole trafficking or intercellular junction remodeling).

Biomechanically, VEGFR2 functions within a mechanosensory complex responsive to shear stress and ECM stiffness. These inputs modulate receptor sensitivity and feedback loops, guiding vascular branching and maturation.

This integrated model underscores the importance of synchronized signaling and physical context in generating organized, functional vascular networks. It provides a conceptual foundation for therapeutic strategies aimed at precisely modulating angiogenesis in pathological settings [242].

#### 4.7.1. Gene Therapy

Gene therapy has also been used to directly transfer pro-angiogenic factors to ischemic tissues to trigger angiogenesis. Delivery methods include plasmid DNA, adenoviruses, AAV, and lipid-based systems expressing VEGF-A, FGF2, HIF-1α, or Angiopoietin-1. Gene therapy can overcome the transitory effects of protein or peptide administration by expressing growth factors continuously.

Clinical experiments employing vectors like AdVEGF121 and AdHIF-1α have shown increased local perfusion and reduced ischemia symptoms in individuals with peripheral arterial disease (PAD). While bigger placebo-controlled studies had varied findings, some have failed to satisfy key effectiveness goals. Poor transfection effectiveness in human tissues, immunological responses to viral vectors, and difficulty adjusting transgene dose, duration, and geographic distribution are major limitations.

The heterogeneity of ischemic disease, particularly in elderly patients with comorbidities like diabetes, further complicates therapeutic response [19].

Peptide drugs represent a middle ground between protein therapeutics and gene therapy. Designed to mimic functional domains of angiogenic factors or their receptors, these molecules are typically more stable than full-length proteins and more selective than small molecules. VEGF mimetic peptides, angiopoietin-derived sequences, and integrin-binding RGD peptides have all shown potential to stimulate endothelial behavior conducive to angiogenesis. Advances in peptide engineering have enabled modifications that enhance half-life, tissue specificity, and resistance to proteolysis. Conjugation to delivery platforms such as nanoparticles, hydrogels, or targeting ligands further enhances their utility. However, few of these peptides have progressed beyond preclinical studies, and, to date,, no peptide-based pro-angiogenic therapy has received FDA approval [243].

#### 4.7.2. Precise Modulation of Chirality, Flow, Signaling Balance) in Therapeutic Strategies

Pro-angiogenic drugs are difficult to develop into therapeutic medicines because of angiogenesis’ intricacy. Endothelial cell activity, extracellular matrix remodeling, mural cell recruitment, and hemodynamic adaptation must be coordinated for vascular expansion. Angiogenesis without control can lead to leaky, non-perfused, and poorly integrated vasculature, highlighting the need for more complex methods than “turning on” VEGF signaling [234].

Recent endothelial cell biology findings reveal previously neglected angiogenic success determinants. Cell chirality—the natural left–right asymmetry of cytoskeletal architecture and directed migration—helps endothelial alignment and lumen creation, but disrupting it damages cell–cell junctions and increases vascular permeability. Hemodynamic cues including shear stress and cyclic strain influence endothelial gene expression, vessel diameter, and remodeling. Flow-dependent activation of the VEGFR2 mechanosensory complex is required for vascular maturation and pruning, suggesting that therapies without biomechanical inputs may stimulate vessel development but not functioning [235].

The future of therapeutic angiogenesis may involve molecular stimulation and biomechanical and structural guiding. Smart delivery systems that release pro-angiogenic drugs based on local oxygen levels, flow patterns, or matrix stiffness are possible. Tissue-engineered scaffolds with VEGF mimetics and matrix-bound guide signals may improve vessel development precision. Additionally, single-cell transcriptomics and biomechanical modeling may improve computational predictions, tailored treatment regimens, and clinical results [236].

While FDA-approved anti-angiogenic medicines like bevacizumab (anti-VEGF-A) are routinely utilized in oncology and ophthalmology, pro-angiogenic therapies are experimental. Several VEGF mimetic peptides, gene therapy vectors, and small-molecule pro-angiogenic drugs are in clinical trials, but none have been FDA-approved. Progress requires technical innovation and a better understanding of human disease-related vascular growth and stabilization [237].

The complexity of vascular development must be accepted to advance therapeutic angiogenesis. Considering flow dynamics, extracellular matrix structure, endothelial chirality, and metabolic context, future therapies should combine molecular activation with microenvironmental management. Therapeutic angiogenesis may potentially cure ischemic illness safely, effectively, and durably by aligning targeted molecular therapies with biomechanical and structural cues [238]. (Figure 2)

Panel (A): VEGF signaling. VEGF binding to endothelial VEGFRs triggers receptor dimerization and autophosphorylation, leading to recruitment of adaptor proteins such as GRB2 and activation of downstream cascades including PI3K–Akt and MAPK/ERK. These pathways drive endothelial proliferation, migration, and survival, making VEGF signaling the primary regulator of angiogenesis, particularly under hypoxic conditions [244].

Panel (B): Notch signaling. This juxtacrine pathway determines tip–stalk cell specification during sprouting angiogenesis. Endothelial ligands such as Delta-like 4 (Dll4) or Jagged bind to Notch receptors on neighboring cells, triggering proteolytic cleavage and release of the Notch intracellular domain (NICD). Once in the nucleus, NICD activates transcription of target genes including Hes and Hey. Through lateral inhibition, Notch reduces VEGFR2 expression in stalk cells, ensuring orderly sprout formation and limiting excessive branching.

Panel (C): Angiopoietin/Tie2 signaling. Angiopoietins modulate vascular stability and maturation. Ang-1 binding to Tie2 receptors promotes endothelial quiescence, tight junction formation, and pericyte recruitment, thereby stabilizing the vascular wall. In contrast, Ang-2 antagonizes Ang-1 activity, sensitizing vessels to sprouting stimuli or promoting regression, depending on context.

Panel (D): BMP/TGF-β signaling. Members of the TGF-β superfamily, such as BMPs, regulate vascular development and homeostasis. BMP ligands bind endothelial type I and type II receptor complexes, activating Smad transcription factors that translocate to the nucleus and control gene expression. This pathway shapes endothelial cell fate, branching morphogenesis, and vascular identity.

Together, these pathways form an integrated molecular network that orchestrates vascular development, angiogenic sprouting, and vessel stabilization (see Figure 3) [240,245].

Top-left panel: VEGF signaling. VEGFR1 and VEGFR2 interact with VEGF to initiate downstream cascades. Upon ligand binding, these receptors activate ERK and PI3K pathways: ERK promotes endothelial cell proliferation, while PI3K regulates migration and permeability. VEGF is therefore a central driver of angiogenesis, particularly under hypoxic conditions [246].

Top-right panel: Notch signaling. A membrane-bound Delta ligand from an adjacent cell engages the Notch receptor, triggering proteolytic cleavage and release of the Notch intracellular domain (NICD). NICD translocates to the nucleus and activates transcription of Hes and Hey genes. By suppressing tip cell fate and reinforcing stalk cell identity, Notch signaling coordinates orderly vessel branching through lateral inhibition [247].

Bottom-left panel: TGF-β signaling. TGF-β ligands bind to a receptor complex consisting of Type I and Type II receptors. This interaction phosphorylates Smad proteins, which move into the nucleus to regulate transcription. Smad-mediated gene expression maintains endothelial quiescence and prevents excessive or uncontrolled proliferation.

Bottom-right panel: PR signaling. Progesterone receptor (PR) signaling occurs when ligand binding activates PR, allowing it to translocate into the nucleus and regulate target gene expression. PR signaling influences vascular remodeling and endothelial gene regulation in hormone-sensitive tissues, though it has been less extensively studied compared with VEGF, Notch, or TGF-β pathways.

Together, these four signaling networks regulate endothelial proliferation, migration, quiescence, and specialization, orchestrating vascular growth and remodeling [248]. Figure 4 illustrates the role of cellular chirality in vascular morphogenesis.

This image shows six conceptual aspects that explain the origin and quantification of endothelial cell chirality during vascular morphogenesis. Each feature reveals a mathematical or mechanistic explanation of chiral activity at the cellular level and how to characterize or quantify it.

The first conceptual element, shown in the top left, introduces the fundamental physics of torque, which is described mathematically by the vector cross product formula T = r × F. Here, $\vec{T}$ represents torque, $\vec{r}$ is the position vector from the center of mass, and $\vec{F}$ is the applied force. This expression captures how a force applied at a distance from the center of mass can generate rotational motion. In the context of a rotating endothelial cell, this torque arises of intracellular troops, particularly those generated by actin and myosin filaments. The direction and magnitude of this torque are central to understanding how physical asymmetries within the cell lead to a rotational bias, quantifiable by directionality metrics, a key manifestation of chirality [249].

Moving to the top center of the diagram, we see a schematic of a polarized endothelial cell with actin filaments extending asymmetrically within the cell body. These cytoskeletal filaments generate mechanical forces as they polymerize. This is described using the equation τfil = k, where τfil is the torque produced by the actin filaments, θ is the angular displacement, and k is a constant representing the system’s stiffness. This relationship indicates that torque increases linearly with angular displacement, resulting from the uneven distribution and activity of the actin cytoskeleton. Such asymmetry is a source of intrinsic chirality in endothelial cells [250].

In the top right section, the diagram provides additional physical context by connecting torque to angular velocity. Two key equations are shown: τfilament = −kθ, reiterating the torque-angular displacement relationship with a negative sign indicating a restoring torque, and ω = dθ/dt, which defines angular velocity ω as the time derivative of angular position θ, also expressed in terms of torque τ and moment of inertia I. These expressions are crucial for describing how internal cytoskeletal forces produce actual rotational motion of the cell, providing a quantitative framework for measuring and predicting chiral behavior [251].

In the bottom left, a conceptual sketch of a random trajectory illustrates how cell motion is not entirely random but often exhibits an average directional bias over time. While the path of individual cells may appear stochastic or wiggly at short timescales, on longer timescales, a preferred rotational direction (leftward or rightward) can emerge due to intrinsic cellular chirality. This directional bias reflects the cumulative effects of underlying asymmetries in force generation or cytoskeletal organization [252].

The bottom center provides a stochastic model to describe this behavior mathematically. The equation dθ/dt = 1/I + η(t) combines deterministic rotation (represented by 1/I with a stochastic noise term η(t), capturing both the predictable and random components of angular motion. This model acknowledges that while there may be a consistent rotational tendency, actual trajectories are also influenced by random fluctuations. To quantify the overall directional bias across a population of cells, the chirality index C is introduced, defined as C = 1/N∑sinθi, where θi are the observed rotation angles of individual cells, and N is the total number of cells measured. A nonzero value of C indicates a consistent population-level preference for rotation in one direction [253].

Finally, in the bottom right, a histogram of rotation angles θ\thetaθ provides a visual summary of the empirical distribution of cellular rotational behavior. The histogram shows a skewed, right-shifted distribution, meaning more cells exhibit positive (rightward or clockwise) rotation. This corresponds to a chirality index C > 0 indicating a net right-handed chirality in the cell population. The bar chart below the histogram further illustrates this directional bias, emphasizing that even within seemingly random behavior, a consistent and measurable chiral trend emerges [254].

Together, these six elements form an integrated conceptual and mathematical framework for understanding how intrinsic cellular chirality emerges, manifests in physical motion, and can be quantified at both the single-cell and population levels during processes such as vascular morphogenesis. The quantitative characterization of cellular chirality is illustrated in Figure 5.

This figure features a 2 × 3 matrix layout of related models and formulas, arranged clearly and concisely for enhanced clarity. Top Left: Torque. Same concept as Diagram 1: T = $\vec{r}$⋅$\vec{F}$. $\vec{F}$ Depicts torque induced by cytoskeletal forces [255]. Top Middle: Angular Velocity ω = dθ/dt Shows how cell rotation is driven by internal torque relative to inertia [256]. Top Right: Filament Curvature τ = −καEI Describes how curvature of actin filaments (with stiffness EI) generates rotational torque [257]. Bottom Left: Biased Random Walk dθ/dt = ω + ε(t) Adds noise (ε(t)) to deterministic angular velocity—this models real biological randomness in cell paths [258]. Bottom Center: Chirality Index C = ∣ω∣/ωs Compares average angular velocity to a reference or maximal value ωs to normalize chirality strength [259]. Bottom Right: Directional Bias cos(α) = η Measures deviation angle α between the movement vector and the ideal direction. The cosine function captures how closely aligned the motion is [260].

These diagrams work together to bridge molecular, physical, and statistical models of cellular chirality. They are excellent visual aids to enhance understanding of how cytoskeletal asymmetry and signaling feedback translate into consistent chiral behavior during vascular development.

This figure presents a multiscale schematic overview of how cellular chirality in endothelial cells can be described, visualized, and quantified through a combination of mechanical forces, cytoskeletal dynamics, and statistical population behavior. Each panel illustrates a different aspect of chirality—from single-cell force dynamics to collective behaviors—linking mathematical formalism to biological function in angiogenesis [261].

A schematic of an endothelial cell shows internal actin filaments generating forces at points offset from the cell’s center of mass. The resulting torque vector T is computed using the cross product:T = r × F
where r is the position vector from the center to the point of force application, and F is the applied actomyosin force. This torque determines the rotational direction of the cell, which, in endothelial cells, typically manifests as a right-handed (clockwise) bias [262].

A curved actin filament is depicted under mechanical strain. Its torque is described as:τ = μL⋅∂^2^u/∂x^2^
where μL is the bending rigidity of the filament, and ∂^2^u/∂x^2^ is its curvature. The balance of filament tension and curvature creates a net moment acting on the cellular architecture, thereby reinforcing local chiral orientation [263].

This panel links net cytoskeletal torque (τ) to angular velocity (ω) via:ω = dθ/dt

Here, I is the moment of inertia of the cell’s mass distribution. This equation illustrates how internal torque drives rotational behavior and contributes to a persistent chiral bias during migration and tube formation [264].

Illustrated here is a 2D trajectory of an endothelial cell undergoing biased migration. The path deviates consistently to the right. This is modeled as:x(t + Δt) = x(t) + vΔt cos(θ + Δθ)y(t + Δt) = y(t) + vΔt sin(θ + Δθ)
where v is velocity, θ is the mean migration angle, and Δθ is a chiral angular deviation sampled from a skewed distribution. This model captures how chirality affects long-term displacement even in noisy environments [265].

This differential equation represents fluctuating angular momentum:dθ/dt = ω + η(t)

Here, η(t) is a Gaussian noise term representing environmental or internal fluctuation. This equation explains how even cells with identical molecular wiring can exhibit statistical variability in their chiral bias due to stochastic perturbations [266].

This graph shows a histogram of cell angular orientations (θ) across a population. The chirality index C is computed as:C = N_R_ − N_L_/N_R_ + N_L_
where $N_{R}$ is the number of clockwise-biased (right-handed) cells and $N_{L}$ the number of counterclockwise-biased (left-handed) cells, a population with C = 1 is uniformly right-handed, while C = 0 is random, and C = −1 is uniformly left-handed. This metric helps assess the collective bias in vitro or silico models of endothelial networks.

Together, these panels bridge single-cell biomechanics (force and torque generation) with tissue-level outcomes (collective alignment and rotational bias, quantifiable by directionality metrics. The integration of actomyosin-based tension, cytoskeletal filament strain, and random walk modeling provides a theoretical framework for understanding how cellular chirality is encoded and sustained during sprouting angiogenesis [267].

These principles explain key experimental observations such as:Right-handed tube helicity in 3D endothelial culturesDisruption of chiral bias upon inhibition of PKC, mDia1, or Cdc42Flipping of vessel orientation with exogenous signaling or cytoskeletal interference

By quantifying chirality across scales—from filament strain to angular velocity to population statistics—this figure highlights the biophysical mechanisms by which endothelial cells translate intracellular asymmetry into vascular architecture. This has implications for vascular tissue engineering, pathological angiogenesis, and understanding the integration of mechanical and molecular signaling in morphogenesis [268].

The integrated visual panels presented in this figure provide a conceptual and quantitative bridge between intracellular biomechanical forces and macroscopic tissue-level vascular architecture. These mathematical representations—grounded in torque generation, filament bending, stochastic angular displacement, and population-level statistics—reveal how seemingly subtle asymmetries at the cytoskeletal level give rise to emergent directional behaviors in endothelial cells [266].

At the single-cell level, chiral migration is governed by the non-uniform distribution of forces generated by the actin-myosin cytoskeleton. The torque equation T = r × F illustrates how peripheral actin polymerization, when applied asymmetrically relative to the cell’s center of mass, leads to a net rotational moment. This behavior is not stochastic; it is structured and biased, as demonstrated by the consistent right-handed rotation in endothelial cultures [232].

The relation ω = dθ/dt = τ/I links this internal torque to observable angular velocity. In this model, cell shape (moment of inertia I) modulates how quickly the cell rotates in response to internal force distributions. This angular behavior is further refined by actin filament bending resistance, as captured in the equation τ = μL ∂^2^u/∂x^2^, which describes how cytoskeletal tension stores and redistributes energy in a mechanically polarized manner [233].

These mechanical frameworks are complemented by stochastic models that account for biological variability. In particular, the biased random walk equations:x(t + Δt) = x(t) + vΔt cos(θ + Δθ)y(t + Δt) = y(t) + vΔt sin(θ + Δθ)
simulate how directional bias (encoded in angular displacement Δθ) manifests over time. Even in the presence of environmental noise, the net trajectory of chiral cells deviates predictably, resulting in curved migration paths and spiral structures observed in vitro.

At the population level, this deviation becomes quantifiable using the chirality index [266]:C = N_R_ − N_L_/N_R_ + N_L_
where $N_{R}$ and $N_{L}$ represent the number of clockwise- and counterclockwise-biased cells, respectively. In endothelial cultures, typical values of C range from +0.4 to +0.8 under normal conditions, with reductions to near zero upon perturbation with cytoskeletal inhibitors. This statistical descriptor serves as a robust metric to assess the degree of collective chiral behavior in both experimental and computational angiogenesis models [269].

These models clarify a number of pivotal experimental observations:

Right-handed tube helicity, observed in 3D fibrin gels and tubular scaffolds, directly corresponds to net positive torque generated by cortical actin asymmetry.

Loss of chirality upon inhibition of PKC, Cdc42, or formin proteins (e.g., mDia1) is mechanistically reflected in the disappearance of net torque and filament curvature, resulting in random or misaligned migration.

Reversal of handedness with pharmacological stimuli (e.g., PMA or Gö6983) aligns with model predictions: a change in sign of τ or in bias of Δθ flips the net angular velocity and alters vessel patterning [270].

These phenomena are not artifacts of in vitro conditions—they are mirrored in embryonic development. In zebrafish intersegmental vessels, asymmetric actin distribution correlates with sprout directionality and efficiency of anastomosis. In murine embryos, endothelial chirality correlates with left-right developmental genes and is perturbed in models of situs inversus, suggesting a deeply conserved role in vascular symmetry [271].

The implications of these insights are significant:

In tumor angiogenesis, where VEGF is elevated but vessel architecture is disorganized, disrupted chirality could explain non-productive sprouting and the formation of tortuous, leaky vessels. Targeting actin polarity regulators may help “normalize” tumor vasculature.

In diabetic microvascular disease, where endothelial barrier function is compromised, impaired chiral coordination may underlie junctional instability and increased permeability.

In vascular tissue engineering, enforcing chiral cues (e.g., via ECM microtopography or magnetic guidance) can align endothelial cells, promoting efficient perfusion and barrier function in artificial grafts [272].

These equations and visual models illustrate that chirality is not an epiphenomenon—it is a foundational organizational axis, alongside chemical gradients (e.g., VEGF), cell–cell signaling (e.g., Notch), and mechanical strain. Its quantitative characterization via torque, angular velocity, biased migration, and chirality indices provides a rigorous framework for evaluating and manipulating vascular patterning across developmental, pathological, and engineered contexts [273].

These equations and visual models demonstrate that cellular chirality is not merely an epiphenomenon or byproduct of stochastic cell behavior; rather, it represents a fundamental spatial and mechanical axis of vascular organization. Angiogenesis is regulated by molecular gradients (e.g., VEGF-A), cell–cell communication systems (e.g., Dll4–Notch lateral inhibition), and biomechanical cues (e.g., shear Chirality regulates endothelial orientation, migration, and vascular network creation with orthogonal, rotating logic [274,275]. This chiral bias helps endothelial monolayers align and rotate, regulating vascular morphogenesis at the multicellular level.

By biasing how cells rotate, align, and exert mechanical tension on one another, chirality influences tissue-level properties, such as vessel curvature, branching angles, and tubular helicity. Importantly, these features are not cosmetic—they directly affect fluid flow, barrier function, and hierarchical organization of the vascular tree.

Quantitatively, chirality can now be measured, modeled, and manipulated. Mechanical concepts, such as torque (T = r × F), angular momentum (ω = τ/I), and filament bending stiffness (τ = μL ∂^2^u/∂x^2^), enable researchers to link intracellular asymmetries to observable phenotypes. Stochastic models of biased random walks and population-level metrics, such as the chirality index (C), allows the evaluation of chiral organization across entire endothelial cultures or developing vessels [274].

This shift from qualitative observation to quantitative formalism marks a paradigm advancement in vascular biology. It means that chirality can be incorporated into predictive models of angiogenesis, used to interpret aberrant vessel formation in disease, and targeted as a modifiable parameter in experimental and clinical contexts. For example:

In regenerative medicine, tissue scaffolds could be engineered with microtopographies or force fields that enforce consistent chirality, guiding endothelial patterning and lumen integrity.

In oncology, reversing or disrupting tumor-induced loss of chirality might help normalize vessel architecture, thereby enhancing drug delivery and oxygenation.

In vascular aging and diabetic pathology, quantifying chirality loss could serve as a biomarker of endothelial dysfunction, complementing existing measures of permeability or inflammation [276].

Ultimately, chirality introduces a rotational dimension to angiogenesis—not only spatially, but also mechanistically and therapeutically. By acknowledging and integrating this axis of control, researchers can construct a more comprehensive, multi-vectorial model of vascular morphogenesis, one that accounts not only for where and when endothelial cells move but also how they organize their movement in relation to each other and to the forces and fields that surround them. Next, cellular chirality as an axis of vascular organization is represented in Figure 6.

This schematic presents cellular chirality as a fundamental axis of angiogenic control, integrating mechanistic, structural, and applied dimensions. It connects intracellular cytoskeletal dynamics with tissue-level behavior and translational applications. The top section of the figure focuses on the cellular and multicellular levels. In sprouting angiogenesis, individual endothelial cells do not migrate in straight lines but instead follow directionally biased angular trajectories. These paths often involve right-handed (clockwise) rotational bias quantifiable by directionality metrics, which is driven by cytoskeletal torque and internal asymmetries regulated by signaling pathways such as VEGFR2, PKC, and Cdc42. Chirality at the cellular level plays a critical role in orienting migration, optimizing the spatial distribution between tip and stalk cells, and facilitating vessel extension in complex microenvironments [277].

At the multicellular level, chirality manifests as coordinated rotational alignment of endothelial cells along vessel walls. This promotes structural features like helical wrapping, monolayer cohesion, and consistent vessel curvature and branching angles. These collective behaviors impact both local hemodynamics and overall tissue perfusion, underscoring the importance of chirality in maintaining vascular function [278].

On the right side of the schematic, several clinical applications of endothelial chirality are highlighted. In regenerative medicine, chirality can be exploited to guide lumen formation and promote alignment of endothelial cells in engineered tissues. Techniques such as microtopographical patterning, externally applied bioelectric fields, or the use of chirality-inducing peptides can be employed to impose a consistent rotational bias, quantifiable by directionality metrics within synthetic scaffolds. In oncology, tumor vasculature often displays chaotic or even reversed chirality, which contributes to poor perfusion, vessel leakiness, and inefficient drug delivery. Therapeutic restoration of normal chiral alignment may improve oxygenation and treatment access within tumors. In vascular aging and diabetes, cytoskeletal disorganization leads to impaired chirality. Measuring this loss of chirality may serve as a quantitative biophysical biomarker for endothelial dysfunction and frailty [279].

The bottom panel introduces the quantitative modeling tools used to describe and simulate cellular chirality. Torque is described by the equation T = $\vec{r}$ × $\vec{F}$, which captures the rotational force generated by actin filaments when they apply force at an offset from the cell center. Angular momentum is given by ω = τ/I, indicating that chiral motion depends on both the internal torque and the moment of inertia determined by the cell’s morphology. Filament bending stiffness is modeled using τ = μL^2^ × ∂u/∂x, which describes how curvature in actin filaments contributes to internal torque. The concept of a biased random walk is also presented, illustrating how directional migration paths systematically diverge from random trajectories due to persistent internal biases. Population-level chirality is quantified using a chirality index, where a value C>0C > 0C>0 indicates a right-handed dominant population. The index is calculated as C = 1/N∑sinθi providing a statistical measure of net rotational bias, quantifiable by directionality metrics across cells [280].

A complementary flowchart figure frames cellular chirality within the broader context of vascular biophysics and angiogenic signaling. On the input side, several key regulators are identified. VEGF-A promotes cytoskeletal polarization through VEGFR2 and downstream signaling cascades involving PI3K, MAPK, and PLCγ. The Dll4–Notch pathway helps define tip versus stalk cell identity and thereby determines how polarity and chirality are spatially distributed across the sprouting front. Shear stress, as a mechanical cue, influences endothelial alignment via mechanotransduction mechanisms that involve VE-cadherin, PECAM-1, and VEGFR2 complexes. This can reinforce or modulate chiral behavior depending on the flow environment [281].

At the center of the flowchart is the concept of cellular chirality itself, depicted as a core node driving asymmetric behavior both at the single-cell and collective levels. It is visually represented by curved actin filaments and bidirectional torque vectors, reflecting the intrinsic rotational bias quantifiable by directionality metricsof endothelial cells. From this central node, several downstream impacts are described. At the cellular level, chirality biases the directional migration of tip cells and is essential for maintaining the structured geometry of vascular sprouts. At the multicellular level, it contributes to the alignment of endothelial layers and governs macroscopic features such as vessel curvature and branching geometry. Loss or reversal of chirality is closely associated with pathological vascular states [218].

Quantitative models now make it possible to simulate and predict the effects of chirality using formal mechanical tools such as torque, angular momentum, and filament stiffness. These models provide a mechanistic framework for understanding vascular organization and for designing experiments or interventions. Applications span regenerative medicine—where guiding endothelial alignment may support functional vessel formation—oncology—where restoring chiral behavior may normalize tumor vasculature—and metabolic or age-related vascular diseases—where chirality loss could serve as an early biophysical indicator of dysfunction [282].

Through this integrated lens, chirality emerges as not just a byproduct of cytoskeletal dynamics but a central, controllable axis of vascular morphogenesis. It offers predictive power, clinical relevance, and new opportunities for therapeutic innovation in angiogenesis-related fields.

Building on these foundations, emerging models of cellular chirality now incorporate feedback between mechanical torque generation and intracellular signaling. For example, the dynamic evolution of angular velocity ω(t) can be modeled as a differential system influenced not only by external torque but also by active remodeling of the cytoskeleton [283]:dω/dt = 1/I (τactive − γω)

Here, τactive represents active, regulated torque driven by actomyosin contractility, γ is a damping coefficient modeling cytoplasmic resistance, and η(t)\eta(t)η(t) remains a stochastic noise term. This formulation captures how persistent chiral rotation can emerge as a limit cycle—a stable, self-sustaining oscillatory state—rather than purely as a decaying response to initial torque. Furthermore, spatial gradients in chirality-related proteins can be introduced into migration models through angular drift terms, such as:Dθ/dt = ω0 + β∇C(x, y) + η(t)
where ∇C(x, y) represents a local chemical or mechanical gradient and β quantifies the cell’s sensitivity to that gradient in a chiral context. These extensions open the door to simulating how local microenvironments modulate chirality on a cell-by-cell basis, potentially explaining how the same cell type can exhibit leftward or rightward bias depending on context. Integrating these frameworks with live-imaging data and single-cell transcriptomics will be crucial for bridging physical models with gene-level regulatory logic, further cementing chirality’s role as a central axis of control in both physiological and pathological vascular remodeling [280].

Vascular morphogenesis can be viewed as a coordinated, multi-scale program in which molecular signals, physical forces, and emergent tissue architecture are tightly coupled. At the molecular scale, VEGFR2 is the central driver of angiogenic signaling, integrating PI3K–Akt for survival and polarity, PLCγ–PKC–ERK for proliferation, and p38 MAPK–HSP27 for cytoskeletal remodeling. Notch continuously refines these pathways–Dll4 lateral inhibition, which enforces orderly tip/stalk patterning, and by VEGFR1 and VEGFR3, which modulate ligand availability and dampen excessive VEGFR2 activity. At the cellular scale, these inputs produce distinct morphogenetic behaviors: (1) in sprouting, asymmetric activation of Cdc42, Rac1, and RhoA directs filopodial extension, migratory polarity, and cell–cell coordination; (2) in intussusception, VEGFR2 engagement within PECAM-1/VE-cadherin mechanosensory complexes under low-flow conditions triggers pillar initiation and vessel splitting; and (3) in lumen formation, Rab GTPase–driven vesicle trafficking and VEGFR2-mediated junctional remodeling establish and stabilize perfusable conduits. At the biomechanical scale, endothelial cells sense and respond to shear stress, matrix stiffness, and three-dimensional curvature, adjusting receptor sensitivity, cytoskeletal tension, and polarity vectors accordingly. The recently discovered organizing principle of cellular chirality biases actin filament alignment, migration trajectories, and junctional geometry, affecting branch angles, vessel curvature, and lumen continuity. Figure 7 shows how biochemical gradients, mechanotransduction, and innate polarity programming create functioning, hierarchically structured vascular networks.

Finally, Figure 8 shows the review flow chart. This review integrates molecular signaling, structural morphogenetic principles, and biomechanical regulation into a multiscale framework to explain vascular formation.

One key novelty lies in the depth and clarity with which the VEGF–VEGFR signaling axis is dissected, especially the distinct and overlapping roles of VEGFR1, VEGFR2, and VEGFR3 in angiogenesis, lymphangiogenesis, and vascular permeability. The review also goes beyond classical molecular regulation by systematically incorporating emerging topics that have been underrepresented in the angiogenesis field, such as cellular chirality and its impact on endothelial polarity, lumen formation, and vessel patterning. Notably, the integration of chirality as a spatial regulator, modulated by both intrinsic cytoskeletal dynamics and extrinsic signals like VEGF gradients and shear stress, provides a fresh conceptual dimension to understanding vascular organization. The review’s structural section unpacks mechanisms such as sprouting, intussusception, and lumenogenesis with mechanistic granularity, linking them to upstream signaling. Furthermore, the manuscript distinguishes itself by exploring VEGF mimetics, gene therapies, and biophysical context-specific interventions, critically evaluating their translational challenges. Finally, it advocates for precision-modulated angiogenesis, stressing the need for treatments that harmonize biochemical signaling with biomechanical context—offering a roadmap toward functional and durable vascular regeneration. This conceptual synthesis sets the review apart by bridging reductionist molecular biology with emergent biophysical and spatial patterning principles.

Taken together, recent advances indicate that any single dominant pathway does not govern vascular morphogenesis, but rather by a distributed and context-dependent network in which VEGFR isoform balance, Notch–Dll4 lateral inhibition, mechanical forces, and intrinsic cellular polarity programs—including chirality—are dynamically integrated. This synthesis highlights that precise spatial and temporal regulation, rather than maximal activation of pro-angiogenic signals, is the key determinant of functional vessel architecture. Moving forward, several priority areas emerge. First, live-cell and in vivo imaging with single-cell resolution will be essential to map dynamic transitions such as tip–stalk switching and pillar insertion during intussusception. Second, integrating biomechanical readouts—shear stress, ECM stiffness, and 3D curvature—into molecular pathway studies will clarify how physical context modulates signaling outputs. Third, chirality-guided engineering approaches, including microtopographic scaffolds and matrix-bound VEGF mimetics, could be exploited to improve vessel alignment and barrier integrity in regenerative therapies. Finally, computational models that combine molecular kinetics, force distribution, and spatial patterning could enable predictive control over angiogenesis in both developmental and pathological contexts. Addressing these challenges will require coordinated efforts across cell biology, bioengineering, and translational research. Still, it offers a clear path toward the rational design of vascular networks that are structurally coherent, functionally perfused, and clinically durable.

## 5. Conclusions

The vascular angiogenesis and subsequent morphogenesis process indicate a complicated molecular network. Additional research is needed to delineate the relationships among the implicated compounds accurately. Concerning the VEGFR pathway, VEGFR2 appears to be a pivotal component. The principal components of the VEGFR pathway are the VEGF receptors VEGFR-1, VEGFR-2, and VEGFR-3. Multiple variants of VEGF are under investigation for potential medicinal use. Investigating the NOTCH and VEGFR signaling pathways may enable the modulation, enhancement, or suppression of vascular morphogenesis.

Cellular chirality—the inherent “handedness” of cells in movement, structure, and organization—plays a pivotal role in angiogenesis, the process by which new blood vessels form from pre-existing ones. This chiral behavior is fundamental to the directional and patterned organization of endothelial cells during vascular development and remodeling. In angiogenesis, cellular chirality guides endothelial cells to adopt precise orientations and migration patterns, which are crucial for forming functionally structured blood vessels that supply tissues with the necessary nutrients and oxygen. Cellular chirality in this context is influenced by various factors, including VEGF/VEGFR signaling, mechanical forces, interactions with the extracellular matrix (ECM), and cytoskeletal dynamics.

These developments underscore that successful therapeutic angiogenesis will not only depend on the stimulation of vessel growth, but also on the precise modulation of vascular morphogenesis through the coordinated control of signaling pathways, cellular behavior, and biomechanical context. Emerging insights into cellular chirality, hemodynamic force sensitivity, and balanced signaling through pathways such as VEGF and Notch reveal that angiogenesis is a highly structured and directional process. To achieve functional and durable vascularization, future strategies must align molecular cues with the spatial orientation, mechanical stimuli, and polarity of endothelial cells, enabling the formation of stable, perfused vessels that are appropriately integrated into the host tissue architecture.

## Figures and Tables

**Figure 1 cimb-47-00851-f001:**
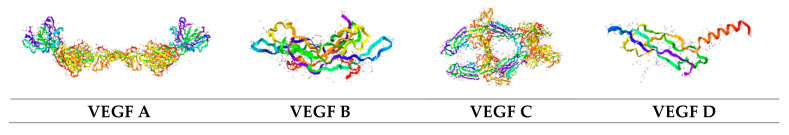
Vascular endothelial growth factors from A-D are represented as ribbons.

**Figure 2 cimb-47-00851-f002:**
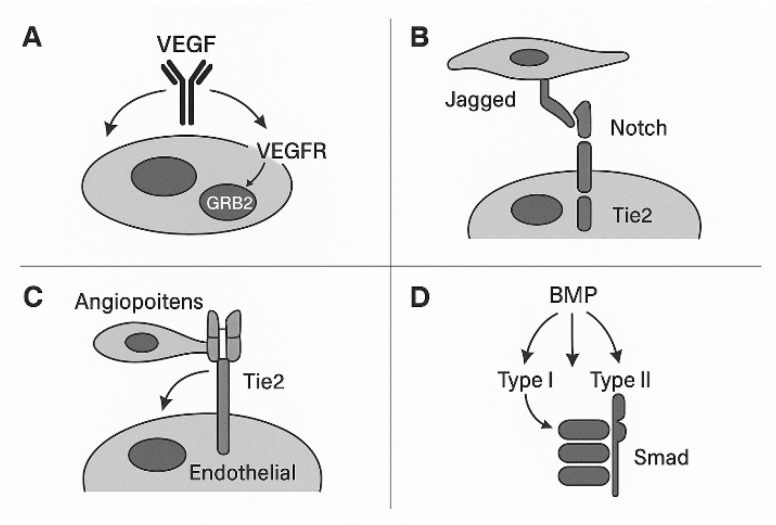
Four major signaling pathways regulate endothelial cell behavior during angiogenesis and vessel remodeling.

**Figure 3 cimb-47-00851-f003:**
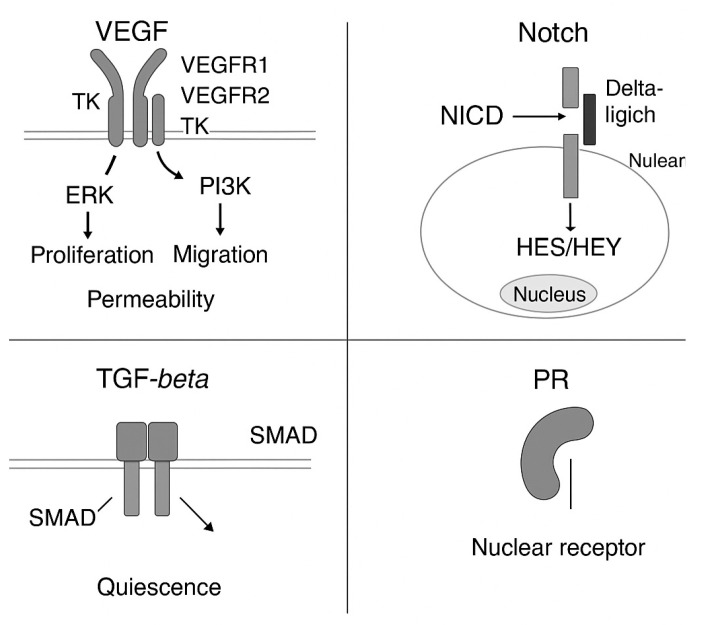
Four panels illustrate key signaling pathways that regulate endothelial cell behavior and vascular morphogenesis, demonstrating how external inputs shape endothelial responses.

**Figure 4 cimb-47-00851-f004:**
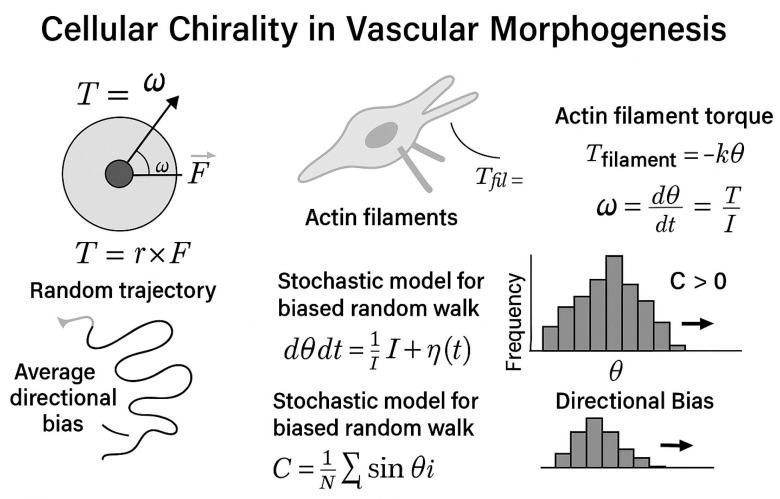
Cellular chirality in vascular morphogenesis.

**Figure 5 cimb-47-00851-f005:**
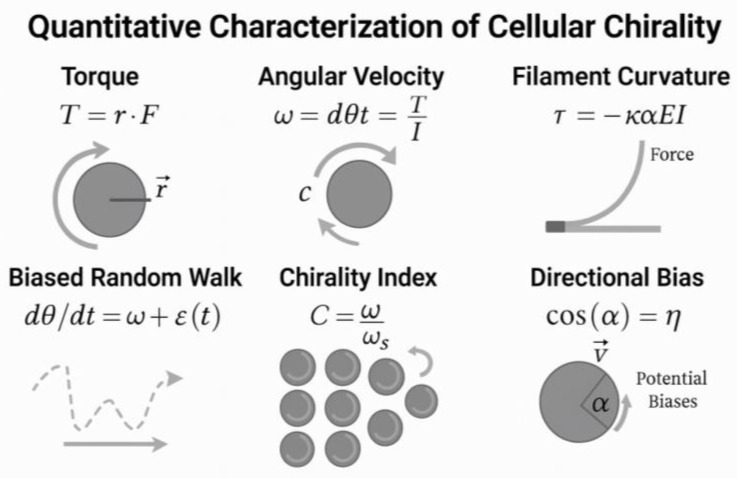
Quantitative characterization of cellular chirality.

**Figure 6 cimb-47-00851-f006:**
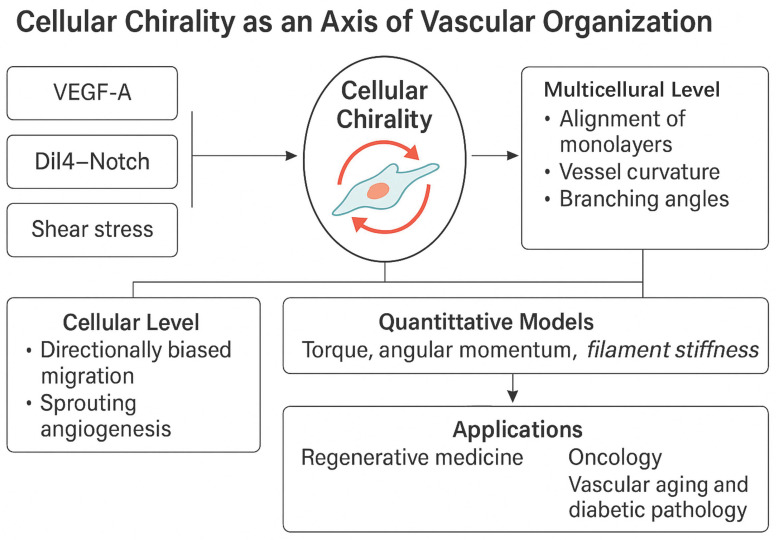
Cellular chirality as an axis of vascular organization.

**Figure 7 cimb-47-00851-f007:**
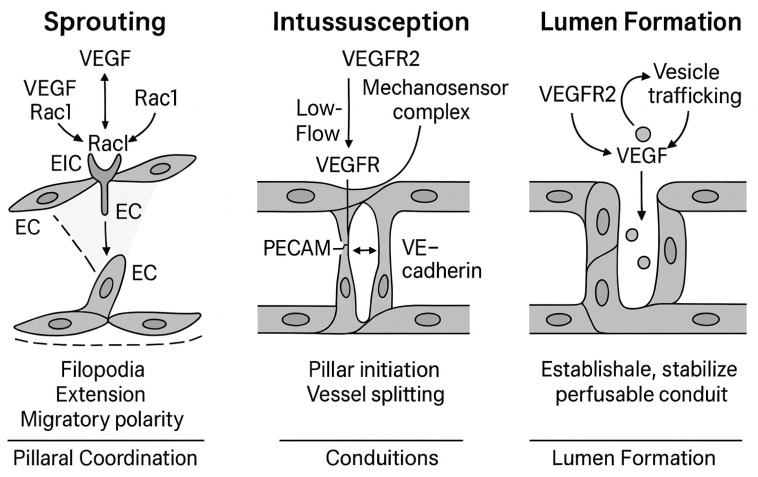
Sprouting, Intussception, and Lumen Formation. This diagram illustrates the three main stages of vascular morphogenesis: sprouting, intussusception, and lumen formation, showing how molecular signals, cellular behaviors, and structural changes are linked. In the first stage, sprouting angiogenesis, vascular endothelial growth factor (VEGF) binds to VEGFR2 on endothelial cells and activates pathways involving small GTPases like Rac1, Cdc42, and RhoA, which remodel the actin cytoskeleton. Tip endothelial cells, with high VEGFR2 activity and low Notch signaling, extend thin filopodia toward the VEGF source, establishing migratory polarity, while adjacent stalk cells, influenced by Notch signaling, proliferate and maintain stable junctions. Together, these coordinated behaviors produce a new vascular branch. In the second stage, intussusceptive angiogenesis, VEGFR2 is activated under low-flow conditions, often in complex with mechanosensory proteins like PECAM-1 and VE-cadherin, which detect shear stress. Endothelial cells from opposite sides of the vessel lumen project into the lumen and meet to form a transluminal pillar, which is then reinforced by extracellular matrix and pericytes, splitting one vessel into two and expanding the network with minimal new cell proliferation. In the final stage, lumen formation, VEGFR2 signaling promotes vesicle trafficking through Rab GTPases and cytoskeletal rearrangements, enabling either the fusion of intracellular vacuoles within a single cell (cell hollowing) or the remodeling of junctions between neighboring cells (cord hollowing). This process establishes a stable, continuous, and perfusable lumen, completing the formation of a functional blood vessel.

**Figure 8 cimb-47-00851-f008:**
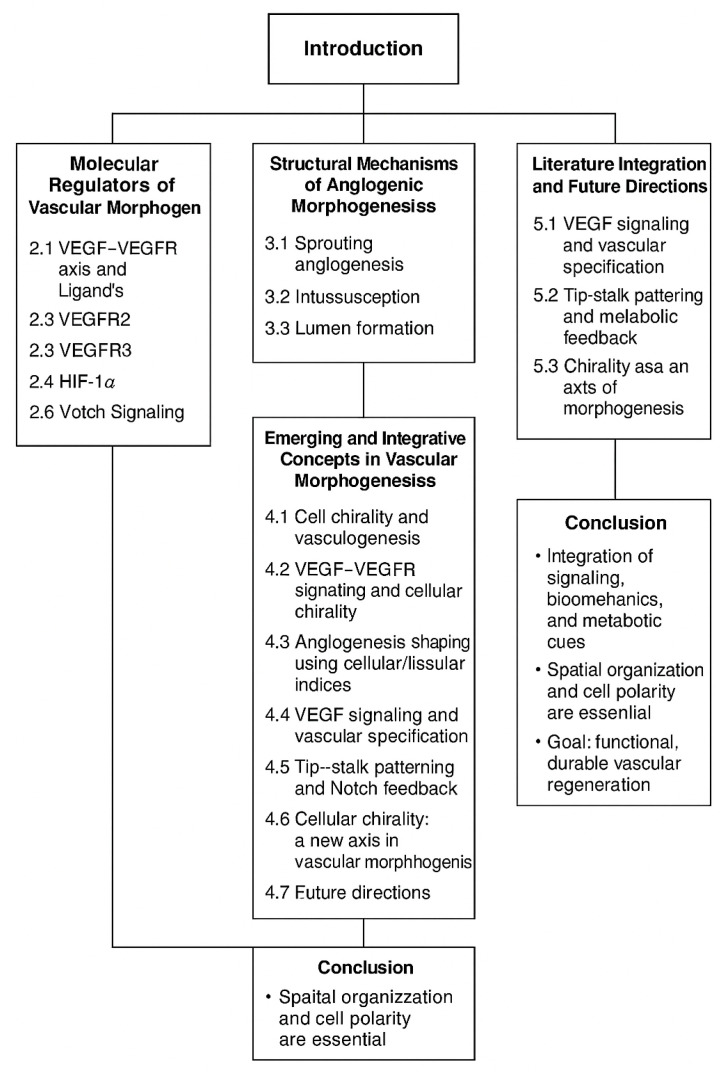
Flow chart of the review.

**Table 1 cimb-47-00851-t001:** VEGFR1 properties.

#	Property	Description	References
1	Structure	VEGFR1 is a transmembrane protein belonging to the receptor tyrosine kinase (R.T.K.) family.	[29]
2	Function	VEGFR1 primarily functions as a receptor for VEGF-A.	[30]
3	Binding Affinity	VEGFR1 has a high affinity for VEGF-A.	[31]
4	Role in Development	VEGFR1 plays a critical role in the formation of the vascular system during embryonic development.	[32]
5	Signal Transduction	Upon activation, VEGFR1 undergoes autophosphorylation and activates downstream signaling pathways.	[33]
6	Angiogenesis Regulation	VEGFR1 is involved in the negative regulation of angiogenesis.	[34]
7	Therapeutic Target	VEGFR1 has been explored as a therapeutic target for antiangiogenic drugs.	[35]
8	Soluble Form	Due to alternative splicing, VEGFR1 can exist in a soluble form (sVEGFR1) and act as a decoy receptor for VEGF-A.	[36]
9	Expression in Cancer	VEGFR1 expression is observed in various cancers and is associated with tumor angiogenesis, progression, and poor prognosis.	[37]
10	Interaction with Neuropilin-1	VEGFR1 can form a complex with neuropilin-1, enhancing VEGF-A binding and signaling.	[38]
11	Regulation by miRNAs	VEGFR1 expression can be regulated by microRNAs (miRNAs) in various physiological and pathological conditions.	[39]
12	Role in Neuroprotection	VEGFR1 has been implicated in neuroprotection and neuronal survival in addition to its role in angiogenesis.	[40]

**Table 2 cimb-47-00851-t002:** VEGFR2 properties.

#	Property	Description	References
1	Structure	VEGFR2 is a transmembrane protein belonging to the receptor tyrosine kinase (R.T.K.) family.	[67]
2	Function	VEGFR2 primarily functions as a receptor for VEGF-A, mediating most—VEGF-induced angiogenic responses.	[26]
3	Signal Transduction	Upon activation by VEGF-A binding, VEGFR2 undergoes autophosphorylation and activates downstream signaling pathways, including the phosphoinositide 3-kinase (PI3K)/Akt pathway and the mitogen-activated protein kinase (MAPK) pathway.	[19]
4	Angiogenic Response	VEGFR2 activation leads to endothelial cell proliferation, migration, and survival, contributing to angiogenesis.	[20]
5	Role in Development:	VEGFR2 plays a crucial role in embryonic vascular development and angiogenesis.	[31]
6	Regulation of Blood Pressure	VEGFR2 signaling plays a crucial role in regulating blood pressure and vascular tone.	[68]
7	Therapeutic Target:	VEGFR2 targets antiangiogenic therapy in cancer and other diseases characterized by abnormal angiogenesis.	[69]
8	Endothelial Barrier Function	VEGFR2 signaling is involved in regulating endothelial barrier function, influencing vascular permeability.	[70]
9	Lymphangiogenesis	VEGFR2 plays a crucial role in lymphangiogenesis, the process by which new lymphatic vessels are formed.	[71]
10	Regulation by miRNAs	VEGFR2 expression can be regulated by microRNAs (miRNAs) in various physiological and pathological conditions	[72]
11	Tie-2 Interaction:	VEGFR2 can form a complex with the Tie-2 receptor, influencing vascular development and stability.	[73]
12	Metastasis Promotion	VEGFR2 signaling has been implicated in promoting tumor metastasis through its effects on tumor vasculature and the migration of cancer cells.	[74]

**Table 3 cimb-47-00851-t003:** VEGFR3 properties.

#	Property	Description	References
1	Structure	VEGFR3 is a transmembrane protein belonging to the receptor tyrosine kinase (R.T.K.) family.	[102]
2	Function	VEGFR3 primarily functions as a receptor for Vascular Endothelial Growth Factor C (VEGF-C) and Vascular Endothelial Growth Factor D (VEGF-D), regulating lymphangiogenesis.	[103]
3	Lymphangiogenesis:	VEGFR3 is a crucial regulator of lymphangiogenesis, the process by which new lymphatic vessels form.	[104]
4	Developmental Role	VEGFR3 plays a crucial role in the development of the lymphatic system, including the sprouting and patterning of lymphatic vessels.	[105]
5	Signal Transduction	Activation of VEGFR3 by its ligands triggers downstream signaling cascades, including the phosphoinositide 3-kinase (PI3K)/Akt pathway and the mitogen-activated protein kinase (MAPK) pathway, which regulate the function of lymphatic endothelial cells.	[106]
6	Role in Cancer Metastasis:	VEGFR3 signaling has been implicated in tumor metastasis by promoting lymphangiogenesis and facilitating the dissemination of cancer cells through lymphatic vessels.	[107]
7	Therapeutic Target	Targeting VEGFR3 has been explored as a potential therapeutic strategy for inhibiting lymphangiogenesis and metastasis in cancer.	[108]
8	Interactions with Neuropilins	VEGFR3 can form complexes with neuropilin receptors, modulating its signaling and function in lymphatic endothelial cells.	[109]
9	Regulation by miRNAs:	VEGFR3 expression can be regulated by microRNAs (miRNAs), which in turn influence lymphangiogenesis and cancer progression.	[110]
10	Angiogenesis in Corneal Lymphatics	VEGFR3 plays a role in angiogenesis in corneal lymphatic vessels, influencing corneal inflammation and wound healing.	[111]
11	Role in Lymphedema:	VEGFR3 signaling is implicated in the pathogenesis of lymphedema, providing potential therapeutic targets for its treatment.	[112]
12	Developmental Disorders	Mutations in VEGFR3 are associated with primary lymphedema and other developmental disorders affecting the lymphatic system.	[113]

**Table 4 cimb-47-00851-t004:** VEGFR pathway latest developments.

#	Property	Description	References
1	Targeting Alternative Isoforms	Alternative VEGF and VEGFR isoforms and cancer and angiogenesis are being studied more. VEGF-A isoforms like VEGF-A165b have been studied for their effects on vascular function and tumor growth.	[128]
2	Therapeutic Resistance Mechanisms	Cancer patients’ resistance to VEGF–VEGFR-targeted anti-angiogenic treatments is attracting attention. Current study suggests activating alternate pro-angiogenic pathways and adapting tumor cells and the microenvironment.	[129]
3	Development of Novel Therapeutics	New VEGF–VEGFR pathway therapies include monoclonal antibodies, small-molecule inhibitors, and gene-based methods are being developed. Combination therapies—targeting various angiogenic pathway components or mixing anti-angiogenic drugs with other treatments—are also being studied to improve efficacy and overcome resistance.	[130]
4	Role of VEGFRs in Non-Canonical Signaling	Recent studies have revealed that VEGFRs participate in non-canonical signaling pathways extending beyond angiogenesis, including roles in immune modulation, neuroprotection, and metabolic regulation. A deeper understanding of these non-angiogenic functions could open new avenues for therapeutic strategies across a range of diseases.	[131]
5	Emerging Biomarkers	VEGF–VEGFR pathway biomarkers are being studied for cancer and other illness prognoses. VEGF concentrations, VEGFR expression patterns, and VEGF/VEGFR gene variants may assist guide treatment and predict patient outcomes.	[20]
6	Role in Neurovascular Diseases	The VEGF–VEGFR pathway has been increasingly implicated in neurovascular disorders such as stroke, Alzheimer’s disease, and diabetic retinopathy. Ongoing research is exploring how VEGF signaling influences neurovascular function and assessing its potential as a therapeutic target in these conditions.	[132]
7	Engineering VEGF Mimetics	Researchers are developing VEGF mimetics and engineered VEGF variants designed to provide improved pharmacokinetics and reduced toxicity. These synthetic ligands aim to enhance the efficacy of VEGF-based therapies while minimizing adverse effects.	[133]
8	Role of VEGFRs in Immune Modulation	Recent studies have demonstrated that VEGF receptors (VEGFRs) play a role in modulating immune responses, particularly within the tumor microenvironment. VEGFR signaling has been shown to influence the expression of inhibitory checkpoints on CD8+ T cells, suggesting a close interplay between angiogenic regulation and immune control in cancer.	[134]
9	Exploring Antiangiogenic Therapies in Combination with Immunotherapy	There is increasing interest in combining anti-angiogenic therapies targeting the VEGF–VEGFR pathway with immunotherapeutic approaches in cancer treatment. Both preclinical and clinical studies have reported encouraging outcomes, underscoring the potential synergistic benefit of simultaneously targeting angiogenesis and immune checkpoints.	[135]
10	Role of VEGF-VEGFR Signaling in Metabolic Regulation:	Recent studies have revealed that VEGF–VEGFR signaling extends beyond angiogenesis to the regulation of cellular metabolism. In endothelial and other cell types, VEGFR signaling influences key metabolic processes, pointing to important implications for metabolic diseases and potential therapeutic strategies.	[136]
11	Therapeutic Targeting of VEGF-VEGFR Pathway in Neurodegenerative Diseases	Recent studies have demonstrated that VEGF–VEGFR signaling extends beyond its classical role in angiogenesis to include the regulation of cellular metabolism. Through VEGFR signaling, endothelial and other cell types can modulate key metabolic processes, suggesting potential implications for metabolic disorders as well as opportunities for novel therapeutic interventions.	[137]
12	Role of VEGF-VEGFR Signaling in Organ Development and Regeneration:	Research has shown that VEGF–VEGFR signaling is critical for both organ development and regeneration. Endothelial-derived endocrine signals mediated by VEGFRs play a central role in initiating and sustaining regenerative processes, highlighting the therapeutic potential of this pathway in tissue engineering and regenerative medicine.	[138]
13	Mechanisms of VEGF-VEGFR Axis in Cancer Metastasis:	Recent studies have highlighted the role of the VEGF–VEGFR pathway in cancer metastasis. Beyond its classical function in angiogenesis, VEGF–VEGFR signaling can directly influence tumor cells and the tumor microenvironment, facilitating tumor cell dissemination and metastatic progression.	[139]
14	Exploring VEGF-VEGFR Signaling in Tissue Engineering and Regenerative Medicine:	To enhance vascularization and tissue repair, research in tissue engineering and regenerative medicine has extensively investigated the VEGF–VEGFR pathway. Both VEGF-based therapeutics and engineered biomaterial constructs have been explored as strategies to improve vascular integration and functional outcomes in regenerative applications.	[140]
15	Role of VEGF-VEGFR Pathway in Age-Related Macular Degeneration (AMD	The VEGF–VEGFR pathway is a key contributor to the pathogenesis of age-related macular degeneration (AMD), one of the leading causes of vision loss in the elderly. The introduction of anti-VEGF therapies has transformed AMD management by suppressing abnormal neovascularization and helping to preserve visual function.	[141]

**Table 5 cimb-47-00851-t005:** VEGF mimetic peptides.

#	Name of VEGF Mimetic Peptide	Aa Sequence	Reference
1	VEGF-Mimetic Peptide (CBO-P11):	CGGSNH2	[159]
2	VEGF-Mimetic Peptide VEGF-A (86–92)	YKHKGFFQ	[160]
3	VEGF-Mimetic Peptide Vintafolide (EC145)	Ac-SGGR-amino deoxyglucose-folic acid	[161]
4	VEGF-Mimetic Peptide QK-B:	QK-B	[166]
5	VEGF-Mimetic Peptide QK-F11:	QK-F11	[167]
6	VEGF-Mimetic Peptide (YP15):	YP15	[168]
7	VEGF-Mimetic Peptide (AV-3):	EELRYYNKNR	[164]
8	Vascular Endothelial Growth Factor Peptide (VEGF-31):	TNPNRKTKGKE	[169]
9	VEGF-Mimetic Peptide (ZGDHu-1):	YPDKHLRGD	[170]
10	VEGF-Mimetic Peptide (VGX-1000):	YTRKYKFKIR	[171]
11	VEGF-Mimetic Peptide (LXY30):	LTTSHLLYHLNTKHCFYGG	[165]
12	VEGF-Mimetic Peptide (PRWTEKT)	PRWTEKT	[172]
13	VEGF-Mimetic Peptide (C7):	C7	[173]
14	VEGF-Mimetic Peptide (ZG29)	AGKHLMFGYWKERGRKG	[174]
15	VEGF-Mimetic Peptide (V1):	CTTGRTPR	[175]
16	VEGF-Mimetic Peptide (MF1):	MFYSYFPSD	[176]
17	VEGF-Mimetic Peptide (YLL3):	YLLDVDTKVTP	[177]
18	VEGF-Mimetic Peptide (YLL9)	YLLGLVITGT	[178]
19	VEGF-Mimetic Peptide (RGD-4C)	CRRETAWAC	[179]
20	VEGF-Mimetic Peptide (UPARANT):	AE105-NH2	[180]

**Table 6 cimb-47-00851-t006:** Angiogenesis stimulating molecules.

#	Name	Description	Reference
1	Vascular Endothelial Growth Factor (VEGF)	The introduction of the VEGF gene aims to stimulate the production of vascular endothelial growth factor, a key factor in angiogenesis.	[181]
2	Fibroblast Growth Factor (FGF)	FGFs, particularly FGF-2, are involved in angiogenesis. Gene therapy delivering FGF genes can enhance blood vessel formation.	[182]
3	Hypoxia-Inducible Factor-1 (HIF-1)	HIF-1 is a transcription factor that regulates responses to low oxygen levels (hypoxia). HIF-1 gene therapy aims to induce angiogenesis under hypoxic conditions.	[183]
4	Platelet-Derived Growth Factor (PDGF)	PDGF plays a role in cell growth and division, including vascular smooth muscle cells. Gene therapy with PDGF aims to promote vessel formation.	[184]
5	Angiopoietin-1 (Ang-1) Gene Therapy	Ang-1 is involved in stabilizing blood vessels. Gene therapy with Ang-1 aims to enhance vessel maturation and stability	[177]
6	Hepatocyte Growth Factor (H.G.F.)	H.G.F. is known for its angiogenic and tissue regeneration properties. Gene therapy with H.G.F. may promote angiogenesis	[185]
7	hymosin Beta-4 (Tβ4)	Tβ4 is a peptide involved in cell migration, angiogenesis, and tissue repair. Gene therapy with Tβ4 may enhance these processes.	[186]
8	Stromal Cell-Derived Factor-1 (SDF-1)	SDF-1 is involved in recruiting stem cells and promoting angiogenesis. Gene therapy with SDF-1 aims to enhance tissue repair.	[187]
9	Granulocyte-Colony Stimulating Factor (G-CSF)	G-CSF stimulates the production of granulocytes and stem cells and has been explored for its angiogenic potential	[188]
10	Notch-1 Gene	Notch signaling is involved in vascular development. Gene therapy targeting Notch-1 may influence angiogenesis.	[189]

**Table 7 cimb-47-00851-t007:** VEGF-VEGFR signaling and cellular chirality.

#	Property	Description	References
1	VEGF and Directional Migration:	To create directed blood arteries, VEGF gradients induce endothelial cell migration via VEGFR signaling. Chiral directed movement indicates bias or rotation. Under VEGF gradients, intrinsically chiral cells move or align. VEGF signaling and chiral cues regulate cell migration, which shapes vascular architecture throughout development and wound healing.	[218]
2	VEGFR and Actin Cytoskeleton in Chiral Migration:	Actin cytoskeleton, which maintains cell polarity and chirality, is influenced by VEGFR activity. One-directional lamellipodia or filopodia improve asymmetric cell migration via VEGF-driven actin rearrangement. In certain studies, VEGF promotes endothelial cell chirality, coordinating polarity and boosting chiral migration during angiogenesis.	[161]
3	Cell Chirality and Tissue-Level Organization:	VEGF/VEGFR signaling coordinates tissue topologies via regulating cellular chirality. Chiral migration by VEGF/VEGFR helps endothelial cells generate spiraling or coiling structures for vascular patterning and function. Organ asymmetry and blood circulation require precise matching of vascular and cellular chirality during heart and brain development. VEGF/VEGFR signaling coordinates tissue topologies via regulating cellular chirality. Chiral migration by VEGF/VEGFR helps endothelial cells generate spiraling or coiling structures for vascular patterning and function. During cardiac and brain development, vascular and cellular chirality must be balanced for organ asymmetry and blood circulation.	[228]
4	VEGF in Developmental Left-Right Asymmetry:	According to developmental biology studies, VEGF signaling may interact with pathways that cause left-right (LR) asymmetry in embryonic development, where cellular chirality is critical. VEGF/VEGFR expression by mesodermal and endothelial cells during organ development causes the asymmetric architecture of organs like the heart and lungs, which require coordinated cellular chirality and directed vascularization.	[229]
5	Cytoskeletal Dynamics and VEGFR Interactions	VEGFR activation promotes lamellipodia and filopodia production by reorganizing the cytoskeleton. These structures are necessary for directed cell migration and can move chirally. Cytoskeletal remodeling by VEGF/VEGFR and downstream effectors such Rho GTPases helps cells migrate in certain directions during angiogenesis by chirality.	[164]
6	VEGF Gradient Formation and Directional Chirality	LAMP and filopodia production is generally promoted by VEGFR activation, which reorganizes the cytoskeleton. They are necessary for directed cell migration and can migrate chirally. VEGF/VEGFR and downstream effectors like Rho GTPases govern cytoskeletal reorganization to help cells migrate in certain directions during angiogenesis.	[165]
7	VEGF-Mediated Chiral Cell Polarity and Planar Cell Polarity (PCP) Pathway	Planar cell polarity (PCP) governs tissue cell orientation and alignment, and VEGF signaling interacts with it. PCP components assist endothelial cells produce coordinated chiral patterns in blood vessel development. This route orients cells appropriately in response to VEGF stimulation, encouraging polar cells.	[230]
8	Interaction with Integrins for Coordinated Chiral Migration	Cell alignment and polarity are improved by integrins and VEGFRs strengthening cell adherence to the ECM. Integrin-VEGFR crosstalk enhances cell-ECM connections needed for chiral migratory patterns during angiogenesis, especially where cells need stable adhesion to travel directionally.	[168]
9	Role of Mechanical Forces and Shear Stress on VEGF Signaling and Chirality	Blood flow-induced shear stress affects VEGF/VEGFR expression and chiral endothelial cell alignment. Mechanotransduction favors chiral polarity and migration in vascular tissues, where cells respond to fluid dynamics.	[171]

**Table 8 cimb-47-00851-t008:** Angiogenesis Shaping Using Cellular/Tissular Indices.

#	Cellular Indices	Tissue Indices
1	Endothelial Cell Density: Measures the number of endothelial cells in a given area. Higher density typically indicates active angiogenesis.	Oxygen Tension (pO2): Hypoxia is a potent stimulus for angiogenesis. Measuring tissue oxygen levels can guide interventions in ischemic tissues.
2	Proliferation Markers: Proteins like Ki-67 or PCNA can indicate cell proliferation rates. Increased expression correlates with angiogenic activity.	pH Levels: The acidity of the microenvironment can influence cellular behavior and angiogenic responses. Analyzing tissue pH can help in understanding disease states.
3	Migration Assays: Evaluating the ability of endothelial cells to migrate towards a gradient of angiogenic factors helps understand their responsiveness to stimuli.	Extracellular Matrix (ECM) Composition: The types and organization of ECM components (like collagen and fibronectin) affect angiogenesis. Changes in ECM composition can be indicative of disease progression.
4	Tube Formation Assays: Assessing the ability of endothelial cells to form capillary-like structures in vitro is a direct measure of angiogenic potential.	Vascular Density: Quantifying the number of blood vessels per unit area in a tissue sample provides a direct measure of angiogenic activity.
5	Gene Expression Profiles: Analyzing the expression of genes involved in angiogenesis (like VEGF, FGF) provides insight into cellular responses to various conditions.	Inflammatory Markers: Assessing levels of pro-inflammatory cytokines (like IL-1, TNF-α) can give insight into the tissue’s angiogenic response, as inflammation often drives angiogenesis.
6	Endothelial Cell Senescence Markers: Markers like p16INK4a and telomerase activity can indicate the aging status of endothelial cells, influencing their angiogenic potential.	Mechanical Properties: The stiffness or elasticity of tissue can influence angiogenesis. Stiffer matrices may promote vascularization, while softer ones may inhibit it.
7	Endothelial Cell Senescence Markers: Markers like p16INK4a and telomerase activity can indicate the aging status of endothelial cells, influencing their angiogenic potential.	Vascular Endothelial Growth Factor (VEGF) Levels: Measuring VEGF concentrations in tissues can indicate angiogenic activity, as it is a key regulator of blood vessel formation.
8	Angiogenic Factor Expression: Levels of factors such as VEGF, FGF, and angiopoietins are crucial for assessing the pro-angiogenic state of cells.	Vascular Density (VD): Quantifying the number of blood vessels per unit area in a tissue sample, often assessed through histological techniques.
9	Adhesion Molecule Expression: The presence of molecules like ICAM-1, VCAM-1, and E-selectin on endothelial cells can indicate their readiness to interact with leukocytes and other cells, influencing angiogenesis.	Microvessel Density (MVD): A specific measure of the density of small blood vessels, typically used in tumor studies to assess angiogenesis.
10	Signal Transduction Pathway Activity: Assessing the activation of pathways like PI3K/Akt, MAPK/ERK, and Notch signaling can provide insights into the cellular mechanisms driving angiogenesis.	Fibrosis Index: Evaluating the extent of fibrosis in tissues can indicate chronic conditions that may affect angiogenesis, as fibrotic tissues may have altered blood supply.
11	Nitric Oxide Production: Measurement of nitric oxide levels can indicate endothelial cell function, as it plays a key role in vascular relaxation and angiogenesis.	Lactate Levels: Elevated lactate in tissues can indicate anaerobic metabolism due to insufficient blood supply, which can influence angiogenic processes.
12	Migration and Invasion Assays: Evaluating the ability of endothelial cells to migrate and invade through a Matrigel matrix can indicate their potential for angiogenesis.	Histological Scoring of Inflammation: Assessing the presence of inflammatory cells and associated markers can help gauge the inflammatory microenvironment that often drives angiogenesis.
13	Expression of Matrix Metalloproteinases (MMPs): The expression levels of MMPs (like MMP-2 and MMP-9) are critical for assessing the ability of cells to remodel the extracellular matrix during angiogenesis.	Proteoglycan and Glycosaminoglycan Levels: The presence and composition of these ECM components can influence endothelial cell behavior and angiogenesis.
14	Cytokine Production: The levels of pro-inflammatory cytokines (like IL-6 and IL-8) produced by endothelial cells can influence angiogenesis and the recruitment of other cells.	Collagen Organization: Analyzing the alignment and type of collagen in the extracellular matrix can provide insights into the structural support for angiogenesis.
15	Apoptotic Markers: Assessing markers of apoptosis (like caspases or Annexin V) can help determine the survival of endothelial cells in the context of angiogenesis.	Tissue Growth Factor Levels: The levels of transforming growth factor-beta (TGF-β) and other growth factors can influence angiogenic processes and tissue remodeling.
16	Staining for Endothelial Cell Markers: Immunostaining for specific markers such as CD31 or VE-cadherin can confirm endothelial cell identity and help quantify their presence.	Mechanical Stiffness: Measuring the stiffness of tissues can provide information on how mechanical properties influence endothelial cell behavior and angiogenesis.

## Data Availability

Not applicable. No new data were created or analyzed in this study.

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
