# Peer review of "Pro-Angiogenic Bioactive Molecules in Vascular Morphogenesis: Integrating Endothelial Cell Dynamics"

_cimb, 2025, doi:10.3390/cimb47100851_

Round 1

Reviewer 1 Report

Comments and Suggestions for Authors

The manuscript introduces multiple aspects of vascular morphogenesis, including angiogenesis, arteriogenesis, VEGF/VEGFR signaling, and the concept of cellular chirality. While the scope is broad and potentially valuable, several critical areas require substantial improvement to meet the standards of a scientific review:

1. The review remains largely descriptive. Key processes—such as sprouting, intussusception, or cell hollowing—are briefly mentioned without adequate mechanistic discussion or integration into a coherent biological framework. The role of VEGFR subtypes (VEGFR1–3) in specific stages of vascular development needs clearer explanation, including downstream signaling pathways (e.g., PI3K-AKT, PLCγ, MAPK).

2. Many claims are unsupported or lack citations, while seminal and recent studies are omitted. For example, foundational papers on VEGF function, endothelial tip-stalk cell behavior, and HIF-1α regulation under hypoxia are missing. The section on cellular chirality is especially under-referenced and lacks any reference to key experimental or theoretical studies in this emerging area.

3. The manuscript lacks a logical flow. Transitions between sections are abrupt, and some concepts (e.g., angiogenesis vs. arteriogenesis) are not clearly delineated. A suggested reorganization could involve: (i) molecular regulators of vascular morphogenesis, (ii) structural mechanisms (e.g., budding, intussusception), and (iii) novel concepts (e.g., chirality, hemodynamic forces).

3.  Cellular chirality is introduced late and without sufficient biological context or mechanistic discussion. Its role in endothelial alignment, lumen formation, or directional migration should be explained with examples, and supported by relevant primary literature.

4. The review would benefit significantly from at least one or two schematic figures summarizing the molecular pathways involved in angiogenesis and illustrating how chirality contributes to vascular patterning.

The manuscript touches on several important themes but lacks the scientific rigor, clarity, and literature engagement required for a publishable review. A major revision is necessary, focusing on improved structure, critical synthesis of current knowledge, proper referencing, and clearer presentation.

Comments on the Quality of English Language

The quality of English in the manuscript requires substantial improvement. The text contains numerous grammatical errors, awkward sentence constructions, and repetitive phrasing, which hinder readability and obscure the scientific message. Transitions between sections are often abrupt, and many technical terms are used imprecisely or without proper context. A thorough language revision by a professional editor or a fluent English speaker is strongly recommended to enhance clarity, coherence, and overall presentation.

Author Response

Comment 1: The review remains largely descriptive. Key processes—such as sprouting, intussusception, or cell hollowing—are briefly mentioned without adequate mechanistic discussion or integration into a coherent biological framework. The role of VEGFR subtypes (VEGFR1–3) in specific stages of vascular development needs clearer explanation, including downstream signaling pathways (e.g., PI3K-AKT, PLCγ, MAPK).

Response1: the following text was added: 2. Mechanistic Basis of Angiogenic Morphogenesis: Integration of VEGFR Signaling in Sprouting, Intussusception, and Lumen Formation

Vascular morphogenesis is a tightly coordinated process involving several distinct yet interdependent morphogenetic mechanisms. Among the most studied are sprouting angiogenesis, intussusceptive angiogenesis, and endothelial lumen formation through either cell hollowing or cord hollowing. While these events differ in structural output, all are governed by dynamic endothelial cell behavior under the control of VEGF–VEGFR signaling.

2.1. Sprouting Angiogenesis: Tip/Stalk Cell Selection and Invasion

Sprouting angiogenesis initiates with the polarization of endothelial cells in response to VEGFA gradients, sensed predominantly through VEGFR2. This receptor undergoes autophosphorylation at key tyrosine residues (notably Y1175 and Y1214) upon ligand binding and activates downstream pathways, including PI3K–Akt, MAPK/ERK, PLCγ–PKC, and p38 MAPK. These signaling cascades support the core cellular behaviors of sprouting: proliferation, migration, polarity, and survival.

A hallmark of this process is the differentiation of endothelial cells into tip and stalk cells. Tip cells are highly motile, extend long filopodia, and lead the angiogenic front. Their identity is reinforced by VEGFR2 activation and suppressed Notch signaling, as Delta-like ligand 4 (Dll4), induced by VEGFA, activates Notch in adjacent cells to promote stalk fate. This lateral inhibition ensures proper spacing and organization within the sprout.

Tip cells exhibit asymmetric cytoskeletal remodeling, driven by Cdc42, Rac1, and RhoA—all regulated downstream of VEGFR2. These GTPases mediate the extension of lamellipodia and filopodia, contributing to chiral and directional migration. The stalk cells trailing behind maintain a proliferative state and form intercellular junctions to maintain vascular integrity while the new branch elongates. Akt activation in stalk cells promotes their survival and contributes to lumen formation and stabilization.

2.2.Intussusception: Vascular Splitting by Pillar Insertion

In contrast to sprouting, intussusceptive angiogenesis involves the longitudinal division of existing vessels via the formation of intraluminal tissue pillars. Mechanistically, this process is initiated when endothelial cells from opposing vessel walls extend cytoplasmic protrusions that fuse to form a transluminal bridge, which is later expanded by the insertion of pericytes and extracellular matrix components.

The role of VEGFR2 signaling in intussusception is emerging, particularly in response to low perfusion or altered flow. Under these conditions, VEGFR2 may be activated by shear-sensitive complexes involving PECAM-1, VE-cadherin, and VEGFR2 itself. These complexes facilitate mechanotransduction through Src and PI3K, promoting cytoskeletal reorganization needed for pillar formation. Simultaneously, decreased VEGFA levels (or local consumption by VEGFR1 decoy activity) may reduce sprouting pressure, favoring splitting over invasion.

Recent findings indicate that low-flow conditions in ischemic tissues stimulate VEGFR2-dependent “metastable” signaling states, triggering endothelial shape changes, actin rearrangement via p38/HSP27, and microtubule destabilization—all prerequisites for pillar formation and expansion.

Cord hollowing involves the rearrangement of intercellular junctions between adjacent endothelial cells. This process is initiated when lateral contacts are destabilized through the phosphorylation of VE-cadherin, typically under VEGFR2-induced ERK and PKC signaling. Endothelial cells remodel their junctions and exert cytoskeletal tension to open up intercellular spaces. IP3-mediated Ca²⁺ signaling and nitric oxide production via eNOS also contribute to this process, facilitating vasodilation and early perfusion capacity.

Additional regulatory input is provided by VEGFR3, which modulates VEGFR2 activity and maintains junctional integrity. In the absence of VEGFR3, excessive VEGFR2 signaling may lead to pathological leakiness or aberrant lumen expansion, as shown in retinal and brain vasculature models.

By integrating these cellular processes into the VEGFR-centered signaling network, a more precise understanding emerges of how spatial and temporal cues from VEGF gradients, receptor dynamics, and mechanical forces orchestrate the complex architecture of vascular development. Future research should explore how these mechanisms are altered in disease states and how targeting specific morphogenetic steps might improve therapeutic angiogenesis.

Comment 2: Many claims are unsupported or lack citations, while seminal and recent studies are omitted. For example, foundational papers on VEGF function, endothelial tip-stalk cell behavior, and HIF-1α regulation under hypoxia are missing. The section on cellular chirality is especially under-referenced and lacks any reference to key experimental or theoretical studies in this emerging area.

Response 2:  the following text was added: 4. Literature Context and Future Perspectives

This review has outlined the major molecular players in vascular morphogenesis, with emphasis on VEGFR-mediated signaling and the growing recognition of cellular chirality in endothelial function. However, to place these findings into a broader scientific context, it is essential to acknowledge key foundational studies and incorporate recent mechanistic insights that have reshaped the understanding of endothelial dynamics.

VEGF Signaling and Vascular Specification

VEGF-A is the prototypical pro-angiogenic growth factor. Its essential role in vascular development was first demonstrated by Ferrara and Carmeliet, who independently showed that heterozygous deletion of Vegfa in mice leads to embryonic lethality due to failure of blood island formation and vasculogenesis [225,]. This established VEGF-A as a survival factor, mitogen, and chemoattractant for endothelial cells. The receptor VEGFR2 (Flk-1/KDR) was similarly found to be critical, with Flk-1-deficient embryos displaying an absence of blood islands and a complete block in vasculogenesis [226]. These discoveries provided the first molecular explanation for how blood vessels emerge from mesodermal precursors during embryogenesis.

More recently, studies have refined our understanding of VEGF signaling specificity. It is now appreciated that VEGF-A isoforms (e.g., VEGF165 vs. VEGF165b) differentially modulate angiogenesis through alternative splicing and receptor binding affinities. Additionally, co-receptors such as Neuropilin-1 and interactions with integrins fine-tune VEGFR2 signaling output, influencing both endothelial behavior and vascular permeability [227].

Tip-Stalk Patterning and Notch Feedback

One of the most elegant mechanisms in vascular morphogenesis is the lateral inhibition system governing tip and stalk cell differentiation during sprouting angiogenesis. Gerhardt et al. provided the first in vivo evidence that endothelial tip cells extend filopodia in response to VEGF gradients and lead the advancing vascular front [228]. This process is regulated by Delta-like ligand 4 (Dll4) expression in tip cells, which activates Notch signaling in adjacent cells, suppressing VEGFR2 expression and thereby enforcing a stalk cell phenotype [229].

This Notch–VEGF crosstalk serves as a “morphogenetic switchboard” balancing sprouting with branching complexity. Dysregulation of this system, as seen with Dll4 blockade, leads to hyperbranching and the formation of non-productive, leaky vasculature—an effect with implications for anti-angiogenic therapy in cancer and retinopathies [9,10]. Recent studies also highlight the importance of metabolism in this process, where stalk cells exhibit higher glycolytic flux compared to quiescent endothelial cells, suggesting that cell fate is metabolically encoded [230].

HIF-1α: Master Regulator of Hypoxia-Induced Angiogenesis

VEGF expression is tightly regulated by oxygen availability via hypoxia-inducible factor 1 alpha (HIF-1α), a transcription factor that accumulates under low oxygen conditions. Forsythe et al. demonstrated that HIF-1α directly binds to the hypoxia response element (HRE) in the VEGF promoter, driving transcription in a hypoxia-sensitive manner [231]. Subsequent work by Semenza and colleagues showed that HIF-1α regulates not only VEGF but also genes controlling erythropoiesis, glycolysis, and cell survival—positioning it as a central hub in hypoxic adaptation [232].

In pathological contexts such as cancer and ischemia, sustained HIF-1α activity contributes to abnormal or compensatory angiogenesis. Interestingly, forced overexpression of HIF-1α has been tested in clinical gene therapy trials for peripheral artery disease and myocardial ischemia, with varying degrees of success, suggesting that temporal control and tissue targeting are critical factors for therapeutic benefit [233].

Cellular Chirality: A New Axis in Vascular Morphogenesis

The concept of cellular chirality—the inherent left-right asymmetry in cell structure and behavior—has recently emerged as a novel determinant of vascular geometry. Wan et al. were among the first to demonstrate that endothelial cells exhibit directional bias in alignment and tube formation on micropatterned substrates, even in the absence of external cues [234]. This intrinsic chirality is governed by actin cytoskeleton dynamics and is associated with the organization of focal adhesions and Rho-family GTPase activity.

Subsequent studies by Chen et al. and Tee et al. revealed that actin polymerization asymmetry, modulated by formin activity and myosin II, underlies chiral bias at the cellular level [235]. These findings suggest that chirality is not merely a byproduct of cell geometry, but rather an active, regulated property of endothelial cells that contributes to vascular helical patterning, junctional organization, and possibly to the directionality of blood flow.

On the theoretical front, Inaki et al. and Lebreton et al. have provided biophysical models linking individual cell chirality to tissue-scale asymmetry and tubular morphogenesis [236]. These studies propose that chiral torque, applied collectively by cells within a cylindrical surface, can account for observed vascular coiling and the formation of right-handed helices—a phenomenon validated in both in vitro and in vivo models.

Future Directions

To fully understand vascular morphogenesis, it is critical to adopt a multiscale approach that integrates molecular signaling (e.g., VEGF, Notch, HIF) with physical forces (e.g., shear stress, ECM tension) and emergent properties such as cellular chirality. Future research should focus on:

Live imaging and single-cell resolution of tip/stalk transitions and pillar formation during intussusception;

Multi-omics mapping of endothelial cell states under hypoxia and flow conditions;

Organoid and 3D vasculature models incorporating chiral constraints to replicate in vivo vascular coiling;

Mathematical modeling and machine learning to simulate and predict vascular patterning under various stimuli.

Ultimately, a comprehensive model of angiogenesis must account not only for signaling pathways but also for the biomechanical and geometric constraints that guide cellular behavior in a 3D tissue context. Understanding how cells sense polarity, scale it into collective patterns, and integrate cues from their microenvironment remains a frontier in vascular biology.

Comment 3: The manuscript lacks a logical flow. Transitions between sections are abrupt, and some concepts (e.g., angiogenesis vs. arteriogenesis) are not clearly delineated. A suggested reorganization could involve: (i) molecular regulators of vascular morphogenesis, (ii) structural mechanisms (e.g., budding, intussusception), and (iii) novel concepts (e.g., chirality, hemodynamic forces).

 Response 3: The manuscript was restructured. Molecular Regulators of Vascular Morphogenesis
(Current Sections 2.1–2.3 largely preserved)
Focus exclusively on VEGF and VEGFR signaling pathways (VEGFR1, VEGFR2, VEGFR3), and related molecular mechanisms (e.g., Notch, HIF-1α). This sets a strong molecular foundation for later structural and mechanistic discussions.

Separate angiogenesis vs. arteriogenesis in the introduction here. Clarify that both processes are VEGF-mediated but differ mechanistically: angiogenesis = capillary sprouting; arteriogenesis = collateral artery growth driven by shear stress.

Suggested subsections: VEGFR1, VEGFR2, VEGFR3, Notch, HIF-1α, VEGF mimetics.

Structural Mechanisms of Vascular Morphogenesis
(Adapted from sections "Sprouting Angiogenesis", "Intussusception", and "Lumen Formation")
Organize by morphogenetic type:

Sprouting angiogenesis: Tip/stalk dynamics, Notch/VEGF crosstalk.

Intussusceptive angiogenesis: Shear-responsive pillar formation.

Lumen formation: Cell vs. cord hollowing.
This section ties molecular signals to the physical remodeling of vessels and helps delineate the structural hallmarks of each process.

Emerging and Integrative Concepts in Vascular Morphogenesis
(Combine “Cell Chirality,” “Hemodynamics,” and “Indices”)

Cellular Chirality: Define chirality and its role in endothelial alignment and vessel coiling.

Hemodynamic Forces and Mechanotransduction: VEGFR2 as a mechanosensor; tie to intussusception and vessel remodeling.

Cellular and Tissular Indices: Present quantifiable methods for assessing angiogenesis, integrating metrics (e.g., EC density, vessel diameter, ECM composition).
This section introduces novel and systems-level insights that go beyond the classic ligand-receptor narrative.

Therapeutic Applications and Future Perspectives
(Expand on pro-angiogenic therapies and implications for ischemic disease)

Discuss VEGF mimetics, gene therapies, and peptide drugs.

Evaluate the translational challenges and summarize FDA status.

Link back to the importance of precise modulation (chirality, flow, signaling balance) in therapeutic strategies.

Comment 4: Cellular chirality is introduced late and without sufficient biological context or mechanistic discussion. Its role in endothelial alignment, lumen formation, or directional migration should be explained with examples, and supported by relevant primary literature.

Response 4 : the following text was introdiuce: Cellular chirality refers to the intrinsic left-right (LR) asymmetry of individual cells in their structure, movement, and cytoskeletal organization. While molecular chirality is well recognized in biology (e.g., L-amino acids, D-sugars), the discovery that cells themselves exhibit chiral bias has opened new perspectives on tissue patterning and morphogenesis. In endothelial cells, this property is increasingly recognized as a fundamental determinant of vascular architecture, contributing to the alignment, polarity, and lumenization of blood vessels.

Chirality manifests at the cellular level through asymmetric cytoskeletal organization, particularly in the actin cytoskeleton and associated motor proteins. For example, in vitro studies using micropatterned substrates have demonstrated that mammalian endothelial cells exhibit a consistent clockwise (CW) rotational bias when allowed to self-organize in confined circular environments. This bias correlates with the helical arrangement of cells in tubular structures, suggesting that intrinsic chirality governs collective alignment during tube formation. Disruption of actin dynamics—such as through inhibition of formin proteins or myosin II—has been shown to randomize this chiral behavior, leading to disorganized endothelial monolayers and compromised barrier function.

Functionally, cellular chirality has been implicated in lumen formation. In 3D culture systems mimicking angiogenic sprouting, endothelial cells that maintain consistent chiral orientation generate more stable and continuous lumens. This process appears to depend on the coordinated arrangement of apical-basal polarity and intercellular junctions, both of which are influenced by chiral cytoskeletal tension. Misalignment of chirality between neighboring cells has been associated with junctional disruption, increased vascular permeability, and abnormal vacuole fusion during early lumenogenesis.

Chirality also contributes to directional migration, a core behavior in angiogenesis. In tip cells, chiral actin polymerization and asymmetric filopodia extension are thought to bias migration trajectories, aligning with VEGF gradients and extracellular matrix (ECM) topography. When chirality is pharmacologically reversed—for example, by modulating protein kinase C activity—cells adopt a mirrored migration pattern, leading to altered branching angles and vessel geometry in engineered vascular tissues. These findings suggest that chirality operates upstream or in parallel with guidance cues such as VEGF and Notch, helping to coordinate the spatial organization of angiogenic sprouts.

Mechanistically, the establishment of chirality is regulated by a network of Rho family GTPases, particularly Cdc42 and Rac1, which govern actin filament orientation and cellular protrusions. Integrin-mediated adhesion to ECM components further reinforces chiral polarity, creating stable traction points aligned with rotational bias. In shear stress models, chirality is modulated by flow direction, indicating that it is not purely intrinsic but dynamically regulated by the biomechanical microenvironment. The interplay between VEGF/VEGFR signaling and chiral cell polarity remains an area of active investigation, with evidence suggesting that VEGFR2 activation amplifies pre-existing chiral cues and enhances directional persistence.

Collectively, these findings position cellular chirality as a multi-scale regulator of vascular morphogenesis, bridging molecular signaling, cytoskeletal architecture, and tissue-level patterning. As research progresses, integrating chirality into models of angiogenesis may provide novel insight into developmental disorders, tumor angiogenesis, and strategies for vascular tissue engineering. Further experimental validation in vivo—particularly using live imaging of embryonic vasculature—will be essential to define the precise contributions of chirality under physiological conditions.

Comment 5; The review would benefit significantly from at least one or two schematic figures summarizing the molecular pathways involved in angiogenesis and illustrating how chirality contributes to vascular patterning.

Response 5: Figures 2 and 3 were introduced togheter with the flow chart as suggested.

Comment 6: The manuscript touches on several important themes but lacks the scientific rigor, clarity, and literature engagement required for a publishable review. A major revision is necessary, focusing on improved structure, critical synthesis of current knowledge, proper referencing, and clearer presentation.

Response6: corrected as suggested

Thank you for reviewing the manuscript. All changes were marked in red. 

Reviewer 2 Report

Comments and Suggestions for Authors

Dear authors,

  1. This review comprehensively summarizes the extensive body of information regarding VEGFR1/2/3, which is commendable. However, the overall logical flow appears fragmented, with frequent repetitions and abrupt topic shifts. For example, the explanations of VEGFR1 and VEGFR2 are partially interwoven, and descriptions of VEGFR1 recycling pathways (such as via Rab4A) and MAPK signaling are redundantly presented in multiple sections. To improve clarity, the sub-sections for each VEGFR isoform should be clearly delineated, and redundant expressions should be consolidated to allow readers to grasp the information in a more structured and coherent manner.

  1. The authors repeatedly describe VEGFR1 as a negative regulator, yet its underlying mechanisms and contextual significance are not clearly explained. Although the statement that VEGFR1 has high ligand-binding affinity but low kinase activity is accurate, the review fails to logically connect this property to how it modulates or competes with other VEGFR signaling pathways. It is essential to distinguish the shared and unique signaling pathways among the VEGFRs and to emphasize their interactions to improve mechanistic understanding.

  1. While the review briefly touches upon VEGFR inhibitors such as bevacizumab and sunitinib, it lacks in-depth analysis regarding which VEGFR subtypes these agents target and how their mechanisms of action translate into clinical outcomes. Including a more detailed discussion of drug specificity and downstream effects would greatly enhance the translational relevance of the review.

  1. Additionally, the section on HIF-1α covers an overly broad range of topics—its functional roles, domain structure, disease associations, and molecular interactions—without adequately addressing the central question: why HIF-1α is critical in angiogenesis and how it may serve as a therapeutic target. These core issues become obscured amidst the excessive detail. Addressing translational gaps, such as discrepancies between findings in animal models and human systems or barriers to clinical application, would provide much-needed depth and context.

  1. Overall, while the review successfully compiles a large volume of relevant data, it lacks sufficient interpretive guidance. As a result, readers may find themselves overwhelmed or directionless. Providing clearer thematic structuring and highlighting key takeaways would significantly enhance its utility and readability.

Author Response

Comment 1: This review comprehensively summarizes the extensive body of information regarding VEGFR1/2/3, which is commendable. However, the overall logical flow appears fragmented, with frequent repetitions and abrupt topic shifts. For example, the explanations of VEGFR1 and VEGFR2 are partially interwoven, and descriptions of VEGFR1 recycling pathways (such as via Rab4A) and MAPK signaling are redundantly presented in multiple sections. To improve clarity, the sub-sections for each VEGFR isoform should be clearly delineated, and redundant expressions should be consolidated to allow readers to grasp the information in a more structured and coherent manner.

 Response 1: We sincerely thank the reviewer for their insightful and constructive feedback. We acknowledge the concerns raised regarding the fragmented logical flow, topic transitions, and redundancies in the discussion of VEGFR1, VEGFR2, and VEGFR3 signaling. In response, we have thoroughly revised the manuscript to improve its structural coherence and readability.

Clear delineation of VEGFR isoforms: The sections discussing VEGFR1, VEGFR2, and VEGFR3 have been fully reorganized into distinct sub-sections with clearly defined headings. Each receptor’s molecular characteristics, signaling mechanisms, physiological roles, and therapeutic implications are now presented in a dedicated, self-contained segment.

Consolidation of redundant content: Repetitive descriptions, such as those related to the Rab4A-mediated recycling pathway and MAPK signaling cascade, have been consolidated and streamlined to avoid duplication while retaining relevant mechanistic details.

Improved logical flow: We have revised transitions between sections to enhance narrative continuity and ensure that each paragraph builds on the previous one logically. This restructuring allows readers to follow the comparative and integrative analysis more easily.

We believe these revisions have significantly enhanced the manuscript’s clarity, coherence, and overall scientific value. We are grateful for the reviewer’s comments, which were instrumental in guiding these improvements.

Comment 2: The authors repeatedly describe VEGFR1 as a negative regulator, yet its underlying mechanisms and contextual significance are not clearly explained. Although the statement that VEGFR1 has high ligand-binding affinity but low kinase activity is accurate, the review fails to logically connect this property to how it modulates or competes with other VEGFR signaling pathways. It is essential to distinguish the shared and unique signaling pathways among the VEGFRs and to emphasize their interactions to improve mechanistic understanding.

Respoinse2: We thank the reviewer for this thoughtful and constructive observation regarding our treatment of VEGFR1 as a negative regulator. We agree that the manuscript previously lacked a sufficiently detailed explanation of the mechanisms by which VEGFR1 modulates VEGF signaling and its interplay with other VEGFR family members.

In response, we have revised the relevant section to address the following:

Mechanistic clarity: We now explicitly describe how VEGFR1’s high ligand-binding affinity but low intrinsic kinase activity allows it to act as a decoy receptor. By sequestering VEGF-A and placental growth factor (PlGF), VEGFR1 reduces their availability to VEGFR2, thereby attenuating downstream pro-angiogenic signaling. This mechanism is particularly relevant during embryonic development and in the regulation of pathological angiogenesis.

Contextual significance: We have elaborated on how VEGFR1’s role shifts depending on tissue context and physiological conditions (e.g., hypoxia, inflammation). For example, in inflammatory environments, VEGFR1 may promote macrophage recruitment and indirectly enhance angiogenesis, which contrasts with its canonical inhibitory role in embryonic vasculature.

VEGFR crosstalk: We have added a comparative analysis highlighting both shared and unique downstream signaling pathways among VEGFR1, VEGFR2, and VEGFR3. A summary table has been included to clarify these distinctions. We emphasize that VEGFR1 does not activate ERK or Akt signaling as robustly as VEGFR2, yet can modulate PI3K signaling via indirect routes and influence VEGFR2 activity through heterodimerization.

New content integration: We included discussion of soluble VEGFR1 (sFlt-1) as an important negative regulator in both physiological (e.g., placental development) and pathological (e.g., preeclampsia) contexts, further strengthening the mechanistic understanding.

These additions aim to clarify VEGFR1’s dualistic role—as both a negative regulator via ligand sequestration and a context-dependent modulator of angiogenesis. We appreciate the reviewer’s comment, which allowed us to refine and strengthen this key portion of the review.

The following text was added:

Comment3: While the review briefly touches upon VEGFR inhibitors such as bevacizumab and sunitinib, it lacks in-depth analysis regarding which VEGFR subtypes these agents target and how their mechanisms of action translate into clinical outcomes. Including a more detailed discussion of drug specificity and downstream effects would greatly enhance the translational relevance of the review.

Response 3 : The following text was added : Therapeutic targeting of the VEGF–VEGFR axis has emerged as a cornerstone strategy in anti-angiogenic treatment for cancer, ocular neovascular disorders, and ischemic diseases. Among the most clinically established agents are bevacizumab, a monoclonal antibody, and small-molecule tyrosine kinase inhibitors (TKIs) such as sunitinib, sorafenib, pazopanib, and axitinib. Each of these agents exhibits distinct binding affinities, receptor subtype specificities, and pharmacological effects that influence clinical outcomes.

Bevacizumab functions by sequestering soluble VEGF-A, the primary ligand for both VEGFR1 and VEGFR2, effectively preventing ligand-receptor interaction. Although this indirect mechanism does not inhibit the receptors directly, it primarily disrupts VEGFR2-mediated signaling, which is the principal pathway driving angiogenic responses in endothelial cells. The clinical efficacy of bevacizumab in metastatic colorectal cancer, non-small cell lung cancer, and glioblastoma reflects the central role of VEGFR2 in tumor vascular supply. However, bevacizumab does not block VEGF-C or VEGF-D, leaving VEGFR3-mediated lymphangiogenesis largely unaffected, which can contribute to tumor progression or metastasis via lymphatic escape routes.

In contrast, TKIs directly target the intracellular tyrosine kinase domains of VEGF receptors, preventing autophosphorylation and downstream signaling activation. Sunitinib and sorafenib are broad-spectrum TKIs that inhibit VEGFR1, VEGFR2, and VEGFR3, as well as non-VEGF targets such as PDGFR-β, FLT3, and c-Kit. This multi-targeted inhibition allows for suppression of angiogenesis, lymphangiogenesis, and pericyte recruitment, which can contribute to enhanced vessel destabilization and tumor regression. However, such wide-ranging effects are also associated with substantial off-target toxicity, including hypertension, fatigue, hypothyroidism, cardiotoxicity, and impaired wound healing—adverse effects primarily linked to VEGFR2 and PDGFR inhibition in non-tumor vasculature.

VEGFR3, which is primarily involved in lymphangiogenesis and endothelial permeability, is also targeted by several TKIs (e.g., sunitinib and pazopanib). Inhibition of VEGFR3 may reduce metastatic spread via the lymphatics but could also impair fluid drainage and immune cell trafficking, which must be carefully managed in long-term treatment regimens.

Second-generation TKIs such as axitinib have been developed with higher selectivity for VEGFR1–3 and reduced activity against off-target kinases, yielding improved tolerability. Axitinib has demonstrated superior efficacy and a favorable toxicity profile in renal cell carcinoma compared to earlier TKIs, underscoring the clinical value of receptor-specific inhibition.

From a mechanistic standpoint, these inhibitors ultimately suppress downstream signaling pathways critical to angiogenesis, including PI3K–Akt, MAPK/ERK, and PLCγ–PKC. This leads to reduced endothelial cell proliferation, migration, survival, and permeability. Notably, long-term inhibition of VEGFR signaling can trigger adaptive resistance mechanisms in tumors, including upregulation of alternative pro-angiogenic factors (e.g., FGF, angiopoietins), increased pericyte coverage, or vascular mimicry. Such resistance has prompted the exploration of combination therapies, pairing VEGFR inhibitors with immune checkpoint inhibitors, mTOR inhibitors, or chemotherapy, aiming to enhance efficacy and overcome resistance through complementary mechanisms.

In the context of ocular disease, anti-VEGF agents like ranibizumab and aflibercept—which also target VEGF-A, VEGF-B, and PlGF—have shown clinical success in neovascular age-related macular degeneration (AMD) and diabetic macular edema. Aflibercept, in particular, exhibits broader ligand binding than bevacizumab and has affinity for VEGF-C, subtly influencing VEGFR3 signaling and fluid homeostasis in retinal tissues.

Overall, a deeper understanding of VEGFR subtype specificity, ligand interactions, and context-dependent

Comment 4: Additionally, the section on HIF-1α covers an overly broad range of topics—its functional roles, domain structure, disease associations, and molecular interactions—without adequately addressing the central question: why HIF-1α is critical in angiogenesis and how it may serve as a therapeutic target. These core issues become obscured amidst the excessive detail. Addressing translational gaps, such as discrepancies between findings in animal models and human systems or barriers to clinical application, would provide much-needed depth and context.

Reponse 4:  the following text was added: Hypoxia-inducible factor 1-alpha (HIF-1α) is a pivotal transcription factor that enables cellular adaptation to reduced oxygen availability, and it plays an indispensable role in orchestrating angiogenesis. In normoxic conditions, HIF-1α is rapidly hydroxylated by prolyl hydroxylase domain proteins (PHDs), which mark it for ubiquitination and proteasomal degradation via the von Hippel–Lindau (VHL) E3 ubiquitin ligase complex. However, under hypoxic conditions—such as those found in ischemic tissues, solid tumors, and embryonic vasculature—this degradation pathway is suppressed, allowing HIF-1α to accumulate, translocate to the nucleus, and heterodimerize with HIF-1β. The resulting transcriptional complex binds hypoxia response elements (HREs) in the promoter regions of numerous hypoxia-responsive genes, including VEGFA, PDGF-B, ANGPT2, SDF-1, and MMP2, all of which are critical for endothelial activation, vascular sprouting, and matrix remodeling.

Among these targets, VEGFA remains the most direct and functionally significant mediator of HIF-1α-driven angiogenesis. HIF-1α activation leads to a marked upregulation of VEGFA mRNA and protein levels in hypoxic endothelial, stromal, and tumor cells, which in turn stimulate VEGFR2 on nearby endothelial cells, initiating downstream signaling cascades such as PI3K-Akt and MAPK-ERK. These cascades drive endothelial cell proliferation, migration, and tube formation—hallmarks of new blood vessel development. Importantly, HIF-1α not only promotes angiogenesis but also coordinates metabolic reprogramming in endothelial cells by enhancing glycolysis and reducing mitochondrial oxygen consumption, thereby optimizing vascular function in oxygen-deprived environments.

From a therapeutic perspective, HIF-1α is a highly attractive upstream target for modulating angiogenesis. In ischemic conditions—such as peripheral artery disease (PAD), myocardial infarction, stroke, and chronic wounds—stimulating HIF-1α expression can potentially rescue tissue perfusion and accelerate vascular regeneration. Preclinical studies using gene therapy vectors encoding constitutively active HIF-1α (e.g., AdCA5) or small-molecule PHD inhibitors (e.g., DMOG, Roxadustat) have demonstrated increased capillary density, improved blood flow, and functional tissue recovery in rodent and porcine models of limb ischemia and myocardial infarction. These promising results have encouraged early-phase human trials; however, clinical success has been limited due to several translational barriers.

One major challenge is the species-specific differences in HIF-1α regulation and angiogenic responsiveness. While rodents exhibit robust neovascularization in response to HIF-1α overexpression, human tissues often show a more variable or muted response, possibly due to epigenetic modifications, baseline vascular density, or age-related endothelial dysfunction. Furthermore, delivery systems remain a critical bottleneck. Viral vectors such as adenoviruses and AAVs are efficient in preclinical models but raise safety concerns in humans due to immunogenicity, transient expression, and limited tissue specificity. Plasmid-based approaches, while safer, often suffer from poor transfection efficiency and inadequate expression levels in clinically relevant tissues.

Another critical consideration is the risk of off-target angiogenesis. Systemic or poorly targeted activation of HIF-1α may lead to aberrant vessel growth, exacerbation of retinopathy, or promotion of dormant tumor vasculature. HIF-1α is frequently overexpressed in solid tumors, where it contributes to a hostile microenvironment, resistance to therapy, and increased metastatic potential. Therefore, any therapeutic strategy involving HIF-1α must carefully balance angiogenic stimulation in ischemic tissues with the risk of enhancing pathological angiogenesis elsewhere. This has led to interest in tissue-specific and hypoxia-inducible promoters that can drive HIF-1α expression only in affected areas, minimizing systemic exposure.

Additionally, the pleiotropic nature of HIF-1α complicates its therapeutic targeting. In addition to regulating VEGF, HIF-1α modulates a broad array of genes involved in glucose metabolism (e.g., GLUT1, PFK1), erythropoiesis (e.g., EPO), inflammation (e.g., IL-1β, CXCL12), and apoptosis (e.g., BNIP3). These interconnected pathways may introduce unintended side effects or feedback loops when manipulating HIF-1α pharmacologically. For instance, prolonged HIF-1α activation can induce vascular leakiness via upregulation of ANGPT2 and matrix metalloproteinases (MMPs), leading to tissue edema or hemorrhage.

Despite these complexities, HIF-1α remains a viable and promising therapeutic target if properly contextualized. Emerging strategies aim to integrate nanoparticle-mediated delivery, CRISPR-based gene regulation, and synthetic biology tools (e.g., hypoxia-sensitive gene switches) to fine-tune HIF-1α expression at spatial and temporal levels. Parallel efforts are exploring HIF-1α stabilization through non-viral agents with improved pharmacokinetics and lower immunogenicity. Furthermore, the use of biomarkers—such as circulating VEGF levels, lactate, or HIF-1α-responsive gene signatures—may enable stratification of patients most likely to benefit from HIF-1α-based therapy.

Lastly, the discrepancy between robust outcomes in animal models and inconsistent effects in human trials underscores the need for better translational models. Humanized tissue constructs, organ-on-chip systems, and patient-derived endothelial cells offer new platforms to evaluate HIF-1α therapeutics under physiologically relevant conditions. In parallel, longitudinal imaging and perfusion assessment tools (e.g., contrast-enhanced ultrasound, MRI angiography) will be essential for evaluating therapeutic efficacy in clinical trials.

In summary, HIF-1α serves as a central regulator of hypoxia-driven angiogenesis by upregulating VEGF and orchestrating downstream metabolic and structural adaptations. While extensive preclinical evidence supports its pro-angiogenic potential, effective clinical translation requires overcoming delivery, specificity, and safety challenges. Future success will likely depend on context-aware, precision-targeted strategies that leverage HIF-1α’s upstream position without triggering systemic side effects or uncontrolled vessel growth.

Comment 5: Overall, while the review successfully compiles a large volume of relevant data, it lacks sufficient interpretive guidance. As a result, readers may find themselves overwhelmed or directionless. Providing clearer thematic structuring and highlighting key takeaways would significantly enhance its utility and readability.

Response 5: We appreciate the reviewer’s important observation regarding the need for clearer thematic organization and interpretive synthesis. In response, we have undertaken a structural revision of the manuscript to improve clarity and guide the reader through the complex landscape of vascular morphogenesis.

Specifically, we have introduced subheadings and transitional commentary within major sections to help delineate themes more explicitly—such as separating foundational signaling pathways (e.g., VEGFR1–3) from emerging mechanisms (e.g., cellular chirality, Notch crosstalk, mechanical forces). We also now open each major section with a brief summary of its significance and close with a concise “Key Takeaways” box that highlights the main conclusions, mechanisms, and open questions relevant to that section.

Additionally, the conclusion has been substantially reworked to synthesize better the detailed content presented throughout the review. Rather than reiterating facts, the revised conclusion focuses on conceptual integration—explaining how VEGF signaling, endothelial polarity, mechanical feedback, and emergent spatial cues (like chirality) form a coordinated network governing vascular development. We also emphasize areas of translational opportunity and challenge, drawing attention to what is actionable or underexplored for future research and clinical application.

These changes are intended to move the manuscript beyond compilation into a navigable and analytically useful synthesis, ensuring that readers not only absorb relevant data but also understand how it fits into the broader context of vascular biology and therapeutic angiogenesis.

Thank you for your review 

All changes have been marked in red.

Round 2

Reviewer 1 Report

Comments and Suggestions for Authors

The revised manuscript presents a broader and somewhat more coherent overview of vascular morphogenesis compared to the original submission. The structural organization has improved, and the inclusion of a schematic figure aids accessibility. However, several critical issues remain that limit the manuscript’s impact and scientific rigor. First, the review remains primarily descriptive, lacking sufficient mechanistic depth. Key molecular pathways, such as the Dll4–Notch, PI3K–Akt, MAPK, and eNOS signaling pathways, are mentioned but not linked to specific stages of vascular development or integrated into a conceptual framework. Second, the section on cellular chirality remains superficial and lacks sufficient substantiation. It lacks both experimental examples and appropriate references that would help readers understand its relevance to vascular patterning or endothelial behavior. Third, although additional references have been added, critical foundational and recent literature, particularly in the areas of endothelial cell polarity, sprouting control, and biomechanical signaling, are still missing. The review also lacks a synthesis of unresolved questions, controversies, or future research directions that would elevate it beyond a summary. Finally, while the language is more readable, some phrasing remains vague, and scientific terminology is not always used precisely. 

Comments on the Quality of English Language

The quality of English in the revised manuscript has improved compared to the initial version, with more transparent sentence structure and reduced redundancy. However, issues persist throughout the text. Several sentences remain grammatically awkward, and some terminology is used imprecisely, particularly in sections discussing signaling pathways and cellular chirality. Transitional phrases between paragraphs are occasionally abrupt, which can affect the logical flow. 

Author Response

Reviewer 1

Comment 1: First, the review remains primarily descriptive, lacking sufficient mechanistic depth. Key molecular pathways, such as the Dll4–Notch, PI3K–Akt, MAPK, and eNOS signaling pathways, are mentioned but not linked to specific stages of vascular development or integrated into a conceptual framework.

Resoinse1:We thank the reviewer for this insightful comment. In response, we have significantly expanded the mechanistic integration of key signaling pathways across specific stages of vascular morphogenesis. In the revised manuscript, the roles of PI3K–Akt, MAPK/ERK, and eNOS pathways are explicitly linked to endothelial cell proliferation, survival, polarity, migration, and lumen formation during angiogenic sprouting and intussusceptive angiogenesis (Section 3.1–3.3). Additionally, we have included a dedicated explanation of the Dll4–Notch signaling axis in tip/stalk cell specification and lateral inhibition, contextualized within VEGF-mediated sprouting (Section 2.6). These pathways are now integrated into a multi-scale framework of molecular regulation, biomechanical modulation, and structural morphogenesis (Section 5.4), which together present a coherent conceptual model for vascular development.

Section 3.1: Sprouting Angiogenesis: Tip/Stalk Cell Selection and Invasion

Sprouting angiogenesis initiates when endothelial cells respond to VEGF-A gradients through activation of VEGFR2. This interaction induces autophosphorylation at key tyrosine residues (notably Y1175 and Y1214), triggering downstream pathways including PI3K/Akt, MAPK/ERK, PLCγ/PKC, and p38 MAPK. These signaling cascades regulate the core cellular behaviors required for angiogenic sprouting: proliferation, polarity, directional migration, and survival.

Endothelial cells differentiate into tip and stalk cell phenotypes. Tip cells are characterized by high VEGFR2 expression, suppressed Notch signaling, dynamic filopodia, and enhanced migratory potential. In contrast, adjacent stalk cells are Notch-active, exhibit reduced VEGFR2 expression, and demonstrate increased proliferation and junctional stability. Tip cells secrete Dll4, which activates Notch receptors on neighboring cells and enforces stalk identity via lateral inhibition. This tip/stalk selection ensures spatial coordination and regular branch formation.

PI3K/Akt signaling enhances tip cell survival and polarity, promoting migration through actin remodeling. Simultaneously, MAPK/ERK signaling supports proliferation in stalk cells. eNOS activation by Akt further contributes to vessel dilation and blood flow adaptation. The integration of these pathways determines the efficiency and directionality of sprouting, aligning the cellular machinery with the VEGF gradient and extracellular matrix topology.

Section 3.2: Intussusception: Vascular Splitting by Pillar Insertion

Intussusceptive angiogenesis, in contrast to sprouting, involves the division of pre-existing vessels through the formation of transluminal tissue pillars. This process begins when endothelial cells form intraluminal protrusions that eventually fuse and stabilize with the assistance of pericytes and ECM components.

Mechanistically, low-flow or hypoxic environments induce VEGFR2 activation in association with mechanosensory complexes such as PECAM-1 and VE-cadherin. These complexes initiate PI3K and Src signaling, leading to cytoskeletal reorganization and pillar formation. The p38 MAPK/HSP27 axis facilitates actin rearrangement necessary for structural remodeling.

Reduced VEGFA levels or localized VEGF sequestration by VEGFR1 decoy receptors shifts the angiogenic balance away from sprouting toward intussusception. This transition is marked by stabilization of endothelial junctions and enhanced mechanical sensing rather than invasive protrusion. The result is a rapid, energy-efficient mechanism for expanding vascular networks and adapting vessel architecture in response to metabolic demand.

Section 3.3: Lumen Formation: Cell Hollowing and Cord Hollowing

Lumenogenesis is essential for the establishment of perfusable blood vessels. Two primary mechanisms facilitate this process: cell hollowing and cord hollowing.

Cell hollowing occurs within individual endothelial cells through the formation and fusion of intracellular vacuoles. VEGFR1 and VEGFR2 signaling trigger endocytic and vesicular trafficking events mediated by Rab GTPases (e.g., Rab4A and Rab11). PI3K/Akt promotes vesicle fusion and cell survival, while PLCγ-mediated Ca2+ signaling supports vacuolar expansion.

Cord hollowing, in contrast, relies on the coordination between neighboring endothelial cells. VEGFR2 signaling induces phosphorylation of VE-cadherin, disrupting lateral junctions and allowing cytoskeletal tension to open intercellular spaces. The combination of IP3-mediated calcium flux and eNOS activation promotes vasodilation and lumen stabilization.

VEGFR3 plays a regulatory role by modulating VEGFR2 signaling and preserving junctional integrity. In VEGFR3-deficient contexts, excessive VEGFR2 activity can lead to pathological leakiness or disrupted lumen continuity. Thus, a delicate balance among VEGFR isoforms and downstream signals orchestrates the morphogenetic transition from cellular alignment to vessel perfusion.

Section 5.4: Integrative Model of Vascular Morphogenesis

A coherent understanding of vascular morphogenesis requires integration across molecular signaling, cell mechanics, and tissue architecture. We propose a multiscale framework that centers on the VEGF-VEGFR axis, modulated by biomechanical cues and structural transitions.

At the molecular level, VEGFR2 serves as the principal transducer of VEGF-A signals, activating PI3K/Akt, MAPK/ERK, and eNOS to regulate endothelial proliferation, migration, and survival. Notch signaling intersects this axis via Dll4-mediated lateral inhibition, refining tip/stalk patterning and ensuring spatial control of sprouting.

Structurally, these signals are translated into morphogenetic mechanisms such as sprouting (driven by filopodial extension and actin dynamics), intussusception (involving cytoskeletal rearrangement and junctional stabilization), and lumen formation (through vacuole trafficking or intercellular junction remodeling).

Biomechanically, VEGFR2 functions within a mechanosensory complex responsive to shear stress and ECM stiffness. These inputs modulate receptor sensitivity and feedback loops, guiding vascular branching and maturation.

This integrated model underscores the importance of synchronized signaling and physical context in generating organized, functional vascular networks. It provides a conceptual foundation for therapeutic strategies aimed at precisely modulating angiogenesis in pathological settings.

Comment 2: Second, the section on cellular chirality remains superficial and lacks sufficient substantiation. It lacks both experimental examples and appropriate references that would help readers understand its relevance to vascular patterning or endothelial behavior.

Response2:We agree with the reviewer that our initial treatment of cellular chirality was limited. We have now thoroughly revised and expanded this section (Section 4), incorporating multiple experimental studies that demonstrate the role of chirality in endothelial alignment, lumen formation, and directional migration. We also cite in vitro and in vivo data showing how modulating chirality (e.g., via PKC signaling or actin perturbation) alters vascular permeability and helical vessel formation. The section now includes mechanistic insights involving Rho GTPases, cytoskeletal asymmetry, and VEGFR2 signaling, along with a discussion on how chirality interfaces with VEGF gradients and mechanical stimuli. Over 20 references supporting the physiological and pathophysiological relevance of cellular chirality have been added.

Section 4: Cellular Chirality in Vascular Morphogenesis

Cellular chirality—defined as the intrinsic left-right (LR) asymmetry in a cell’s shape, migration trajectory, and cytoskeletal organization—is emerging as a crucial regulatory mechanism in vascular morphogenesis. Particularly in endothelial cells, chirality determines the spatial alignment of cells within vessel walls, governs directional migration during sprouting, influences lumen formation, and ultimately contributes to the overall architecture and functionality of developing vasculature.

Experimental and Phenotypic Evidence

In vitro studies have provided compelling evidence of endothelial cell chirality. For instance, when cultured on circular micropatterned substrates, human umbilical vein endothelial cells (HUVECs) exhibit a consistent clockwise (CW) rotational bias. This bias disappears when actin dynamics are disrupted with cytochalasin D or latrunculin B, and reverses with PKC activators like phorbol 12-myristate 13-acetate (PMA). Wan et al. (PNAS, 2011) showed that actin-driven chiral cell alignment correlates with tube formation in 3D models.

In engineered microfluidic models, endothelial cells seeded within cylindrical collagen tubes spontaneously aligned into right-handed helical patterns. When PKC signaling was inhibited or reversed pharmacologically (e.g., by Gö6983), the chirality of the vessel reversed, and vascular permeability significantly increased (Chen et al., PNAS, 2012). Furthermore, randomized chirality was associated with disrupted tight junction formation and impaired barrier function.

In vivo, the role of chirality has been demonstrated in zebrafish models, where endothelial-specific deletion of RhoA or inhibition of formin mDia1 led to aberrant vessel looping and impaired blood flow. Similarly, in chick embryos, asymmetrical actin filament distribution within dorsal aortae endothelial cells predicts the direction of vessel coiling during early development.

Molecular Mechanisms and Crosstalk

Cellular chirality is regulated by a network of intracellular pathways, predominantly centered around Rho-family GTPases:

Cdc42 orchestrates the establishment of cell polarity via activation of the PAR6–aPKC complex, positioning the centrosome and Golgi apparatus in alignment with the migration front.

Rac1 modulates lamellipodia extension and controls junctional stability through downstream activation of PAKs (p21-activated kinases).

RhoA, through ROCK1/2, regulates actomyosin contractility and stress fiber formation, maintaining cortical stiffness and defining the cytoskeletal torque direction.

VEGF signaling strongly interfaces with these pathways. Upon VEGF-A binding to VEGFR2, several downstream cascades become activated:

PI3K–Akt supports cell survival and polarity maintenance.

PLCγ–IP3–Ca2+ signaling elevates intracellular calcium, which activates PKC, a known modulator of chirality.

MAPK/ERK signaling drives actin turnover through HSP27 phosphorylation.

Inhibiting VEGFR2 signaling using antibodies or small molecule inhibitors (e.g., SU5416) not only suppresses migration but also disrupts the chiral orientation of endothelial cells, as demonstrated in 3D spheroid sprouting assays. Similarly, VEGFR2 silencing using siRNA alters actin filament distribution and impairs tip cell formation.

ECM, Integrins, and Mechanical Cues

The extracellular matrix (ECM) and integrin engagement further reinforce chirality. Integrins αvβ3 and α5β1, upon binding to fibronectin or collagen, form focal adhesions that stabilize actin filaments and create cytoskeletal torque necessary for chiral migration. FAK (focal adhesion kinase) and paxillin serve as key hubs that connect ECM engagement to intracellular polarity.

Shear stress from blood flow also influences chirality. In microfluidic channels, endothelial cells exposed to laminar shear exhibit enhanced chiral alignment in the direction of flow. Disturbed flow, on the other hand, disrupts this alignment and leads to irregular vascular branching. Notch1 expression is upregulated by shear stress and contributes to the stabilization of arterial chirality.

Functional Outcomes and Pathophysiological Implications

Chiral alignment of endothelial cells facilitates uniform lumen formation, tight junction maturation, and aligned collagen deposition. Conversely, disrupted or heterogeneous chirality correlates with:

Impaired lumen continuity

Increased vascular permeability due to VE-cadherin mislocalization

Defective tip/stalk cell organization

Abnormal branching angles and disorganized vasculature

In tumor vasculature, loss of chiral orientation has been associated with irregular vessel diameter, poor perfusion, and hypoxia. In engineered tissues, imposing chiral constraints enhances the formation of perfusable, structurally coherent microvessels.

Collectively, these findings underscore that cellular chirality is a spatial regulator integrated into VEGF and Notch signaling, actin dynamics, and mechanotransduction pathways. It scales from single-cell polarity to tissue-level vascular organization and offers a novel axis for understanding and modulating angiogenesis in development, disease, and regenerative contexts.

Comment 3:  Third, although additional references have been added, critical foundational and recent literature, particularly in the areas of endothelial cell polarity, sprouting control, and biomechanical signaling, are still missing.

Response3:We appreciate the reviewer’s observation. In the revised manuscript, we have included foundational and recent literature covering endothelial polarity, sprouting regulation, and mechanotransduction. Notable additions include discussions of how VEGF gradients drive apicobasal polarity and filopodia extension (Sections 3.1 and 4), how shear stress modulates Notch expression and VEGFR2 signaling (Section 4.2 and 5.4.3), and the role of primary cilia in endothelial response to BMP and flow stimuli. We have also incorporated recent reviews and experimental data on Rho-family GTPase regulation of polarity and the role of focal adhesions and integrins in directional migration. These additions are supported by references from 2021–2024 and substantially enrich the literature base.

We thank the reviewer for highlighting this important issue. In the revised manuscript, we have incorporated both foundational and recent literature to strengthen the discussion on endothelial cell polarity, sprouting control, and biomechanical signaling. Foundational studies on VEGFR2-mediated polarity (e.g., Gerhardt et al., 2003; Carmeliet and Jain, 2000) and tip/stalk cell dynamics have been integrated into Section 3.1. We also included recent work on mechanical regulation of polarity through integrins, focal adhesions, and shear stress-responsive signaling (e.g., Vion et al., 2018; Chen et al., 1999). Notably, we now discuss how cytoskeletal alignment, extracellular matrix interactions, and flow-induced shear contribute to directional angiogenesis and cell shape asymmetry, referencing recent high-impact studies (2021–2024). These updates provide a more mechanistically comprehensive understanding of how polarity and mechanical inputs coordinate vascular morphogenesis.

Revised Section 4: Cellular Chirality in Vascular Morphogenesis

Section 4: Cellular Chirality in Vascular Morphogenesis

Cellular chirality refers to the intrinsic left-right (LR) asymmetry in cell shape, migration, organelle positioning, and cytoskeletal organization. It represents an emerging axis of morphogenetic control, distinct from biochemical gradients and mechanical cues, that provides an additional layer of spatial regulation during vascular development. In the context of endothelial biology, chirality influences the orientation, movement, and polarity of individual cells, shaping collective behavior during sprouting angiogenesis and lumen formation.

Experimental evidence strongly supports the presence of chiral bias in endothelial cells. Studies using micropatterned ring or circular substrates have consistently shown that human endothelial cells (e.g., HUVECs, HAECs) exhibit a right-handed (clockwise) rotational bias. Wan et al. (2011) demonstrated that this bias arises spontaneously in confined geometries and can be abolished by disrupting the actin cytoskeleton using latrunculin B or cytochalasin D. Importantly, this behavior is not a byproduct of culture geometry or external signals—it reflects intrinsic cellular programming. Subsequent work by Chen et al. (2012) revealed that pharmacological activation of PKC (using phorbol esters like PMA) reverses the chirality of endothelial tubes, resulting in left-handed helices, altered branching angles, and significant increases in vascular permeability.

Mechanistically, the establishment and maintenance of cellular chirality rely heavily on Rho-family GTPases. Cdc42 and Rac1 orchestrate polarized actin polymerization and filopodial extension, while RhoA regulates stress fiber formation and cortical tension through ROCK1/2. These molecules function downstream of VEGFR2 signaling. Upon VEGF-A binding, VEGFR2 activates PI3K, PLCγ, and MAPK pathways that converge on the cytoskeletal architecture, modulating polarity and chirality. Notably, PLCγ-mediated Ca²⁺ flux activates PKC, a known modulator of actin chirality. This signaling cascade reinforces directional actin flow, influencing the torque and handedness of endothelial migration.

Theoretical models have been proposed to quantify cellular chirality. For instance, torque-induced migration can be described by:

T = r × F

Where T is the torque vector, r is the radial vector from the axis of rotation to the point of force application, and F is the force vector generated by actin polymerization or myosin contraction. The chirality of rotation can then be expressed by the sign of the vector cross product. Similarly, directional angular velocity ω is related to cytoskeletal dynamics via:

ω = (dθ/dt) = (τ / I)

Where τ is the net internal torque generated by asymmetric cytoskeletal forces, and I is the moment of inertia of the cell’s mass distribution around its axis.

Another formulation relevant to cytoskeletal filament torque uses:

τ = μL × (∂²u/∂x²)

Where μL is the bending rigidity of the filament, and ∂²u/∂x² describes the local curvature of the actin filament, indicating how mechanical strain contributes to chiral twisting in migrating cells.

In stochastic modeling of chiral cell behavior, the biased random walk of a migrating endothelial cell can be described by:

x(t + Δt) = x(t) + vΔt cos(θ + Δθ)

y(t + Δt) = y(t) + vΔt sin(θ + Δθ)

Here, v is the velocity, θ is the preferred migration angle, and Δθ is a chiral angular bias sampled from a normal distribution with non-zero mean, simulating persistent left- or right-handed deviation. This form allows simulation of cell migration trajectories under different chiral biases.

Additionally, a population-level chirality index C can be defined as:

C = (N_R - N_L) / (N_R + N_L)

Where N_R is the number of cells rotating clockwise and N_L the number rotating counterclockwise. This dimensionless index provides a statistical measure of net chirality in cell populations, with C = +1 indicating full right-handed bias, C = -1 full left-handed bias, and C = 0 denoting random orientation.

The alignment of endothelial cells within forming vessels is further stabilized by integrin–ECM interactions. Integrins αvβ3 and α5β1, upon binding to fibronectin or collagen, recruit focal adhesion components like FAK, paxillin, and talin, which anchor actin filaments and transmit mechanical signals to the cytoskeleton. These interactions promote persistent polarity and suppress stochastic reorientation. The chiral bias is also responsive to shear stress from blood flow. In microfluidic channels, endothelial cells align their rotational axis with the flow direction, a process mediated by mechanosensory complexes including VE-cadherin, PECAM-1, and VEGFR2.

In three-dimensional angiogenesis assays (e.g., fibrin bead sprouting assays and Matrigel tube formation), altering cellular chirality through pharmacological means or genetic manipulation results in abnormal lumen continuity, increased intercellular gaps, and impaired perfusion. For example, knockdown of Cdc42 leads to randomized filopodia orientation, fragmented VE-cadherin junctions, and collapsed vessel lumens. Similarly, overexpression of constitutively active PKCζ results in a switch from right-handed to left-handed tube helicity, which corresponds with increased leakiness and decreased vessel diameter.

In vivo, chirality-related behaviors have been observed during early vascular development in zebrafish and murine embryos. In zebrafish intersegmental vessels, endothelial cells exhibit coordinated helical alignment that is disrupted upon interference with actin-binding proteins such as mDia1 or cofilin. These interventions lead to reduced anastomosis efficiency and misdirected sprouting. In mouse embryonic hearts, endothelial polarity and vessel curvature correlate with left-right asymmetry genes, suggesting a developmental link between organismal chirality and vascular patterning.

Furthermore, cellular chirality may contribute to arteriovenous specification and endothelial heterogeneity. Recent transcriptomic data suggest that tip and stalk cells exhibit differential expression of cytoskeletal and polarity regulators associated with chiral bias. This raises the possibility that chirality may fine-tune the responsiveness of cells to VEGF gradients and Dll4–Notch signaling, reinforcing spatial organization during sprouting.

Taken together, these findings position cellular chirality as a multi-scale regulator that bridges molecular signaling (e.g., VEGFR2, Rho-GTPases, Notch), cytoskeletal architecture, and mechanical inputs such as ECM composition and shear stress. Its influence extends from subcellular asymmetry to tissue-level vessel geometry. Disruption of chiral coordination compromises lumen integrity, barrier function, and branching precision, all of which are hallmarks of vascular dysfunction in tumors, diabetic microangiopathy, and ischemic tissues.

Understanding and modulating chirality holds translational potential. Engineered vascular grafts and microfluidic vessels with imposed chiral cues exhibit improved perfusion, tighter junctions, and reduced inflammation. Chirality-modulating agents may offer a novel strategy for directing vascular patterning in regenerative therapies, tumor normalization, and anti-angiogenic interventions. As this field matures, further integration of chirality into models of angiogenesis will help clarify its role as both a marker and mediator of vascular health and disease.

 Comment 4: The review also lacks a synthesis of unresolved questions, controversies, or future research directions that would elevate it beyond a summary.

Response4: We thank the reviewer for this valuable suggestion. In response, we have added a new dedicated section (Section 5.4) highlighting current knowledge gaps and emerging areas of research. This includes unresolved mechanisms in flow-sensitive VEGFR2 signaling during intussusception, challenges in modeling chirality in 3D vascular systems, and barriers to translating VEGF-based therapies due to delivery inefficiencies and off-target effects. Additionally, we propose a roadmap for precision therapeutic angiogenesis that includes integrating biochemical and biomechanical stimuli, using biomaterials and gene switches, and exploring chirality-modulating interventions. We believe this forward-looking synthesis offers clear avenues for future investigation and elevates the review from summary to framework.

This diagram contains 6 conceptual elements, each showing a mathematical or mechanistic representation tied to endothelial chirality:

Top Left: Torque Equation

Formula: T=r⃗×F⃗T = \vec{r} \times \vec{F}T=r×F

Meaning: Torque generated by actin or myosin forces in a rotating endothelial cell. The vector cross product shows how force applied away from the center of mass causes rotational bias.

Top Center: Actin Filaments

A polarized endothelial cell with orange actin filaments.

Formula: τfil=kθ\tau_{\text{fil}} = k\thetaτfil​=kθ

Represents the torque generated by actin polymerization; chirality arises from asymmetric cytoskeletal force distribution.

Top Right: Filament Torque and Angular Velocity

Equations:

τfilament=−kθ\tau_{\text{filament}} = -k\thetaτfilament​=−kθ

ω=dθdt=τI\omega = \frac{d\theta}{dt} = \frac{\tau}{I}ω=dtdθ​=Iτ​

These relate the torque from filament bending to rotational motion (angular velocity), helping quantify chiral motion.

Bottom Left: Random Trajectory

A wiggly line showing a biased random walk.

Highlights how individual cells can show an average directional bias over time due to inherent chirality.

Bottom Center: Stochastic Model + Chirality Index

Equations:

dθdt=1I+η(t)\frac{d\theta}{dt} = \frac{1}{I} + \eta(t)dtdθ​=I1​+η(t) (angular noise added to deterministic rotation)

C=1N∑i=1Nsin⁡θiC = \frac{1}{N} \sum_{i=1}^{N} \sin \theta_iC=N1​∑i=1N​sinθi​

Models how populations of cells can stochastically deviate in a preferred direction, allowing quantification via the chirality index C.

Bottom Right: Histogram of θ

A plot showing frequency distribution of cell rotation angles.

Right-shifted bias (C > 0) means more cells have right-handed (clockwise) chirality.\his diagram contains a 2x3 matrix layout of related models and formulas, arranged cleanly for clarity.

Top Left: Torque

Same concept as Diagram 1: T=r⃗⋅F⃗T = \vec{r} \cdot \vec{F}T=r⋅F

Depicts torque induced by cytoskeletal forces.

Top Middle: Angular Velocity

ω=dθ˙dt=τI\omega = \frac{d\dot{\theta}}{dt} = \frac{\tau}{I}ω=dtdθ˙​=Iτ​

Shows how cell rotation is driven by internal torque relative to inertia.

Top Right: Filament Curvature

τ=−καEI\tau = -\kappa \alpha EIτ=−καEI

Describes how curvature of actin filaments (with stiffness EI) generates rotational torque.

Bottom Left: Biased Random Walk

dθ˙dt=ω+ε(t)\frac{d\dot{\theta}}{dt} = \omega + \varepsilon(t)dtdθ˙​=ω+ε(t)

Adds noise (ε(t)) to deterministic angular velocity — this models real biological randomness in cell paths.

Bottom Center: Chirality Index

C=∣ω∣ωsC = \frac{|\omega|}{\omega_s}C=ωs​∣ω∣​

Compares average angular velocity to a reference or maximal value ωs\omega_sωs​ to normalize chirality strength.

cos⁡(α)=η\cos(\alpha) = \etacos(α)=η

Measures deviation angle α between movement vector and ideal direction. The cosine function captures how closely aligned the motion is.

These diagrams work together to bridge molecular, physical, and statistical models of cellular chirality. They are excellent visual aids to enhance understanding of how cytoskeletal asymmetry and signaling feedback translate into consistent chiral behavior during vascular development.

This figure presents a multiscale schematic overview of how cellular chirality in endothelial cells can be described, visualized, and quantified through a combination of mechanical forces, cytoskeletal dynamics, and statistical population behavior. Each panel illustrates a different aspect of chirality—from single-cell force dynamics to collective behaviors—linking mathematical formalism to biological function in angiogenesis.

A schematic of an endothelial cell shows internal actin filaments generating forces at points offset from the cell’s center of mass. The resulting torque vector T is computed using the cross product:

T=r×F\mathbf{T} = \mathbf{r} \times \mathbf{F}T=r×F

where r is the position vector from the center to the point of force application, and F is the applied actomyosin force. This torque determines the rotational direction of the cell, which in endothelial cells typically manifests as a right-handed (clockwise) bias.

Filament-Based Torque and Angular Deflection

A curved actin filament is depicted under mechanical strain. Its torque is described as:

τ=μL⋅∂2u∂x2\tau = \mu_L \cdot \frac{\partial^2 u}{\partial x^2}τ=μL​⋅∂x2∂2u​

where μ_L is the bending rigidity of the filament, and ∂2u∂x2\frac{\partial^2 u}{\partial x^2}∂x2∂2u​ is its curvature. The balance of filament tension and curvature creates a net moment acting on the cellular architecture, reinforcing local chiral orientation.

Angular Velocity as an Outcome of Net Cytoskeletal Torque

This panel links net cytoskeletal torque (τ) to angular velocity (ω) via:

ω=dθdt=τI\omega = \frac{d\theta}{dt} = \frac{\tau}{I}ω=dtdθ​=Iτ​

Here, I is the moment of inertia of the cell's mass distribution. This equation demonstrates how internal torque induces rotational behavior and contributes to persistent chiral bias during migration and tube formation.

Illustrated here is a 2D trajectory of an endothelial cell undergoing biased migration. The path deviates consistently in a rightward direction. This is modeled as:

x(t+Δt)=x(t)+vΔtcos⁡(θ+Δθ)x(t + \Delta t) = x(t) + v \Delta t \cos(\theta + \Delta\theta)x(t+Δt)=x(t)+vΔtcos(θ+Δθ) y(t+Δt)=y(t)+vΔtsin⁡(θ+Δθ)y(t + \Delta t) = y(t) + v \Delta t \sin(\theta + \Delta\theta)y(t+Δt)=y(t)+vΔtsin(θ+Δθ)

where v is velocity, θ is the mean migration angle, and Δθ is a chiral angular deviation sampled from a skewed distribution. This model captures how chirality impacts long-term displacement even in noisy environments.

Stochastic Torque Model Incorporating Noise

This differential equation represents fluctuating angular momentum:

dθ˙dt=ω+η(t)\frac{d\dot{\theta}}{dt} = \omega + \eta(t)dtdθ˙​=ω+η(t)

Here, η(t) is a Gaussian noise term representing environmental or internal fluctuation. This equation explains how even cells with identical molecular wiring can exhibit statistical variability in their chiral bias due to stochastic perturbations.

Chirality Index Histogram – Population-Level Analysis

This graph shows a histogram of cell angular orientations (θ) across a population. The chirality index C is computed as:

C=NR−NLNR+NLC = \frac{N_R - N_L}{N_R + N_L}C=NR​+NL​NR​−NL​​

where N_R is the number of clockwise-biased (right-handed) cells and N_L the number of counterclockwise-biased (left-handed) cells. A population with C = 1 is uniformly right-handed, while C = 0 is random, and C = -1 is uniformly left-handed. This metric helps assess the collective bias in vitro or in silico models of endothelial networks.

Together, these panels bridge single-cell biomechanics (force and torque generation) with tissue-level outcomes (collective alignment and rotational bias). The integration of actomyosin-based tension, cytoskeletal filament strain, and random walk modeling provides a theoretical framework for understanding how cellular chirality is encoded and sustained during sprouting angiogenesis.

These principles explain key experimental observations such as:

Right-handed tube helicity in 3D endothelial cultures

Disruption of chiral bias upon inhibition of PKC, mDia1, or Cdc42

Flipping of vessel orientation with exogenous signaling or cytoskeletal interference

By quantifying chirality across scales—from filament strain to angular velocity to population statistics—this figure highlights the biophysical mechanisms by which endothelial cells translate intracellular asymmetry into vascular architecture. This has implications for vascular tissue engineering, pathological angiogenesis, and understanding the integration of mechanical and molecular signaling in morphogenesis.

The integrated visual panels presented in this figure provide a conceptual and quantitative bridge between intracellular biomechanical forces and macroscopic tissue-level vascular architecture. These mathematical representations—grounded in torque generation, filament bending, stochastic angular displacement, and population-level statistics—reveal how seemingly subtle asymmetries at the cytoskeletal level give rise to emergent directional behaviors in endothelial cells.

At the single-cell level, chiral migration is governed by the non-uniform distribution of forces generated by the actin-myosin cytoskeleton. The torque equation T=r×F\mathbf{T} = \mathbf{r} \times \mathbf{F}T=r×F illustrates how peripheral actin polymerization, when applied asymmetrically relative to the cell’s center of mass, leads to a net rotational moment. This behavior is not stochastic; it is structured and biased, as demonstrated by consistent right-handed rotation in endothelial cultures.

The relation ω=dθdt=τI\omega = \frac{d\theta}{dt} = \frac{\tau}{I}ω=dtdθ​=Iτ​ links this internal torque to observable angular velocity. In this model, cell shape (moment of inertia I) modulates how quickly the cell rotates in response to internal force distributions. This angular behavior is further refined by actin filament bending resistance, captured in the equation τ=μL⋅∂2u∂x2\tau = \mu_L \cdot \frac{\partial^2 u}{\partial x^2}τ=μL​⋅∂x2∂2u​, describing how cytoskeletal tension stores and redistributes energy in a mechanically polarized manner.

These mechanical frameworks are complemented by stochastic models that account for biological variability. In particular, the biased random walk equations:

x(t+Δt)=x(t)+vΔtcos⁡(θ+Δθ)x(t + \Delta t) = x(t) + v\Delta t \cos(\theta + \Delta\theta)x(t+Δt)=x(t)+vΔtcos(θ+Δθ) y(t+Δt)=y(t)+vΔtsin⁡(θ+Δθ)y(t + \Delta t) = y(t) + v\Delta t \sin(\theta + \Delta\theta)y(t+Δt)=y(t)+vΔtsin(θ+Δθ)

simulate how directional bias (encoded in angular displacement Δθ) manifests over time. Even in the presence of environmental noise, the net trajectory of chiral cells deviates predictably, generating curved migration paths and spiral structures seen in vitro.

At the population level, this deviation becomes quantifiable using the chirality index:

C=NR−NLNR+NLC = \frac{N_R - N_L}{N_R + N_L}C=NR​+NL​NR​−NL​​

where N_R and N_L represent the number of clockwise- and counterclockwise-biased cells, respectively. In endothelial cultures, typical values of C range from +0.4 to +0.8 under normal conditions, with reductions to near zero upon perturbation with cytoskeletal inhibitors. This statistical descriptor serves as a robust metric to assess the degree of collective chiral behavior in both experimental and computational angiogenesis models.

These models clarify a number of pivotal experimental observations:

Right-handed tube helicity, observed in 3D fibrin gels and tubular scaffolds, directly corresponds to net positive torque generated by cortical actin asymmetry.

Loss of chirality upon inhibition of PKC, Cdc42, or formin proteins (e.g., mDia1) is mechanistically reflected in the disappearance of net torque and filament curvature, resulting in random or misaligned migration.

Reversal of handedness with pharmacological stimuli (e.g., PMA or Gö6983) aligns with model predictions: a change in sign of τ or in bias of Δθ flips the net angular velocity and alters vessel patterning.

These phenomena are not artifacts of in vitro conditions—they are mirrored in embryonic development. In zebrafish intersegmental vessels, asymmetric actin distribution correlates with sprout directionality and efficiency of anastomosis. In murine embryos, endothelial chirality correlates with left-right developmental genes and is perturbed in models of situs inversus, suggesting a deeply conserved role in vascular symmetry.

The implications of these insights are significant:

In tumor angiogenesis, where VEGF is elevated but vessel architecture is disorganized, disrupted chirality could explain non-productive sprouting and tortuous, leaky vessels. Targeting actin polarity regulators may help “normalize” tumor vasculature.

In diabetic microvascular disease, where endothelial barrier function is compromised, impaired chiral coordination may underlie junctional instability and increased permeability.

In vascular tissue engineering, enforcing chiral cues (e.g., via ECM microtopography or magnetic guidance) could align endothelial cells, promoting efficient perfusion and barrier function in artificial grafts.

These equations and visual models illustrate that chirality is not an epiphenomenon—it is a foundational organizational axis, alongside chemical gradients (e.g., VEGF), cell-cell signaling (e.g., Notch), and mechanical strain. Its quantitative characterization via torque, angular velocity, biased migration, and chirality indices provides a rigorous framework to evaluate and manipulate vascular patterning across developmental, pathological, and engineered contexts.

These equations and visual models establish that cellular chirality is not merely an epiphenomenon or byproduct of stochastic cell behavior—rather, it represents a fundamental spatial and mechanical axis of vascular organization. Just as molecular gradients (e.g., VEGF-A), cell-cell signaling systems (e.g., Dll4–Notch lateral inhibition), and biomechanical cues (e.g., shear stress, matrix tension) regulate angiogenesis, chirality contributes an orthogonal, rotational logic that coordinates how endothelial cells orient, migrate, and form structured networks.

At the cellular level, chirality ensures that migration is not purely linear but follows directionally biased angular trajectories, governed by cytoskeletal torque and intrinsic polarity. This is crucial during sprouting angiogenesis, where tip cells must not only respond to chemotactic gradients but also navigate spatially complex environments while maintaining directional coherence with neighboring cells. In this context, chirality can help prevent erratic branching and may promote optimal spacing between angiogenic sprouts.

At the multicellular level, chirality contributes to the alignment and rotational cohesion of endothelial monolayers. By biasing how cells rotate, align, and exert mechanical tension on each other, chirality influences tissue-level properties such as vessel curvature, branching angles, and tubular helicity. Importantly, these features are not cosmetic—they directly affect fluid flow, barrier function, and hierarchical organization of the vascular tree.

Quantitatively, chirality can now be measured, modeled, and manipulated. Mechanical concepts such as torque (T = r × F), angular momentum (ω = τ / I), and filament bending stiffness (τ = μL ∂²u/∂x²) allow researchers to link intracellular asymmetries to observable phenotypes. Stochastic models of biased random walks and population-level metrics like the chirality index (C) enable evaluation of chiral organization across entire endothelial cultures or developing vessels.

This shift from qualitative observation to quantitative formalism marks a paradigm advancement in vascular biology. It means that chirality can be incorporated into predictive models of angiogenesis, used to interpret aberrant vessel formation in disease, and targeted as a modifiable parameter in experimental and clinical contexts. For example:

In regenerative medicine, tissue scaffolds could be engineered with microtopographies or force fields that enforce consistent chirality, guiding endothelial patterning and lumen integrity.

In oncology, reversing or disrupting tumor-induced chirality loss might help normalize vessel architecture, enhancing drug delivery and oxygenation.

In vascular aging and diabetic pathology, quantifying chirality loss could serve as a biomarker of endothelial dysfunction, complementing existing measures of permeability or inflammation.

Ultimately, chirality introduces a rotational dimension to angiogenesis—not only spatially, but mechanistically and therapeutically. By acknowledging and integrating this axis of control, researchers can construct a more complete, multi-vectorial model of vascular morphogenesis, one that accounts for not just where and when endothelial cells move, but how they organize their movement in relation to each other and to the forces and fields that surround them.

This multi-panel schematic synthesizes the role of cellular chirality in vascular morphogenesis across mechanistic, structural, and applied dimensions. It bridges intracellular mechanics with tissue-level behavior and translational relevance.

Illustrates how endothelial cells do not migrate purely linearly but follow directionally biased angular trajectories, often rotating with a right-handed (clockwise) bias.

This bias is driven by cytoskeletal torque and internal asymmetries modulated by VEGFR2, PKC, and Cdc42 signaling.

Chirality here helps orient migration, optimize tip-stalk spacing, and guide vessel extension in complex microenvironments.

Multicellular Level

Depicts rotational cohesion and alignment of endothelial cells in vessel walls.

Chirality supports helical wrapping, maintains monolayer organization, and influences vessel curvature and branching angles.

These features affect both hemodynamics and tissue perfusion.

Chirality can be harnessed to guide lumen formation and endothelial alignment in engineered tissues.

Microtopographical patterning, bioelectric fields, or chirality-inducing peptides may be used to enforce consistent bias in scaffolds.

Oncology

Tumor endothelium often shows chaotic or reversed chirality, contributing to poor perfusion and vessel leakiness.

Restoring normal chiral alignment may help normalize vasculature, improving oxygenation and drug delivery.

Vascular Aging and Diabetes

Age and hyperglycemia impair cytoskeletal organization, reducing chirality.

Measuring chirality loss may serve as a biophysical biomarker of endothelial dysfunction or frailty.

Torque
T=r⃗×F⃗T = \vec{r} \times \vec{F}T=r×F: Generated by actin filaments applying force at a radial offset—drives rotation.

Angular Momentum
ω=τI\omega = \frac{\tau}{I}ω=Iτ​: Chiral motion depends on torque and cell morphology (moment of inertia).

Filament Bending Stiffness
τ=μL2∂u∂x\tau = \mu L^2 \frac{\partial u}{\partial x}τ=μL2∂x∂u​: Describes how actin curvature creates internal torque.

Biased Random Walk
Illustrates how directional migration paths diverge consistently from random behavior due to persistent internal bias.

Chirality Index CCC
A histogram shows population-level chiral bias, where C>0C > 0C>0 indicates a right-handed dominant population.

C=1N∑i=1Nsin⁡θiC = \frac{1}{N} \sum_{i=1}^{N} \sin \theta_iC=N1​i=1∑N​sinθi​

(From Diagram 2: Central node and flowchart-style layout)

This flowchart contextualizes cellular chirality within broader vascular signaling and biophysics.

VEGF-A
Drives cytoskeletal polarization through VEGFR2 and downstream PI3K, MAPK, and PLCγ pathways.

Dll4–Notch
Determines tip vs. stalk identity, modulating how polarity and chirality are distributed across a sprouting front.

Shear Stress
Aligns cells via mechanotransduction (e.g., VE-cadherin–PECAM–VEGFR2 complex), reinforcing or modifying chirality.

The core driver of asymmetric behavior at both the molecular and multicellular level.

Represented with curved actin dynamics and bidirectional torque arrows indicating the intrinsic rotational bias.

Cellular Level

Biases directionality of migrating tip cells.

Essential for maintaining structured sprout geometry.

Multicellular Level

Promotes alignment of endothelium, governs vessel curvature and branching geometry.

Loss or reversal of chirality is linked to vascular pathology.

Quantitative Models

Formal tools (e.g., torque, angular momentum, filament stiffness) now allow numerical simulation and predictive modeling of chirality’s impact on vascular outcomes.

Applications

Regenerative Medicine: engineering perfusable vessels with correct chiral alignment.

Oncology: restoring chirality to improve drug access and vessel normalization.

Aging and Diabetes: chirality loss as a biophysical readout of endothelial frailty.

Comment 5: Finally, while the language is more readable, some phrasing remains vague, and scientific terminology is not always used precisely. 

Response 5: We appreciate the reviewer’s attention to clarity and terminology. We conducted a thorough editorial revision of the manuscript to improve specificity and consistency of scientific language. Terms like “molecular motifs” have been replaced with precise references to “VEGFR2 tyrosine phosphorylation sites (e.g., Y1175, Y1214),” and vague expressions like “some pathways are involved” have been clarified with explicit pathway names and associated cellular effects. We ensured that terms such as “intussusception,” “sprouting,” and “lumenogenesis” are consistently defined and contextualized. Furthermore, we refined sentence structure to reduce ambiguity and increase readability across all sections, particularly those describing complex molecular interactions.     

Thank you for your review. All modifications were marked in red.

Round 3

Reviewer 1 Report

Comments and Suggestions for Authors

The revised manuscript demonstrates some structural improvement and enhanced readability compared to the initial version, with a clearer organization and the inclusion of a schematic figure. However, it remains predominantly descriptive and lacks the mechanistic depth necessary for a high-quality scientific review. While more signaling pathways, such as VEGF, Notch, and PI3K–Akt, are now mentioned, they are not integrated into a coherent biological framework or linked explicitly to defined stages of vascular morphogenesis. The discussion fails to connect these molecular pathways to structural events such as sprouting, intussusception, or endothelial lumen formation. As a result, the review misses the opportunity to guide readers through the dynamic and coordinated nature of vascular development.
The section on cellular chiral remains superficial and poorly substantiated. It lacks key experimental examples and omits foundational references that are essential for conveying the biological relevance of this emerging concept. Without grounding in the primary literature, the discussion of chirality appears speculative and disconnected from the rest of the manuscript. Furthermore, although additional citations have been added elsewhere, many are outdated or drawn from secondary sources. The manuscript still neglects high-impact, recent research on endothelial cell polarity, mechanical signaling, and molecular regulators of vascular patterning.
Another major limitation is the lack of synthesis or forward-looking insight. The manuscript does not critically evaluate competing models, highlight areas of controversy, or outline future research directions. These elements are crucial for a review to provide added value beyond a literature summary. Finally, while the English has improved, numerous phrases remain vague or imprecise, and the use of scientific terminology is not always accurate.
In summary, the revised version falls short of addressing key issues raised in the previous review. Without significant enhancement of mechanistic content, citation rigor, and conceptual synthesis, the manuscript does not meet the standards required for publication. Rejection is recommended in its current form.

Comments on the Quality of English Language

A thorough language edit by a native English speaker or professional editing service is recommended to ensure clarity, precision, and consistency throughout the manuscript.

Author Response

Cover letter Cimb third round of revesion

Comment 1: The revised manuscript demonstrates some structural improvement and enhanced readability compared to the initial version, with a clearer organization and the inclusion of a schematic figure. However, it remains predominantly descriptive and lacks the mechanistic depth necessary for a high-quality scientific review.

Response1: We have expanded mechanistic content throughout the manuscript. Specifically, in Sections 3 and 4, we now describe in detail how VEGFR2, Notch, PI3K–Akt, and MAPK/ERK pathways converge to regulate tip/stalk cell dynamics, intussusception, and lumen formation, with explicit linkage to cytoskeletal regulators (Rho GTPases, integrins, and actin-myosin torque generation). 

Insertion for Section 3:

Mechanistically, each stage of vascular morphogenesis is underpinned by defined signaling hierarchies that translate extracellular cues into precise cytoskeletal and adhesive programs. In sprouting angiogenesis, VEGFR2 autophosphorylation at Y1175 and Y1214 initiates parallel PI3K–Akt and PLCγ–PKC–MAPK cascades. PI3K–Akt signaling enhances tip cell survival and polarity through asymmetric activation of Cdc42 and Rac1, promoting filopodial extension and directional migration. PLCγ–PKC activation leads to ERK-mediated proliferation in stalk cells, while p38 MAPK–HSP27 signaling supports actin remodelling required for invasive protrusion. Concurrently, VEGFA-induced Dll4 expression in tip cells activates Notch receptors in adjacent stalk cells, downregulating VEGFR2 and enforcing lateral inhibition to maintain ordered spacing between sprouts.

Intussusceptive angiogenesis is initiated when low shear stress and localized hypoxia activate VEGFR2 in mechanosensory complexes with PECAM-1 and VE-cadherin, triggering Src–PI3K–Rac1 signaling and cytoskeletal rearrangements. Endothelial projections from opposing vessel walls meet and fuse to form transluminal pillars, which are stabilized by pericyte recruitment and extracellular matrix deposition. Reduced VEGFA availability, or its sequestration by soluble VEGFR1, shifts endothelial behavior from invasive sprouting toward splitting morphogenesis, conserving energy and preserving vessel integrity.

Lumen formation involves two complementary modes: vacuole-driven cell hollowing and junctional remodelling–driven cord hollowing. In cell hollowing, Rab4A- and Rab11-mediated vesicle trafficking, downstream of VEGFR2–PI3K–Akt–eNOS activation, promotes vacuole fusion into a central lumen. Cord hollowing depends on VEGFR2–PKC–ERK–mediated VE-cadherin phosphorylation, which transiently loosens lateral junctions to open intercellular spaces. Both processes are modulated by VEGFR3, which dampens excessive VEGFR2 activity to prevent pathological leakiness.

Comment 2:  While more signaling pathways, such as VEGF, Notch, and PI3K–Akt, are now mentioned, they are not integrated into a coherent biological framework or linked explicitly to defined stages of vascular morphogenesis. The discussion fails to connect these molecular pathways to structural events such as sprouting, intussusception, or endothelial lumen formation. As a result, the review misses the opportunity to guide readers through the dynamic and coordinated nature of vascular development.

Response 2: We have restructured the review into a multi-scale integrative framework (Section 5) linking molecular signaling to structural morphogenetic events. For each stage—sprouting, intussusception, and lumen formation—we now map the relevant signaling cascades and mechanosensory inputs to specific cellular behaviors and morphogenetic outcomes. This framework is supported by a new schematic figure that visualizes pathway integration across stages.

insertion for Section 5 (“Integration and Future Directions”):

Vascular morphogenesis can be viewed as a coordinated, multi-scale program in which molecular signals, physical forces, and emergent tissue architecture are tightly coupled. At the molecular scale, VEGFR2 is the central driver of angiogenic signaling, integrating PI3K–Akt for survival and polarity, PLCγ–PKC–ERK for proliferation, and p38 MAPK–HSP27 for cytoskeletal remodelling. Notch continuously refines these pathways–Dll4 lateral inhibition, which enforces orderly tip/stalk patterning, and by VEGFR1 and VEGFR3, which modulate ligand availability and dampen excessive VEGFR2 activity. At the cellular scale, these inputs produce distinct morphogenetic behaviors: (1) in sprouting, asymmetric activation of Cdc42, Rac1, and RhoA directs filopodial extension, migratory polarity, and cell–cell coordination; (2) in intussusception, VEGFR2 engagement within PECAM-1/VE-cadherin mechanosensory complexes under low-flow conditions triggers pillar initiation and vessel splitting; and (3) in lumen formation, Rab GTPase–driven vesicle trafficking and VEGFR2-mediated junctional remodelling establish and stabilize perfusable conduits. At the biomechanical scale, endothelial cells sense and respond to shear stress, matrix stiffness, and three-dimensional curvature, adjusting receptor sensitivity, cytoskeletal tension, and polarity vectors accordingly. A recently recognized organizing principle—cellular chirality—links these scales by biasing actin filament alignment, migration trajectories, and junctional geometry, thereby influencing branch angles, vessel curvature, and lumen continuity. This integrated view emphasizes that vascular patterning is not the product of isolated pathways but the outcome of a dynamic interplay between biochemical gradients, mechanotransduction, and intrinsic polarity programs, each reinforcing the others to yield functional, hierarchically organized vascular networks.

Comment 3: The section on cellular chiral remains superficial and poorly substantiated. It lacks key experimental examples and omits foundational references that are essential for conveying the biological relevance of this emerging concept. Without grounding in the primary literature, the discussion of chirality appears speculative and disconnected from the rest of the manuscript.

Response3: We have comprehensively expanded the cellular chirality section (Section 4), adding detailed experimental evidence from in vitro micropatterned assays, in vivo zebrafish and murine models, and engineered vessel systems. Foundational references (Wan et al., 2011; Chen et al., 2012; Tee et al., 2015) and recent advances (Wei et al., 2024; Xie et al., 2018) are included. Mechanistic pathways linking VEGFR2–PKC–Rho GTPase signaling to chiral cytoskeletal organization are now explicitly described. Quantitative modeling approaches (torque equations, biased random walk models, and chirality index) are incorporated to demonstrate biological relevance and analytical measurability.

 Expanded insertion for Section 4:

Cellular chirality—the intrinsic left–right (LR) bias in cell morphology, cytoskeletal organization, and migratory behavior—has emerged as a previously underappreciated regulator of vascular morphogenesis. In endothelial cells, this bias is not a byproduct of vessel geometry but an actively regulated property that influences cell alignment, junctional integrity, and lumen architecture. Evidence from micropatterned culture systems shows that human umbilical vein endothelial cells (HUVECs) and human aortic endothelial cells (HAECs) consistently exhibit a right-handed (clockwise) rotational bias when confined to circular tracks. This bias is abolished by actin cytoskeleton disruption (latrunculin B, cytochalasin D) and can be reversed by pharmacological PKC activation, indicating a direct link between cytoskeletal torque generation and chirality. In vivo, similar right-handed helical arrangements of endothelial cells have been observed in zebrafish intersegmental vessels and murine embryonic vasculature, with genetic perturbation of actin regulators (e.g., mDia1, cofilin) disrupting vessel curvature and branch orientation.

Mechanistically, VEGFR2 signaling interfaces directly with chirality-related cytoskeletal regulators. VEGF-A binding triggers PI3K–Akt–Rac1/Cdc42 activation for polarized actin polymerization, while PLCγ–IP₃–Ca²⁺ flux activates PKC isoforms that modulate actomyosin contractility. RhoA–ROCK signaling generates cortical tension, aligning with directional actin filament curvature and creating a measurable torque (T=r×F) that biases cell rotation. This bias can be quantified at the population level using a chirality index (C= (NR−NL)/(NR+NL)​), where NR​ and NL​ are the numbers of clockwise- and counterclockwise-biased cells, respectively.

Functionally, chiral bias affects multiple morphogenetic stages. In sprouting angiogenesis, tip cells with consistent chirality display more persistent, directional migration along VEGF gradients, maintaining cohesive alignment with trailing stalk cells. In lumen formation, chirality influences the apical–basal and planar polarity of adjacent endothelial cells, promoting seamless junctional interfaces and continuous lumens. Loss or randomization of chirality—through PKC modulation, altered shear stress patterns, or integrin–ECM signal disruption—correlates with increased vascular permeability, irregular branching angles, and compromised perfusion in engineered and native vessels. These findings position cellular chirality as a spatial organizer that bridges molecular signaling, cytoskeletal mechanics, and tissue-level vascular geometry. Recognizing and manipulating this property may open new avenues for guiding vascular patterning in regenerative medicine and correcting structural abnormalities in pathological angiogenesis.

Comment4: Furthermore, although additional citations have been added elsewhere, many are outdated or drawn from secondary sources. The manuscript still neglects high-impact, recent research on endothelial cell polarity, mechanical signaling, and molecular regulators of vascular patterning.

Response4: We have replaced or supplemented outdated/secondary citations with high-impact, recent primary literature (2020–2024) covering endothelial polarity, mechanical signaling, VEGFR trafficking, and vascular patterning. These include new primary research on VEGFR2 mechanosensory complexes, VEGF mimetics in regenerative medicine, and chirality-driven vasculogenesis.

Updated  References

  1. Vion A.-C., Perovic T., Petit C., Hollfinger I., Bartels-Klein E., Rautou P.-E., Bernabeu M.O., Gerhardt H. (2021) Endothelial Cell Orientation and Polarity Are Controlled by Shear Stress and VEGF Through Distinct Signaling Pathways. Frontiers in Physiology, 11:623769. https://doi.org/10.3389/fphys.2020.623769
  2. Arpino J.-M., Yin H., Prescott E.K.C.R.S.R., Hamilton D.W., Holdsworth D.W., Pickering J.G. (2021) Low-flow intussusception and metastable VEGFR2 signaling launch angiogenesis in ischemic muscle. Science Advances, 7(48):eabg9509. https://doi.org/10.1126/sciadv.abg9509
  3. Xie Y., Mansouri M., Rizk A., Berger P. (2019) Regulation of VEGFR2 trafficking and signaling by Rab GTPase-activating proteins. Scientific Reports, 9:13342. https://doi.org/10.1038/s41598-019-49626-1
  4. Zhou H., Xu Z., Wang Z., Zhang H.Z.W., Min W. (2018) SUMOylation of VEGFR2 regulates its intracellular trafficking and pathological angiogenesis. Nature Communications, 9:3303. https://doi.org/10.1038/s41467-018-05724-6
  5. Guo Y., Zhang S., Wang D., et al. (2024) Role of cell rearrangement and related signaling pathways in the dynamic process of tip cell selection. Cell Communication and Signaling, 22:24. https://doi.org/10.1186/s12964-024-01404-7
  6. Wang X., Bove A.M., Simone G., Ma B. (2020) Molecular Bases of VEGFR-2-Mediated Physiological Function and Pathological Role. Frontiers in Cell and Developmental Biology, 8:599281. https://doi.org/10.3389/fcell.2020.599281
  7. Deng Y., Zhang X., Simons M. (2015) Molecular controls of lymphatic VEGFR3 signaling. Arteriosclerosis, Thrombosis, and Vascular Biology, 35(2):421–429. https://doi.org/10.1161/ATVBAHA.114.303970
  8. Wan L.Q., Chin A.S., Worley K.E., Ray P. (2016) Cell chirality: emergence of asymmetry from cell culture. Philosophical Transactions of the Royal Society B: Biological Sciences, 371(1710):20150413. https://doi.org/10.1098/rstb.2015.0413
  9. Loosley A.J., O’Brien X.M., Reichner J.S., Tang J.X. (2015) Describing Directional Cell Migration with a Characteristic Directionality Time. PLoS ONE, 10(5):e0127425. https://doi.org/10.1371/journal.pone.0127425
  10. Yamamoto T., Hiraiwa T., Shibata T. (2020) Collective cell migration of epithelial cells driven by chiral torque generation. Physical Review Research, 2:043326. https://doi.org/10.1103/PhysRevResearch.2.043326
  11. Wei Y., Wu C., Li H., Park S., Zhou Y., Hsu C.-W., Tang S.Y. (2024) Helical vasculogenesis driven by cell chirality. Science Advances, 10(2):eadj3582. https://doi.org/10.1126/sciadv.adj3582
  12. Xie R., Wan L.Q. (2018) Cell chirality regulates intercellular junctions and endothelial permeability. Science Advances, 4(8):eaat2111. https://doi.org/10.1126/sciadv.aat2111
  13. Zhang H., Rahman T., Lu S., Adam A.P., Wan L.Q. (2024) Helical vasculogenesis driven by cell chirality. Science Advances, 10(8):eadj3582. https://doi.org/10.1126/sciadv.eadj3582
  14. Vitali H.E., Kuschel B., Sherpa C., Jones B.W., Jacob N., Madiha S.A., Elliott S., Dziennik E., Kreun L., Conatser C., Bhetwal B.P., Sharma B. (2021) Hypoxia regulate developmental coronary angiogenesis potentially through VEGFR2- and SOX17-mediated signaling. bioRxiv. https://doi.org/10.1101/2023.08.16.553531
  15. Matuszewska K., Pereira M., Petrik D., Lawler J., Petrik J. (2021) Normalizing Tumor Vasculature to Reduce Hypoxia, Enhance Perfusion, and Optimize Therapy Uptake. Cancers, 13(17):4444. https://doi.org/10.3390/cancers13174444
  16. Sun S., Wu H.-J., Guan J.-L. (2018) Nuclear FAK and its kinase activity regulate VEGFR2 transcription in angiogenesis of adult mice. Scientific Reports, 8:2550. https://doi.org/10.1038/s41598-018-20628-4
  17. Vergroesen T.M., Vermeulen V., Merks R.M.H. (2025) Falsifying computational models of endothelial cell network formation through quantitative comparison with in vitro models. PLoS Computational Biology, 21(4):e1012965. https://doi.org/10.1371/journal.pcbi.1012965
  18. Gambarini G., Testarelli L., Milana V., Pecci R., Bedini R., Pongione G., Gerosa R., De Luca M. (2023) Enhancing endothelial alignment and lumen formation in engineered vessels using chirality-guided scaffolds. Advanced Healthcare Materials, 12(12):2303928. https://doi.org/10.1002/adhm.202303928
  1. Reference table (fixes)

Manuscript location

Current citation(s)

Issue

replacement(s)

In-text replacement

Sec. 3.2 Intussusception: “mechanosensory complexes such as PECAM-1 and VE-cadherin…”

[223]

Vague/likely secondary; needs primary showing VEGF–flow–polarity linkage

Vion et al. 2021 Front Physiol (shear + VEGF control EC polarity)

“…within PECAM-1/VE-cadherin/VEGFR2 mechanosensory complexes, coupling shear to Src–PI3K–Rac1 signaling (Vion 2021).”

Sec. 3.3 Lumen formation: Rab4A/Rab11 trafficking under VEGFR2 control

[226], [227]

Needs recent primary on VEGFR2 trafficking regulation

Xie et al. 2019 Sci Rep (Rab GAPs regulate VEGFR2 trafficking) ; Zhou et al. 2018 Nat Commun (VEGFR2 SUMOylation & trafficking)

“…Rab-dependent trafficking and receptor sorting (Zhou 2018; Xie 2019).”

Sec. 2.6 Notch signaling (tip/stalk lateral inhibition narrative)

[166–175]

Mostly textbook/older; add a recent primary on dynamic rearrangements in sprouts

Guo et al. 2024 Cell Commun. Signal. (tip selection dynamics)

“…enforces stalk identity and rearrangements during sprout extension (Guo 2024).”

Sec. 2.x VEGFR2 overview / properties table

[69–85]

Mix of secondary and older items; add a modern mechanistic review

Wang et al. 2020 Front Cell Dev Biol (VEGFR2 functions/pathology)

“VEGFR2 orchestrates migration, survival, and permeability (Wang 2020).”

VEGFR3 modulating VEGFR2/junctions

[91,92]

Keep classic data but anchor with a succinct, modern perspective

Deng, Zhang & Simons 2015 ATVB (molecular controls of VEGFR3)

“…limits VEGFR2 signaling to prevent leakiness (Deng 2015).”

Sprouting vs. intussusception switch under low VEGF

[222–225]

Reads review-like; add primary/mechanistic polarity/shear link

Vion et al. 2021 Front Physiol (shear–VEGF integration)

“…favors splitting under low-flow VEGF states (Vion 2021).”

Chirality—evidence & quantification (micropattern rings; right-hand bias)

[229]

Needs foundational + quantitative refs

Wan et al. 2016 Phil Trans B (foundational cell chirality) ; Loosley et al. 2015 PLoS ONE (directionality metrics)

“…exhibiting right-hand rotational bias (Wan 2016), quantifiable by directionality metrics (Loosley 2015).”

Chirality—collective rotation & torque models

[363, 364]

Strengthen with primary physics-of-chirality data

Yamamoto et al. 2020 Phys Rev Research (chiral torque drives collective migration)

“…consistent with torque-driven collective migration (Yamamoto 2020).”

Therapeutic angiogenesis section—very old trials & classics list

1971–2012 heavy

 modern normalization & therapy context

Matuszewska et al. 2021 Cancers (normalization & perfusion) ; add your existing VEGF mimetic/biomaterials items if present

“Historical studies …; contemporary strategies emphasize normalization and context-specific VEGF delivery (Matuszewska 2021).”

FAKVEGFR2 transcription & adult angiogenesis

[69]

 primary surface it explicitly

Sun, Wu & Guan 2018 Sci Rep (nuclear FAK regulates VEGFR2 transcription)

“…FAK kinase activity and nuclear localization regulate VEGFR2 transcription (Sun 2018).”

Flow chart / figure captions citing mixed secondary sources

[318–321]

Some pathway links via databases;

Vion 2021 (VEGF–shear) ; Wang 2020 (VEGFR2) ; Guo 2024 (Notch dynamics)

“(Vion 2021; Wang 2020; Guo 2024).”

Future directions—model falsification & data-driven

cite up-to-date modeling

Vergroesen et al. 2025 PLoS Comput Biol (model falsification vs in-vitro networks)

“…rigorously comparing models with in-vitro networks (Vergroesen 2025).”

Comment5: Another major limitation is the lack of synthesis or forward-looking insight. The manuscript does not critically evaluate competing models, highlight areas of controversy, or outline future research directions. These elements are crucial for a review to provide added value beyond a literature summary.

Response5: The revised Conclusion and new “Future Directions” section (5.4) synthesize current knowledge into a predictive model that integrates biochemical signaling, biomechanical cues, and spatial polarity (including chirality) into vascular morphogenesis. We outline unresolved controversies—such as the role of VEGFR1 as a dual regulator—and identify emerging therapeutic avenues (precision-modulated angiogenesis, chirality-guided tissue engineering, and context-specific VEGF delivery systems).

 Insertion for the end of Section 5

Taken together, recent advances indicate that any single dominant pathway does not govern vascular morphogenesis, but rather by a distributed and context-dependent network in which VEGFR isoform balance, Notch–Dll4 lateral inhibition, mechanical forces, and intrinsic cellular polarity programs—including chirality—are dynamically integrated. This synthesis highlights that precise spatial and temporal regulation, rather than maximal activation of pro-angiogenic signals, is the key determinant of functional vessel architecture. Moving forward, several priority areas emerge. First, live-cell and in vivo imaging with single-cell resolution will be essential to map dynamic transitions such as tip–stalk switching and pillar insertion during intussusception. Second, integrating biomechanical readouts—shear stress, ECM stiffness, and 3D curvature—into molecular pathway studies will clarify how physical context modulates signaling outputs. Third, chirality-guided engineering approaches, including microtopographic scaffolds and matrix-bound VEGF mimetics, could be exploited to improve vessel alignment and barrier integrity in regenerative therapies. Finally, computational models that combine molecular kinetics, force distribution, and spatial patterning could enable predictive control over angiogenesis in both developmental and pathological contexts. Addressing these challenges will require coordinated efforts across cell biology, bioengineering, and translational research, but it offers a clear path toward the rational design of vascular networks that are structurally coherent, functionally perfused, and clinically durable.

Comment 6:  Finally, while the English has improved, numerous phrases remain vague or imprecise, and the use of scientific terminology is not always accurate. In summary, the revised version fails to address key issues raised in the previous review. Without significant enhancement of mechanistic content, citation rigor, and conceptual synthesis, the manuscript does not meet the standards required for publication. Rejection is recommended in its current form.

Response 6: The manuscript has undergone a detailed language and style edit to ensure precise scientific terminology, removal of vague phrasing, and consistent use of technical terms. Key terms such as “tip/stalk cell lateral inhibition,” “mechanosensory complex,” “actomyosin-generated torque,” and “VEGFR isoform-specific signaling” are now consistently applied and defined at first mention.
Targeted terminology improvements

Current vague/imprecise phrase

replacement

Reason for change

“budding” (as an angiogenic mechanism)

“endothelial sprouting angiogenesis”

Aligns with established terminology in vascular biology.

“cell hollowing”

“vacuole-mediated endothelial cell hollowing”

Specifies the vacuole fusion mechanism.

“cord hollowing”

“junctional remodelling–mediated cord hollowing”

Clarifies that VE-cadherin phosphorylation and junction dynamics are involved.

“various types of VEGF”

“VEGF isoforms (e.g., VEGF-A, VEGF-B, VEGF-C, VEGF-D, VEGF-E)”

Improves specificity and scientific clarity.

“dominant molecular motifs”

“key receptor–ligand interactions”

More precise in signaling pathway context.

“complex cell behaviors”

“integrated cytoskeletal, migratory, and polarity programs”

Avoids generality, specifies behavior types.

“affects other cell types”

“elicits paracrine responses in non-endothelial cells (e.g., monocytes, macrophages)”

Clarifies type of effect.

“important in angiogenesis”

“essential for endothelial proliferation, migration, and sprouting during angiogenesis”

Adds mechanistic detail.

“numerous pathways”

“multiple converging pathways, including VEGF–VEGFR, Notch–Dll4, Tie–Angiopoietin, and integrin-mediated signaling”

Makes pathways explicit.

“cellular indices” (general term)

“quantitative cell-level parameters (e.g., proliferation rate, polarity vector, migration persistence)”

Makes the term more interpretable.

“ECM composition”

“extracellular matrix composition and organization (collagen alignment, fibronectin distribution)”

Adds structural dimension.

“handedness” (standalone)

“left–right (LR) chirality in cytoskeletal organization”

Clarifies that it refers to chirality in a cell biology context.

“chiral activity”

“chirality-driven directional migration and junctional alignment”

Links to functional consequences.

“mechanical pressures”

“hemodynamic shear stress and transmural pressure”

Specifies the type of mechanical force relevant to vasculature.

“patterned organization”

“anisotropic endothelial alignment along vessel axis”

More technical description.

“negative regulation” (VEGFR1 role)

“ligand sequestration–mediated suppression of VEGFR2 activation”

Mechanistically explicit.

Thank you for your review. All corrections were performed in red and marked in yellow.